# Spectral Entry-wise Matrix Estimation for Low-Rank Reinforcement Learning

**Stefan Stojanovic**
EECS
KTH, Stockholm, Sweden
`stesto@kth.se`

**Yassir Jedra**
EECS
KTH, Stockholm, Sweden
`jedra@kth.se`

**Alexandre Proutiere**
EECS
KTH, Stockholm, Sweden
`alepro@kth.se`

## Abstract

We study matrix estimation problems arising in reinforcement learning (RL) with low-rank structure. In low-rank bandits, the matrix to be recovered specifies the expected arm rewards, and for low-rank Markov Decision Processes (MDPs), it may for example characterize the transition kernel of the MDP. In both cases, each entry of the matrix carries important information, and we seek estimation methods with low entry-wise error. Importantly, these methods further need to accommodate for inherent correlations in the available data (e.g. for MDPs, the data consists of system trajectories). We investigate the performance of simple spectral-based matrix estimation approaches: we show that they efficiently recover the singular subspaces of the matrix and exhibit nearly-minimal entry-wise error. These new results on low-rank matrix estimation make it possible to devise reinforcement learning algorithms that fully exploit the underlying low-rank structure. We provide two examples of such algorithms: a regret minimization algorithm for low-rank bandit problems, and a best policy identification algorithm for reward-free RL in low-rank MDPs. Both algorithms yield state-of-the-art performance guarantees.

## 1 Introduction

Learning succinct representations of the reward function or of the system state dynamics in bandit and RL problems is empirically known to significantly accelerate the search for efficient policies [38, 55, 13]. It also comes with interesting theoretical challenges. The design of algorithms learning and leveraging such representations and with provable performance guarantees has attracted considerable attention recently, but remains largely open. In particular, significant efforts have been made towards such design when the representation relies on a low-rank structure. In bandits, assuming such a structure means that the arm-to-reward function can be characterized by a low-rank matrix [37, 32, 7, 27]. In MDPs, it implies that the reward function, the $Q$-function or the transition kernels are represented by low-rank matrices [56, 4, 46, 60, 49]. In turn, the performance of algorithms exploiting low-rank structures is mainly determined by the accuracy with which we are able to estimate these matrices.

In this paper, we study matrix estimation problems arising in low-rank bandit and RL problems. Two major challenges are associated with these problems. (i) The individual entries of the matrix carry important operational meanings (e.g. in bandits, an entry could correspond to the average reward of an arm), and we seek estimation methods with low entry-wise error. Such requirement calls for a

37th Conference on Neural Information Processing Systems (NeurIPS 2023).

fine-grained analysis, typically much more involved than that needed to only upper bound the spectral or Frobenius norm of the estimation error [22, 21, 12, 2, 54, 15, 53, 47]. (ii) Our estimation methods should further accommodate for inherent correlations in the available data (e.g., in MDPs, we have access to system trajectories, and the data is hence Markovian). We show that, essentially, spectral methods successfully deal with these challenges.

**Contributions.** 1) We introduce three matrix estimation problems. The first arises in low-rank bandits. The second corresponds to scenarios in RL where the learner wishes to estimate the (low-rank) transition kernel of a Markov chain and to this aim, has access to a generative model. The last problem is similar but assumes that the learner has access to system trajectories only, a setting referred to as the forward model in the RL literature. For all problems, we establish strong performance guarantees for simple spectral-based estimation approaches: these efficiently recover the singular subspaces of the matrix and exhibit nearly-minimal entry-wise error. To prove these results, we develop and combine involved leave-one-out arguments and Poisson approximation techniques (to handle the correlations in the data).

2) We apply the results obtained for our first matrix estimation problem to devise an efficient regret-minimization algorithm for low-rank bandits. We prove that the algorithm enjoys finite-time performance guarantees, with a regret at most roughly scaling as $(m + n) \log^3(T) \bar{\Delta} / \Delta_{\min}^2$ where $(m, n)$ are the reward matrix dimensions, $T$ is the time horizon, $\bar{\Delta}$ is the average of the reward gaps between the best arm and all other arms, and $\Delta_{\min}$ is the minimum of these gaps.

3) Finally, we present an algorithm for best policy identification in low-rank MDPs in the reward-free setting. The results obtained for the second and last matrix estimation problems imply that our algorithm learns an $\epsilon$-optimal policy for any reward function using only a number of samples scaling as $O(nA/\epsilon^2)$ up to logarithmic factors, where $n$ and $A$ denote the number of states and actions, respectively. This sample complexity is mini-max optimal [28], and illustrates the gain achieved by leveraging the low-rank structure (without this structure, the sample complexity would be $\Omega(n^2 A/\epsilon^2)$).

**Notation.** For any matrix $A \in \mathbb{R}^{m \times n}$, $A_{i,:}$ (resp. $A_{:,j}$) denotes its $i$-th row (resp. its $j$-th column), $A_{\min} = \min_{(i,j)} A_{i,j}$ and $A_{\max} = \max_{(i,j)} A_{i,j}$. We consider the following norms for matrices: $\|A\|$ denotes the spectral norm, $\|A\|_{1\to\infty} = \max_{i \in [m]} \|A_{i,:}\|_1$, $\|A\|_{2\to\infty} = \max_{i \in [m]} \|A_{i,:}\|_2$, and finally $\|A\|_\infty = \max_{(i,j) \in [m] \times [n]} |A_{i,j}|$. If the SVD of $A$ is $U\Sigma V^\top$, we denote by $\mathrm{sgn}(A) = UV^\top$ the matrix sign function of $A$ (see Definition 4.1 in [14]). $\mathcal{O}^{r \times r}$ denotes the set of $(r \times r)$ real orthogonal matrices. For any finite set $\mathcal{S}$, let $\mathcal{P}(\mathcal{S})$ be the set of distributions over $\mathcal{S}$. The notation $a(n, m, T) \lesssim b(n, m, T)$ (resp. $a(n, m, T) = \Theta(b(n, m, T))$) means that there exists a universal constant $C > 0$ (resp. $c, C > 0$) such that $a(n, m, T) \leq Cb(n, m, T)$ (resp. $cb(n, m, T) \leq a(n, m, T) \leq Cb(n, m, T)$) for all $n, m, T$. Finally, we use $a \wedge b = \min(a, b)$ and $a \vee b = \max(a, b)$.

## 2 Models and Objectives

Let $M \in \mathbb{R}^{m \times n}$ be an unknown rank $r$ matrix that we wish to estimate from $T$ noisy observations of its entries. We consider matrices arising in two types of learning problems with low-rank structure, namely low-rank bandits and RL. The SVD of $M$ is $U\Sigma V^\top$ where the matrices $U \in \mathbb{R}^{m \times r}$ and $V \in \mathbb{R}^{n \times r}$ contain the left and right singular vectors of $M$, respectively, and $\Sigma = \mathrm{diag}(\sigma_1, \ldots, \sigma_r)$. We assume without loss of generality that the singular values have been ordered, i.e., $\sigma_1 \geq \ldots \geq \sigma_r$. The accuracy of our estimate $\widehat{M}$ of $M$ will be assessed using the following criteria:

- *(i)* *Singular subspace recovery.* Let the SVD of $\widehat{M}$ be $\widehat{U}\widehat{\Sigma}\widehat{V}^\top$. To understand how well the singular subspaces of $M$ are recovered, we will upper bound $\min_{O \in \mathcal{O}^{r \times r}} \|U - \widehat{U}O\|_{2\to\infty}$ and $\min_{O \in \mathcal{O}^{r \times r}} \|V - \widehat{V}O\|_{2\to\infty}$ (the $\min_{O \in \mathcal{O}^{r \times r}}$ problem corresponds to the orthogonal Procrustes problem and its solution aligns $\widehat{U}$ and $U$ as closely as possible, see Remark 4.1 in [14]).

- *(ii)* *Matrix estimation.* To assess the accuracy of $\widehat{M}$, we will upper bound the row-wise error $\|\widehat{M} - M\|_{1\to\infty}$ or $\|\widehat{M} - M\|_{2\to\infty}$, as well as the entry-wise error $\|\widehat{M} - M\|_\infty$ (the spectral error $\|\widehat{M} - M\|$ is easier to deal with and is presented in appendix only).

We introduce two classical quantities characterizing the heterogeneity and incoherence of the matrix $M$ [11, 48]. Let $\kappa = \sigma_1/\sigma_r$, and let $\mu(U) = \sqrt{m/r}\|U\|_{2\to\infty}$ (resp. $\mu(V) = \sqrt{n/r}\|V\|_{2\to\infty}$) denote the row-incoherence (resp. column-incoherence) parameter of $M$. Let $\mu = \max\{\mu(U), \mu(V)\}$. Next, we specify the matrices $M$ of interest in low-rank bandits and RL, and the way the data used for their estimation is generated.

**Model I: Reward matrices in low-rank bandits.** For bandit problems, $M$ corresponds to the average rewards of various arms. To estimate $M$, the learner has access to data sequentially generated as follows. In each round $t = 1, \dots, T$, an arm $(i_t, j_t) \in [m] \times [n]$ is randomly selected (say uniformly at random for simplicity) and the learner observes $M_{i_t,j_t} + \xi_t$, an unbiased sample of the corresponding entry of $M$. $(\xi_t)_{t\geq 1}$ is a sequence of zero-mean and bounded random variables. Specifically, we assume that for all $t \geq 1$, $|\xi_t| \leq c_1\|M\|_\infty$ a.s., for some constant $c_1 > 0$.

**Model II: Transition matrices in low-rank MDPs.** In low-rank MDPs, we encounter Markov chains whose transition matrices have low rank $r$ (refer to Section 5 for details). Let $P \in \mathbb{R}^{n \times n}$ be such a transition matrix. We assume that the corresponding Markov chain is irreducible with stationary distribution $\nu$. The objective is to estimate $P$ from the data consisting of samples of transitions of the chain. More precisely, from the data, we will estimate the *long-term frequency matrix* $M = \mathrm{diag}(\nu)P$ ($M_{ij}$ is the limiting proportion of transitions from state $i$ to state $j$ as the trajectory grows large). Observe that $M$ is of rank $r$, and that $P_{i,:} = M_{i,:}/\|M_{i,:}\|_1$. To estimate $M$, the learner has access to the data $(x_1, \dots, x_T) \in [n]^T$ generated according to one of the following two models.

(a) In the *generative* model, for any $t \in [T]$, if $t$ is odd, $x_t$ is selected at random according to some distribution $\nu_0$, and $x_{t+1}$ is sampled from $P_{x_t,:}$.

(b) In the *forward* model, the learner has access to a trajectory $(x_1, \dots, x_T)$ of length $T$ of the Markov chain, where $x_1 \sim \nu_0$ and for any $t \geq 1$, $x_{t+1} \sim P_{x_t,:}$.

## 3 Matrix Estimation via Spectral Decomposition

In the three models (Models I, II(a) and II(b)), we first construct a matrix $\widetilde{M}$ directly from the data, and from there, we build our estimate $\widehat{M}$, typically obtained via spectral decomposition, i.e., by taking the best rank-$r$ approximation of $\widetilde{M}$. In the remaining of this section, we let $\widehat{U}\widehat{\Sigma}\widehat{V}^\top$ denote the SVD of $\widehat{M}$. Next, we describe in more details how $\widehat{M}$ is constructed in the three models, and analyze the corresponding estimation error.

### 3.1 Reward matrices

For Model I, for $t = 1, \dots, T$, we define $\widetilde{M}_t = \left((M_{i_t,j_t} + \xi_t)\mathbb{1}_{\{(i,j)=(i_t,j_t)\}}\right)_{i,j\in[m]\times[n]}$ and $\widetilde{M} = \frac{nm}{T}\sum_{t=1}^T \widetilde{M}_t$. Let $\widehat{M}$ denote the best rank-$r$ approximation of $\widetilde{M}$.

**Theorem 1.** *Let $\delta > 0$. We introduce:*

$$\mathcal{B} = \sqrt{\frac{nm}{T}}\left(\sqrt{(n+m)\log\left(\frac{e(n+m)T}{\delta}\right)} + \log^{3/2}\left(\frac{e(n+m)T}{\delta}\right)\right).$$

*Assume that $T \geq c\mu^4\kappa^2 r^2(n+m)\log^3(e(m+n)T/\delta)$ for some universal constant $c > 0$. Then there exists a universal constant $C > 0$ such that the following inequalities hold with probability at least $1 - \delta$:*

$$(i) \qquad \max\left(\|U - \widehat{U}(\widehat{U}^\top U)\|_{2\to\infty}, \|V - \widehat{V}(\widehat{V}^\top V)\|_{2\to\infty}\right) \leq C\frac{(\mu^3\kappa^2 r^{3/2})}{\sqrt{mn(n\wedge m)}}\mathcal{B},$$

$$(ii) \qquad \|\widehat{M} - M\|_{2\to\infty} \leq C\frac{(\mu^3\,\kappa^2 r^{3/2})}{\sqrt{m\wedge n}}\|M\|_\infty\mathcal{B},$$

$$(iii) \qquad \|\widehat{M} - M\|_\infty \leq C\left(\mu^{11/2}\,\kappa^2 r^{1/2} + \mu^3\kappa r^{3/2}\frac{m+n}{\sqrt{mn}}\right)\frac{1}{(n\wedge m)}\|M\|_\infty\mathcal{B}.$$

**Corollary 2.** *(Homogeneous reward matrix) When $m = \Theta(n)$, $\kappa = \Theta(1)$, $\mu = \Theta(1)$, $\|M\|_\infty = \Theta(1)$, $r = \Theta(1)$, we say that the reward matrix $M$ is homogeneous. In this case, for any $\delta > 0$, when $T \geq c(n + m) \log^3 \left( e(m + n)T/\delta \right)$ for some universal constant $c > 0$, we have with probability at least $1 - \delta$:*

$$\max \left( \|U - \widehat{U}(\widehat{U}^\top U)\|_{2 \to \infty}, \|V - \widehat{V}(\widehat{V}^\top V)\|_{2 \to \infty} \right) \lesssim \frac{1}{\sqrt{T}} \log^{3/2} \left( \frac{(n + m)T}{\delta} \right),$$

$$\|\widehat{M} - M\|_{2 \to \infty} \lesssim \frac{(n + m)}{\sqrt{T}} \log^{3/2} \left( \frac{(n + m)T}{\delta} \right),$$

$$\|\widehat{M} - M\|_\infty \lesssim \sqrt{\frac{(n + m)}{T}} \log^{3/2} \left( \frac{(n + m)T}{\delta} \right).$$

For a homogeneous reward matrix, $\|U\|_{2 \to \infty} = \Theta(1/\sqrt{m})$ and $\|M\|_\infty = \Theta(1)$, and hence, from the above corollary, we obtain estimates whose relative errors (e.g., $\|\widehat{M} - M\|_\infty/\|M\|_\infty$) scale at most as $\sqrt{m/T}$ up to the logarithmic factor.

We may also compare the results of the above corollary to those of Theorem 4.4 presented in [14]. There, the data consists for each pair $(i, j)$ of a noisy observation $M_{i,j} + E_{i,j}$. The $E_{i,j}$'s are independent across $(i, j)$. This model is simpler than ours and does not include any correlation in the data. But it roughly corresponds to the case where $T = nm$ in our Model I. Despite having to deal with correlations, we obtain similar results as those of Theorem 4.4: for example, $\|\widehat{M} - M\|_\infty \lesssim \sqrt{1/(n + m)}$ (up to logarithmic terms) with high probability.

## 3.2 Transition matrices under the generative model

For Model II(a), the matrix $\widetilde{M}$ records the empirical frequencies of the transitions: for any pair of states $(i, j)$, $\widetilde{M}_{i,j} = \frac{1}{\lfloor T/2 \rfloor} \sum_{k=1}^{\lfloor T/2 \rfloor} \mathbb{1}_{\{(x_{2k-1}, x_{2k}) = (i,j)\}}$. $\widehat{M}$ is the best rank-$r$ approximation of $\widetilde{M}$ and the estimate $\widehat{P}$ of the transition matrix $P$ is obtained normalizing the rows of $\widehat{M}$: for all $i \in [n]$,

$$\widehat{P}_{i,:} = \begin{cases} (\widehat{M}_{i,:})_+ / \|(\widehat{M}_{i,:})_+\|_1, & \text{if } \|(\widehat{M}_{i,:})_+\|_1 > 0, \\ \frac{1}{n} \mathbf{1}_n, & \text{if } \|(\widehat{M}_{i,:})_+\|_1 = 0. \end{cases} \tag{1}$$

where $(\cdot)_+$ is the function applying $\max(0, \cdot)$ component-wise and $\mathbf{1}_n$ is the $n$-dimensional vector of ones. The next theorem is a simplified version and a consequence of a more general and tighter theorem presented in App. B.2. To simplify the presentation of our results, we define

$$g(M, T, \delta) = n \log(\tfrac{n\sqrt{T}}{\delta}) \max \left\{ \mu^6 \kappa^6 r^3, \frac{\log(\frac{n\sqrt{T}}{\delta}) \mathbb{1}_{\{\exists \ell : T \|M_{\ell,:}\|_\infty \leq 1\}}}{\log(1 + \frac{1}{T \|M\|_\infty})} \right\}.$$

**Theorem 3.** *Let $\delta > 0$. Introduce $\mathcal{B} = \mu \kappa \sqrt{(r\|M\|_\infty/T) \log(n\sqrt{T}/\delta)}$. Assume that we have $(\nu_0)_{\min} = \min_{i \in [n]} (\nu_0)_i > 0$. If (a) $n \geq c \log^2(nT^{3/2}/\delta)$ and (b) $T \geq cg(M, T, \delta)$ for some universal constant $c > 0$, then there exists a universal constant $C > 0$ such that the following inequalities hold with probability at least $1 - \delta$:*

$$(i) \qquad \max \left\{ \|U - \widehat{U}(\widehat{U}^\top U)\|_{2 \to \infty}, \|V - \widehat{V}(\widehat{V}^\top V)\|_{2 \to \infty} \right\} \leq C \frac{\kappa \mu^2 r}{n \|M\|_\infty} \mathcal{B},$$

$$(ii) \qquad \|\widehat{M} - M\|_{2 \to \infty} \leq C \kappa \mathcal{B}, \quad \|\widehat{P} - P\|_{1 \to \infty} \leq C \frac{\kappa \sqrt{n}}{(\nu_0)_{\min}} \mathcal{B},$$

$$(iii) \qquad \|\widehat{M} - M\|_\infty \leq C \frac{\kappa \mu^2 r}{\sqrt{n}} \mathcal{B},$$

$$(iv) \qquad \|\widehat{P} - P\|_\infty \leq C \frac{\mathcal{B}}{(\nu_0)_{\min}} \left[ \sqrt{n} \kappa \frac{\|M\|_\infty}{(\nu_0)_{\min}} + \left(1 + \frac{\kappa \mathcal{B}}{\sqrt{n} \|M\|_\infty}\right) \frac{\kappa \mu^2 r}{\sqrt{n}} \right],$$

*where (iv) holds if in addition $T \geq cn\|M\|_\infty(\nu_0)_{\min}^{-2} r \mu^2 \kappa^4 \log(n\sqrt{T}/\delta)$*

Note that in theorem, the condition (a) on $n$ has been introduced just to simplify the expression of $\mathcal{B}$ (refer to App. B.2 for a full statement of the theorem without this condition).

**Corollary 4.** *(Homogeneous transition matrix) When $\kappa = \Theta(1)$, $\mu = \Theta(1)$, $r = \Theta(1)$, $M_{\max} = \Theta(M_{\min})$, we say that the frequency matrix $M$ is homogeneous. If $T \geq cn\log(nT)$ for some universal constant $c > 0$, then we have with probability at least $1 - \min\{n^{-2}, T^{-1}\}$:*

$$\max\left\{\|U - \widehat{U}(\widehat{U}^\top U)\|_{2\to\infty}, \|V - \widehat{V}(\widehat{V}^\top V)\|_{2\to\infty}\right\} \lesssim \sqrt{\frac{\log(nT)}{T}},$$

$$\|\widehat{M} - M\|_{2\to\infty} \lesssim \frac{1}{n}\sqrt{\frac{\log(nT)}{T}}, \ \|\widehat{M} - M\|_\infty \lesssim \frac{1}{n}\sqrt{\frac{\log(nT)}{nT}},$$

$$\|\widehat{P} - P\|_{1\to\infty} \lesssim \sqrt{\frac{n\log(nT)}{T}}, \ \|\widehat{P} - P\|_\infty \lesssim \sqrt{\frac{\log(nT)}{nT}}.$$

For a homogeneous frequency matrix, $\|U\|_{2\to\infty} = \Theta(1/\sqrt{n})$, $\|M\|_{2\to\infty} = \Theta(1/n\sqrt{n})$, $\|M\|_\infty = \Theta(1/n^2)$, $\|P\|_{1\to\infty} = 1$, $\|P\|_\infty = \Theta(1/n)$. Thus for all these metrics, our estimates achieve a relative error scaling at most as $\sqrt{n/T}$ up to the logarithmic factor.

### 3.3 Transition matrices under the forward model

For Model II(b), we first split the data into $\tau$ subsets of transitions: for $k = 1, \ldots, \tau$, the $k$-th subset is $((x_k, x_{k+1}), (x_{k+\tau}, x_{k+1+\tau}), \ldots, (x_{k+(T_\tau-1)\tau}, x_{k+1+(T_\tau-1)\tau}))$ where $T_\tau = \lfloor T/\tau \rfloor$. By separating two transitions in the same subset, we break the inherent correlations in the data if $\tau$ is large enough. Now we let $\widetilde{M}^{(k)}$ be the matrix recording the empirical frequencies of the transitions in the $k$-th subset: $\widetilde{M}^{(k)}_{i,j} = \frac{1}{T_\tau}\sum_{l=0}^{T_\tau-1}\mathbb{1}_{\{(x_{k+l\tau}, x_{k+1+l\tau})=(i,j)\}}$ for any pair of states $(i, j)$. Let $\widehat{M}^{(k)}$ be the best $r$-rank approximation of $\widetilde{M}^{(k)}$. As in (1), we define the corresponding $\widehat{P}^{(k)}$. Finally we may aggregate these estimates $\widehat{M} = \frac{1}{\tau}\sum_{k=1}^\tau \widehat{M}^{(k)}$ and $\widehat{P} = \frac{1}{\tau}\sum_{k=1}^\tau \widehat{P}^{(k)}$. We present below the performance analysis for the estimates coming from a single subset; the analysis of the aggregate estimates easily follows.

For any $\varepsilon > 0$, we define the $\varepsilon$-mixing time of the Markov chain with transition matrix $P$ as $\tau(\varepsilon) = \min\{t \geq 1 : \max_{1\leq i\leq n}\frac{1}{2}\|P^t_{i,:} - \nu^\top\|_1 \leq \varepsilon\}$, and its mixing time as $\tau^\star = \tau(1/4)$. The next theorem is a simplified version and a consequence of a more general and tighter theorem presented in App. B.3. To simplify the presentation, we define:

$$h(M, T, \delta) = n\tau^\star\log(\tfrac{n\sqrt{T}}{\delta})\log(T\nu_{\min}^{-1})\max\left\{\mu^6\kappa^6 r^3, \frac{\log^2(\tfrac{n\sqrt{T_\tau}}{\delta})\mathbb{1}_{\{\exists \ell: T_\tau\|M_{\ell,:}\|_\infty \leq 1\}}}{\log^2(1 + \tfrac{1}{T_\tau\|M\|_\infty})}\right\}.$$

**Theorem 5.** *Let $\delta > 0$. Assume that $\nu_{\min} = \min_{i\in[n]}\nu_i > 0$ and that $\tau/(\tau^\star\log(T\nu_{\min}^{-1})) \in [c_1, c_2]$ for some universal constants $c_2 > c_1 \geq 2$. Introduce:*

$$\mathcal{B} = \mu\kappa\sqrt{\frac{r\tau^\star\|M\|_\infty}{T}}\log\left(\frac{n\sqrt{T_\tau}}{\delta}\right)\log\left(\frac{T}{\nu_{\min}}\right).$$

*If (a) $n \geq c\tau^\star\log^{3/2}(nT^{3/2}/\delta)\log^{1/2}(T\nu_{\min}^{-1})$ and (b) $T \geq ch(M, T, \delta)$ for some universal constant $c > 0$, then there exists a universal constant $C > 0$ such that the following inequalities hold with probability at least $1 - \delta$:*

*(i)* $\quad \max\left\{\|U - \widehat{U}(\widehat{U}^\top U)\|_{2\to\infty}, \|V - \widehat{V}(\widehat{V}^\top V)\|_{2\to\infty}\right\} \leq C\dfrac{\kappa\mu^2 r}{n\|M\|_\infty}\mathcal{B},$

*(ii)* $\quad \|\widehat{M} - M\|_{2\to\infty} \leq C\kappa\mathcal{B}, \quad \|\widehat{P} - P\|_{1\to\infty} \leq C\dfrac{\kappa\sqrt{n}}{\nu_{\min}}\mathcal{B},$

*(iii)* $\quad \|\widehat{M} - M\|_\infty \leq C\dfrac{\kappa\mu^2 r}{\sqrt{n}}\mathcal{B},$

*(iv)* $\quad \|\widehat{P} - P\|_\infty \leq C\dfrac{\mathcal{B}}{\nu_{\min}}\left[\sqrt{n}\kappa\dfrac{\|M\|_\infty}{\nu_{\min}} + \left(1 + \dfrac{\kappa\mathcal{B}}{\sqrt{n}\|M\|_\infty}\right)\dfrac{\kappa\mu^2 r}{\sqrt{n}}\right],$

*where (iv) holds if in addition $T \geq cn\|M\|_\infty\nu_{\min}^{-2}\tau^\star r\mu^2\kappa^4\log(n\sqrt{T}/\delta)\log(T\nu_{\min}^{-1})$.*

Note that our guarantees hold when $\tau$ roughly scales as $\tau^\star\log(T\nu_{\min}^{-1})$. Hence to select $\tau$, one would need an idea of the latter quantity. It can be estimated typically using $\tau^\star\nu_{\min}^{-1}$ samples [61] (which

is small when compared to the constraint $T \geq ch(M, T, \delta)$ as soon as $\nu_{\min} = \Omega(1/n))$. Further observe that in the theorem, the condition (a) can be removed (refer to App. B.3 for a full statement of the theorem without this condition).

**Corollary 6.** *(Homogeneous transition matrices) Assume that $M$ is homogeneous (as defined in Corollary 4). Let $\tau = \log(Tn)$. If $T \geq cn \log^2(nT)$ for some universal constant $c > 0$, then we have with probability at least $1 - \min\{n^{-2}, T^{-1}\}$:*

$$\max\left\{\|U - \widehat{U}(\widehat{U}^\top U)\|_{2\to\infty}, \|V - \widehat{V}(\widehat{V}^\top V)\|_{2\to\infty}\right\} \lesssim \frac{1}{\sqrt{T}} \log(nT),$$

$$\|\widehat{M} - M\|_{2\to\infty} \lesssim \frac{1}{n\sqrt{T}} \log(nT), \quad \|\widehat{M} - M\|_\infty \lesssim \frac{1}{n\sqrt{nT}} \log(nT),$$

$$\|\widehat{P} - P\|_{1\to\infty} \lesssim \sqrt{\frac{n}{T}} \log(nT), \quad \|\widehat{P} - P\|_\infty \lesssim \frac{1}{\sqrt{nT}} \log(nT).$$

As for the generative model, for a homogeneous frequency matrix, our estimates achieve a relative error scaling at most as $\sqrt{n/T}$ up to the logarithmic factor for all metrics. Note that up to a logarithmic factor, the upper bound for $\|\widehat{P} - P\|_{1\to\infty}$ (and similarly for $\widehat{M}$) matches the minimax lower bound derived in [63].

### 3.4 Elements of the proofs

The proofs of the three above theorems share similar arguments. We only describe elements of the proof of Theorem 5, corresponding to the most challenging model. The most difficult result concerns the singular subspace recovery (the upper bounds (i) in our theorems), and it can be decomposed into the following three steps. The first two steps are meant to deal with the Markovian nature of the data. The third step consists in applying a leave-one-out analysis to recover the singular subspaces.

*Step 1: Multinomial approximation of Markovian data.* We treat the matrix $\widetilde{M}^{(k)}$ arising from one subset of data, and for simplicity, we remove the superscript $(k)$, i.e., $\widetilde{M} = \widetilde{M}^{(k)}$. Note that $T_\tau \widetilde{M}$ is a matrix recording the numbers of transitions observed in the data for any pair of states: denote by $N_{i,j}$ this number for $(i, j)$. We approximate the joint distribution of $N = (N_{i,j})_{(i,j)}$ by a multinomial distribution with $n^2$ components and parameter $T_\tau M_{i,j}$ for component $(i, j)$. Denote by $Z = (Z_{i,j})_{(i,j)}$ the corresponding multinomial random variable. Using the mixing property of the Markov chain and the choice of $\tau$, we establish (see Lemma 21 in App. C) that for any subset $\mathcal{Z}$ of $\{z \in \mathbb{N}^{n^2} : \sum_{(i,j)} z_{i,j} = T_\tau\}$, we have $\mathbb{P}[N \in \mathcal{Z}] \leq 3\mathbb{P}[Z \in \mathcal{Z}]$.

*Step 2: Towards Poisson random matrices with independent entries.* The random matrix $Z$ does not have independent entries. Independence is however a requirement if we wish to apply the leave-one-out argument. Consider the random matrix $Y$ whose entries are independent Poisson random variables with mean $T_\tau M_{i,j}$ for the $(i, j)$-th entry. We establish the following connection between the distribution of $Z$ and that of $Y$: for any $\mathcal{Z} \subset \mathbb{N}^{n^2}$, we have $\mathbb{P}[Z \in \mathcal{Z}] \leq e\sqrt{T_\tau}\mathbb{P}[Y \in \mathcal{Z}]$. Refer to Lemma 22 in App. C for details.

*Step 3: The leave-one-out argument for Poisson matrices.* Combining the two first steps provides a connection between the observation matrix $\widetilde{M}$ and a Poisson matrix $Y$ with independent entries. This allows us to apply a leave-one-out analysis to $\widetilde{M}$ as if it had independent entries (replacing $\widetilde{M}$ by $Y$). The analysis starts by applying the standard dilation trick (see Section 4.10 in [14]) so as to make $\widetilde{M}$ symmetric. Then, we can decompose the error $\|U - \widehat{U}(\widehat{U}^\top U)\|_{2\to\infty}$ (see Lemma 32 in App. E) into several terms. The most challenging of these terms is $\|(M - \widetilde{M})(U - \widehat{U}(\widehat{U}^\top U))\|_{2\to\infty} = \max_{l \in [n]} \|(M_{l,:} - \widetilde{M}_{l,:})(U - \widehat{U}(\widehat{U}^\top U))\|_2$ because of inherent dependence between $M - \widetilde{M}$ and $U - \widehat{U}(\widehat{U}^\top U)$. The leave-one-out analysis allows us to decouple this statistical dependency. It consists in exploiting the row and column independence of matrix $\widetilde{M}$ to approximate $\|(M_{l,:} - \widetilde{M}_{l,:})(U - \widehat{U}(\widehat{U}^\top U))\|_2$ by $\|(M_{l,:} - \widetilde{M}_{l,:})(U - \widehat{U}^{(l)}((\widehat{U}^{(l)})^\top U))\|_2$ where $\widehat{U}^{(l)}$ is the matrix of eigenvectors of matrix $\widetilde{M}^{(l)}$ obtained by zeroing the $l$-th row and column of $\widetilde{M}$. By construction, $(M_{l,:} - \widetilde{M}_{l,:})$ and $U - \widehat{U}^{(l)}((\widehat{U}^{(l)})^\top U)$ are independent, which simplifies the analysis. The proof is completed by a

further appropriate decomposition of this term, combined with concentration inequalities for random Poisson matrices (see App. D).

# 4 Regret Minimization in Low-Rank Bandits

Consider a low-rank bandit problem with a homogeneous rank-$r$ reward matrix $M$. We wish to devise an algorithm $\pi$ with low regret. $\pi$ selects in round $t$ an entry $(i_t^\pi, j_t^\pi)$ based on previous observations, and receives as a feedback the noisy reward $M_{i_t^\pi, j_t^\pi} + \xi_t$. The regret up to round $T$ is defined by $R^\pi(T) = TM_{i^\star, j^\star} - \mathbb{E}[\sum_{t=1}^T M_{i_t^\pi, j_t^\pi}]$, where $(i^\star, j^\star)$ is an optimal entry. One could think of a simple Explore-Then-Commit (ETC) algorithm, where in the first phase entries are sampled uniformly at random, and where in a second phase, the algorithm always selects the highest entry of $\widehat{M}$ built using the samples gathered in the first phase and obtained by spectral decomposition. When the length of the first phase is $T^{2/3}(n+m)^{1/3}$, the ETC algorithm would yield a regret upper bounded by $O(T^{2/3}(n+m)^{1/3})$ for $T = \Omega((n+m)\log^3(n+m))$.

To get better regret guarantees, we present SME-AE (Successive Matrix Estimation and Arm Elimination), an algorithm meant to identify the best entry as quickly as possible with a prescribed level of certainty. After the SME-AE has returned the estimated best entry, we commit and play this entry for the remaining rounds. The pseudo-code of SME-AE is presented in Algorithm 1. The algorithm runs in epochs: in epoch $\ell$, it samples $T_\ell$ entries uniformly at random among all entries (in $T_\ell$, the constant $C$ just depends on upper bounds of the parameters $\mu$, $\kappa$, and $\|M\|_\infty$, refer to App. G); from these samples, a matrix $\widehat{M}^{(\ell)}$ is estimated and $\mathcal{A}_\ell$, the set of candidate arms, is pruned. The pruning procedure is based on the estimated gaps: $\widehat{\Delta}_{i,j}^{(\ell)} = \widehat{M}_\star^{(\ell)} - \widehat{M}_{i,j}^{(\ell)}$ where $\widehat{M}_\star^{(\ell)} = \max_{i,j} \widehat{M}_{i,j}^{(\ell)}$.

---

**Algorithm 1:** **S**uccesive **M**atrix **E**stimation and **A**rm **E**limination (**SME-AE**)

**Input:** Arms $[m] \times [n]$, confidence level $\delta$
$\ell = 1$ ;
$\mathcal{A}_1 = [m] \times [n]$;
**while** $|\mathcal{A}_\ell| > 1$ **do**

    $\delta_\ell = \delta/\ell^2$;

    $T_\ell = \left\lceil C \left(2^{\ell+2}\right)^2 (m+n) \log^3\left(2^{2\ell+4}(m+n)/\delta_\ell\right) \right\rceil$ ;

    Sample uniformly at random $T_\ell$ entries from $\mathcal{A}_1$: $(M_{i_t, j_t} + \xi_t)_{t=1,\ldots,T_\ell}$ ;

    Estimate $\widehat{M}^{(\ell)}$ via spectral decomposition as described in Section 3.1 ;

    $\mathcal{A}_{\ell+1} = \left\{(i,j) \in \mathcal{A}_\ell : \widehat{\Delta}_{i,j}^{(\ell)} \leq 2^{-(\ell+2)}\right\}; \ell = \ell + 1$;

**end**
**Output:** Recommend the remaining pair $(\hat{\imath}_\tau, \hat{\jmath}_\tau)$ in $\mathcal{A}_\ell$.

---

The following theorem characterizes the performance of SME-AE and the resulting regret. To simplify the notation, we introduce the gaps: for any entry $(i,j)$, $\Delta_{i,j} = (M_{i^\star, j^\star} - M_{i,j})$, $\Delta_{\min} = \min_{(i,j):\Delta_{i,j}>0} \Delta_{i,j}$, $\Delta_{\max} = \max_{(i,j)} \Delta_{i,j}$, and $\bar{\Delta} = \sum_{(i,j)} \Delta_{i,j}/(mn)$. We define the function $\psi(n,m,\delta) = \frac{c(m+n)\log(e/\Delta_{\min})}{\Delta_{\min}^2} \log^3\left(\frac{e(m+n)\log(e/\Delta_{\min})}{\Delta_{\min}\delta}\right)$ for some universal constant $c > 0$.

**Theorem 7.** *(Best entry identification) For any $\delta \in (0,1)$, SME-AE($\delta$) stops at time $\tau$ and recommends arm $(\hat{\imath}_\tau, \hat{\jmath}_\tau)$ with the guarantee $\mathbb{P}\left((\hat{\imath}_\tau, \hat{\jmath}_\tau) = (i^\star, j^\star), \tau \leq \psi(n,m,\delta)\right) \geq 1 - \delta$.*
*Moreover, for any $T \geq 1$ and $\alpha > 0$, the sample complexity $\tau$ of SME-AE($1/T^\alpha$) satisfies $\mathbb{E}[\tau \wedge T] \leq \psi(n,m,T^{-\alpha}) + T^{1-\alpha}$.*
*(Regret) Let $T \geq 1$. Consider the algorithm $\pi$ that first runs SME-AE($1/T^2$) and then commits to its output $(\hat{\imath}_\tau, \hat{\jmath}_\tau)$ after $\tau$. We have: $R^\pi(T) \leq \bar{\Delta}\left(\psi(n,m,T^{-2}) + 1\right) + \frac{\Delta_{\max}}{T}$.*

The proof of Theorem 7 is given in App. G. Note that the regret upper bounds hold for any time horizon $T \geq 1$, and that it scales as $O((m+n)\log^3(T)\bar{\Delta}/\Delta_{\min}^2)$ (up to logarithmic factors in $m, n$ and $1/\Delta_{\min}$). The cubic dependence in $\log^3(T)$ is an artifact of our proof techniques. More precisely, it is due to the Poisson approximation used to obtain entry-wise guarantees. Importantly, for any time horizon, the regret upper bound only depends on $(m+n)$ rather than $mn$ (the number

of arms / entries), and hence, the low-rank structure is efficiently exploited. If we further restrict our attention to problems with gap ratio $\Delta_{\max}/\Delta_{\min}$ upper bounded by $\zeta$, our regret upper bound becomes $O(\zeta(m+n)\log^3(T)/\Delta_{\min})$, and can be transformed into the minimax gap-independent upper bound $O(\zeta((m+n)T)^{1/2}\log^2(T))$, see App. G. Finally note that $\Omega(((m+n)T)^{1/2})$ is an obvious minimax regret lower bound for our low-rank bandit problem.

A very similar low-rank bandit problem has been investigated in [7]. There, under similar assumptions (see Assumption 1 and Definition 1), the authors devise an algorithm with both gap-dependent and gap-independent regret guarantees. The latter are difficult to compare with ours. Their guarantees exhibit a better dependence in $T$ and $\Delta_{\min}$, but worse in the matrix dimensions $n$ and $m$. Indeed in our model, $b^\star$ in [7] corresponds to $\|M\|_{2\to\infty}$ and scales as $\sqrt{n}$. As a consequence, the upper bounds in [7] have a dependence in $n$ and $m$ scaling as $\sqrt{n}(n+m)$ in the worst case for gap-dependent guarantees and even $nm$ (through the constant $C_2$ in [7]) for gap-independent guarantees.

# 5    Representation Learning in Low-Rank MDPs

The results derived for Models II(a) and II(b) are instrumental towards representation learning and hence towards model-based or reward-free RL in low-rank MDPs. In this section, we provide an example of application of these results, and mention other examples in Section 7. A low-rank MDP is defined by $(\mathcal{S}, \mathcal{A}, \{P^a\}_{a\in\mathcal{A}}, R, \gamma)$ where $\mathcal{S}, \mathcal{A}$ denote state and action spaces of cardinalities $n$ and $A$, respectively, $P^a$ denotes the rank-$r$ transition matrix when taking action $a$, $R$ is the reward function, and $\gamma$ is the discount factor. We assume that all rewards are in $[0, 1]$. The value function of a policy $\pi : \mathcal{S} \to \mathcal{A}$ is defined as $V_R^\pi(x) = \mathbb{E}[\sum_{t=1}^\infty \gamma^{t-1} R(x_t^\pi, \pi_t(x_t^\pi))|x_1^\pi = x]$ where $x_t^\pi$ is the state visited under $\pi$ in round $t$. We denote by $\pi^\star(R)$ an optimal policy (i.e., with the highest value function).

**Reward-free RL.** In the reward-free RL setting (see e.g. [36, 30, 66]), the learner does not receive any reward signal during the exploration process. The latter is only used to construct estimates $\{\widehat{P}^a\}_{a\in\mathcal{A}}$ of $\{P^a\}_{a\in\mathcal{A}}$. The reward function $R$ is revealed at the end, and the learner may compute $\hat{\pi}(R)$ an optimal policy for the MDP $(\mathcal{S}, \mathcal{A}, \{\widehat{P}^a\}_{a\in\mathcal{A}}, R, \gamma)$. The performance of this model-based approach is often assessed through $\Gamma = \sup_R \|V_R^{\pi^\star(R)} - V_R^{\hat{\pi}(R)}\|_\infty$. In tabular MDP, to identify an $\epsilon$-optimal policy for all reward functions, i.e., to ensure that $\Gamma \le \epsilon$, we believe that the number of samples that have to be collected should be $\Omega(\text{poly}(\frac{1}{1-\gamma})\frac{n^2 A}{\epsilon^2})$ (the exact degree of the polynomial in $1/(1-\gamma)$ has to be determined). This conjecture is based on the sample complexity lower bounds derived for reward-free RL in episodic tabular MDP [30, 43]. Now for low-rank MDPs, the equivalent lower bound would be $\Omega(\text{poly}(\frac{1}{1-\gamma})\frac{nA}{\epsilon^2})$ [28] (this minimax lower bound is valid for Block MDPs, a particular case of low-rank MDPs).

Leveraging our low-rank matrix estimation guarantees, we propose an algorithm matching the aforementioned sample complexity lower bound (up to logarithmic factors) at least when the frequency matrices $\{M^a\}_{a\in\mathcal{A}}$ are homogeneous. The algorithm consists of two phases: (1) in the model estimation phase, it collects $A$ trajectories, each of length $T/A$, corresponding to the Markov chains with transition matrices $\{P^a\}_{a\in\mathcal{A}}$. From this data, it uses the spectral decomposition method described in §3 to build estimates $\{\widehat{P}^a\}_{a\in\mathcal{A}}$. (2) In the planning phase, based on the reward function $R$, it computes the best policy $\hat{\pi}(R)$ for the MDP $(\mathcal{S}, \mathcal{A}, \{\widehat{P}^a\}_{a\in\mathcal{A}}, R, \gamma)$. The following theorem summarizes the performance of this algorithm. To simplify the presentation, we only provide the performance guarantees of the algorithm for homogeneous transition matrices (guarantees for more general matrices can be derived plugging in the results from Theorem 5).

**Theorem 8.** *Assume that for any $a \in \mathcal{A}$, $M^a$ is homogeneous (as defined in Corollary 4). If $T \ge cnA\log^2(nAT)$ for some universal constant $c > 0$, then we have with probability at least $1 - \min\{n^{-2}, T^{-1}\}$: $\Gamma = \sup_R \|V_R^{\pi^\star(R)} - V_R^{\hat{\pi}(R)}\|_\infty \lesssim \frac{1}{(1-\gamma)^2}\sqrt{\frac{nA}{T}}\log(nAT)$.*

Theorem 8 is a direct consequence of Corollary 6 and of the fact that for any reward function $R$: $\|V_R^{\pi^\star(R)} - V_R^{\hat{\pi}(R)}\|_\infty \le \frac{2\gamma}{(1-\gamma)^2}\max_{a\in\mathcal{A}}\|P^a - \widehat{P}^a\|_{1\to\infty}$, see App. A. The theorem implies that if we wish to guarantee $\Gamma \le \epsilon$, we just need to collect $O(\frac{nA}{\epsilon^2(1-\gamma)^4})$ samples up to a logarithmic factor.

This sample complexity is minimax optimal in $n$, $A$, and $\epsilon$ in view of the lower bound presented in [28].

# 6 Related Work

**Low-rank matrix estimation.** Until recently, the main efforts on low-rank matrix recovery were focused on guarantees w.r.t. the spectral or Frobenius norms, see e.g. [19] and references therein. The first matrix estimation and subspace recovery guarantees in $\ell_{2\to\infty}$ and $\ell_\infty$ were established in [22], [21] via a more involved perturbation analysis than the classical Davis-Kahan bound. An alternative approach based on a leave-one-out analysis was proposed in [2], and further refined in [10, 12, 17], see [14] for a survey. Some work have also adapted the techniques beyond the independent noise assumption [39, 1, 5], but for very specific structural dependence. We deal with a stronger dependence, and in particular with Markovian data (an important scenario in RL).

The estimation of low-rank transition matrices of Markov chains has been studied in [63, 9] using spectral methods and in [40, 67] using maximum-likelihood approaches. [63] does not conduct any fine-grained subspace recovery analysis (such as the leave-one-out), and hence the results pertaining to the $\|\cdot\|_{1\to\infty}$-guarantees are questionable; refer to App. H for a detailed justification. All these papers do not present entry-wise guarantees.

It is worth mentioning that there exist other methods for matrix estimation that do not rely on spectral decompositions like ours, yet enjoy entry-wise matrix estimation guarantees [51, 3, 50]. However, these methods require different assumptions than ours that may be too strong for our purposes, notably having access to the so-called anchor rows and columns. Moreover, we do not know if these methods also lead to guarantees for subspace recovery in the norm $\|\cdot\|_{2\to\infty}$, nor how to extend those results to settings with dependent noise.

**Low-rank bandits.** Low-rank structure in bandits has received a lot of attention recently [35, 37, 32, 57, 41, 7, 34, 27]. Different set-ups have been proposed (refer to App. H for a detailed exposition, in particular, we discuss how the settings proposed in [32, 7] are equivalent), and regret guarantees in an instance dependent and minimax sense have been both established.

Typically minimax regret guarantees in bandits scale as $\sqrt{T}$, but the scaling in dimension may defer when dealing with a low rank structure [32, 34, 7]. In [32], the authors also leverage spectral methods. They reduce the problem to a linear bandit of dimension $nm$ but where only roughly $n + m$ dimensions are relevant. This entails that a regret lower bound of order $(n + m)\sqrt{T}$ is inevitable. Actually, in their reduction to linear bandits, they only use a subspace recovery in Frobenius norm, which perhaps explains the scaling $(n + m)^{3/2}$ in their regret guarantees. It is worth noting that in [34], the authors manage to improve upon the work [32] and obtain a scaling order $(m + n)$ in the regret. Our algorithm leverages entry-wise guarantees which rely on a stronger subspace recovery guarantee. This allows us to obtain a scaling $\sqrt{n + m}$ in the regret. The work of [27] is yet another closely related work to ours. There, the authors propose an algorithm achieving a regret of order $\mathrm{polylog}(n + m)\sqrt{T}$ for a contextual bandit problem with low rank structure. However, their result only holds for rank 1 and their observation setup is different than ours because in their setting, the learner observes $m$ entries per round while in ours the learner only observes one entry per round. In [7], the authors use matrix estimation with nuclear norm penalization to estimate the matrix $M$. Their regret guarantees are already discussed in §4.

Some instance-dependent guarantees with logarithmic regret for low rank bandits have been established in [35, 37, 57]. However, these results suffer what may be qualified as serious limitations. Indeed, [35, 57] provide instance dependent regret guarantees but only consider low-rank bandits with rank 1, and the regret bounds of [35] are expressed in terms of the so-called column and row gaps (see their Theorem 1) which are distinct from the standard gap notions. [37] extend the results in [35] to rank $r$ with the limitation that they require stronger assumptions than ours. Moreover, the computational complexity of their algorithm depends exponentially on the rank $r$; they require a search over spaces of size $\binom{m}{r}$ and $\binom{n}{r}$. Our proposed algorithm does not suffer from such limitations.

We wish to highlight that our entry-wise guarantees for matrix estimation are the key enabling tool that led us to the design and analysis of our proposed algorithm. In fact, the need for such guarantees arises naturally in the analysis of gap-dependent regret bounds (see Appendix G.1). Therefore, we

believe that such guarantees can pave the way towards better, faster, and efficient algorithms for bandits with low-rank structure.

**Low-rank Reinforcement Learning.** RL with low rank structure has been recently extensively studied but always in the function approximation framework [29, 18, 20, 44, 24, 65, 56, 4, 46, 59, 60, 49]. There, the transition probabilities can be written as $\phi(x,a)^{\top}\mu(x')$ where the unknown feature functions $\phi(x,a), \mu(x') \in \mathbb{R}^r$ belong to some specific class $\mathcal{F}$ of functions. The major issue with algorithms proposed in this literature is that they rely on strong computational oracles (e.g., ERM, MLE), see [33, 25, 64] for detailed discussions. In contrast, we do not assume that the transition matrices are constructed based on a given restricted class of functions, and our algorithms do not rely on any oracle and are computationally efficient. In [51, 50], the authors also depart from the function approximation framework. There, they consider a low rank structure different than ours. Their matrix estimation method enjoys an entry-wise guarantee, but requires to identify a subset of rows and columns spanning the range of the full matrix. Moreover, their results are only limited the generative models, which allows to actually rely on independent data samples.

## 7 Conclusion and Perspectives

In this paper, we have established that spectral methods efficiently recover low-rank matrices even in correlated noise. We have investigated noise correlations that naturally arise in RL, and have managed to prove that spectral methods yield nearly-minimal entry-wise error. Our results for low-rank matrix estimation have been applied to design efficient algorithms in low-rank RL problems and to analyze their performance. We believe that these results may find many more applications in low-rank RL. They can be applied (i) to reward-free RL in episodic MDPs (this setting is easier than that presented in §5 since successive episodes are independent); (ii) to scenarios corresponding to offline RL [62] where the data consists of a single trajectory generated under a given behavior policy (from this data, we can extract the transitions $(x,a,x')$ where a given action $a$ is involved and apply the spectral method to learn $\widehat{P}^a$); (iii) to traditional RL where the reward function $R$ has to be learnt (learning $R$ is a problem that lies in some sense between the inference problems in our Models I and II); (iv) to model-free RL where we would directly learn the $Q$ function as done in [52] under a generative model; (v) to low-rank RL problems with continuous state spaces (this can be done if the transition probabilities are smooth in the states, and by combining our methods to an appropriate discretization of the state space).

## Acknowledgment

This research was supported by the Wallenberg AI, Autonomous Systems and Software Program (WASP) funded by the Knut and Alice Wallenberg Foundation.

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
