# Contents

# A  Preliminaries

In this section, we present a few results that are used throughout our analysis.

## A.1  Matrix norms

**Lemma 9.** *Let $A \in \mathbb{R}^{n \times m}, B \in \mathbb{R}^{m \times r}$. Then:*

$$\|AB\|_{2 \to \infty} \leq \|A\|_{1 \to \infty} \|B\|_{2 \to \infty}, \tag{2}$$

$$\|AB\|_{2 \to \infty} \leq \|A\|_{2 \to \infty} \|B\|, \tag{3}$$

$$\|AB\|_{\infty} \leq \|A\|_{2 \to \infty} \|B^{\top}\|_{2 \to \infty}. \tag{4}$$

*Proof.* The proof of the lemma directly follows from Hölder's inequality (see for example Proposition 6.5 in [12]). □

## A.2  Mixing time

**Lemma 10.** *(Lemma 5 in [63]) Let $\tau(\varepsilon)$ be the $\varepsilon$-mixing time of an irreducible Markov chain. Then if $\varepsilon \leq \delta < 1/2$,*

$$\tau(\varepsilon) \leq \tau(\delta) \left( 1 + \left\lceil \frac{\log(\delta/\varepsilon)}{\log(1/(2\delta))} \right\rceil \right).$$

## A.3  Value difference lemmas

The following lemmas are used in Section 5 to prove Theorem 8. Recall the definition of value function of a policy $\pi$: $V_R^\pi(x) = \mathbb{E}[\sum_{t=1}^\infty \gamma^{t-1} R(x_t^\pi, \pi_t(x_t^\pi)) | x_1^\pi = x]$. The (state, action) value function of $\pi$ is also defined as: for any state $x \in \mathcal{S}$ and action $a \in \mathcal{A}$,

$$Q_R^\pi(x, a) = R(x, a) + \gamma \mathbb{E}_{x' \sim P(\cdot | x, a)}[V_R^\pi(x')].$$

We denote by $\widehat{Q}_R^\pi$ the (state, action) value function of $\pi$ in the MDP where $P$ is replaced by its estimate $\widehat{P}$, and let $\widehat{\pi}(R)$ be the optimal policy for this MDP.

**Lemma 11.** *We have that*

$$\|V_R^{\pi^\star(R)} - V_R^{\widehat{\pi}(R)}\|_\infty \leq 2 \sup_\pi \|Q_R^\pi - \widehat{Q}_R^\pi\|_\infty$$

*Proof.* We remove the subscript $R$ to simplify the notation. For any $s$, we have

$$
\begin{aligned}
V^{\pi^\star}(s) - V^{\widehat{\pi}}(s) = Q^{\pi^\star}(s, \pi^\star(s)) - Q^{\widehat{\pi}}(s, \widehat{\pi}(s)) &= [Q^{\pi^\star}(s, \pi^\star(s)) - \widehat{Q}^{\pi^\star}(s, \pi^\star(s))] \\
&+ [\widehat{Q}^{\widehat{\pi}}(s, \widehat{\pi}(s)) - Q^{\widehat{\pi}}(s, \widehat{\pi}(s))] + [\widehat{Q}^{\pi^\star}(s, \pi^\star(s)) - \widehat{Q}^{\widehat{\pi}}(s, \widehat{\pi}(s))] \\
&\leq 2 \sup_\pi \|Q^\pi - \widehat{Q}^\pi\|_\infty,
\end{aligned}
$$

since $\widehat{Q}^{\pi^\star}(s, \pi^\star(s)) \leq \widehat{Q}^{\widehat{\pi}}(s, \widehat{\pi}(s))$ by definition of $\widehat{\pi}$. □

**Lemma 12.** *(Proposition 2.1 in [3]) For all policies $\pi$:*

$$\|Q_R^\pi - \widehat{Q}_R^\pi\|_\infty \leq \frac{\gamma}{(1-\gamma)^2} \max_{a \in \mathcal{A}} \|P^a - \widehat{P}^a\|_{1 \to \infty}$$

Combining the two lemmas, we get:

$$\|V_R^{\pi^\star(R)} - V_R^{\widehat{\pi}(R)}\|_\infty \leq \frac{2\gamma}{(1-\gamma)^2} \max_{a \in \mathcal{A}} \|P^a - \widehat{P}^a\|_{1 \to \infty}.$$

This inequality is used in the proof of Theorem 8.

# B  Statement and proofs of the main results

In this appendix, we present the proofs of the main theorems. In Subsection §B.1, we provide the proof of Theorem 1 and Corollary 2. In Subsection §B.2, we give a complete, non-simplified version of Theorem 3 from which one can deduce Theorem 3 and Corollary 4 given in the main text. Finally, in Subsection §B.3, we present a complete, non-simplified version of Theorem 5 and from the latter, deduce Theorem 5 and Corollary 6.

## B.1  Reward matrix estimation – Model I

In this subsection, we present the proofs of Theorem 1. The proof of Corollary 2 is in fact immediate from Theorem 1.

*Proof of Theorem 1.* **Proof of (i).** Recall the results from Lemma 30: for all $\delta \in (0,1)$, if

$$\mathcal{B} = \sqrt{\frac{nm}{T}} \left( \sqrt{(n+m) \log \left( \frac{e(n+m)T}{\delta} \right)} + \log^{3/2} \left( \frac{e(n+m)T}{\delta} \right) \right), \tag{5}$$

then for all $T \geq c_1(\mu^4 \kappa^2 r^2 + 1)(m+n) \log^3 \left( e^2(m+n)T/\delta \right)$, the event

$$\max(\|U - \widehat{U}(\widehat{U}^\top U)\|, \|V - \widehat{V}(\widehat{V}^\top V)\|) \leq C_1 \frac{\|M\|\|M\|_\infty}{\sigma_r(M)^2} \max(\|V\|_{2\to\infty} \|U\|_{2\to\infty})\mathcal{B}$$

holds with probability at least $1 - \delta$ for some universal constants $c_1, C_1 > 0$. To obtain the form presented in Theorem 1, we simply recall the definitions $\kappa = \|M\|/\sigma_r(M)$, $\mu = \max(\sqrt{m/r}\|U\|_{2\to\infty}, \sqrt{n/r}\|V\|_{2\to\infty})$ and the bound $\|M\|_\infty/\sigma_r(M) \leq (\mu^2\kappa r)/\sqrt{mn}$ from Lemma 17. We then substitute in the upper bound above. Note that $\mu, \kappa$ and $r$ are larger than 1 by definition.

**Proof of (ii).** To establish the desired bound, we use the decomposition error established in Lemma 34. Namely, under the event that $\|\widetilde{M} - M\| \leq c_1\sigma_r(M)$ for some universal constant $c_1 > 0$ sufficiently small, there exists a universal constant $c_2 > 0$ such that

$$\|\widehat{M} - M\|_{2\to\infty} \leq c_2\sigma_1(M) \left[ \|U - \widehat{U}(\widehat{U}^\top U)\|_{2\to\infty} + \mu\sqrt{\frac{r}{m}} \frac{\|\widetilde{M} - M\|}{\sigma_r(M)} \right]. \tag{6}$$

Hence, we only need high probability bounds on $\|U - \widehat{U}(\widehat{U}^\top U)\|_{2\to\infty}$ which we established in (i), and on $\|\widetilde{M} - M\|$ which we also established in Proposition 26 under the compound Poisson entries model described (15). We can extend the latter result under our observation model using the Poisson approximation Lemma 20, and finally write that for all $\delta \in (0,1)$, using the same definition of $\mathcal{B}$ as above in (5), for all for all $T \geq c_3 \log^3 ((n+m)/\delta)$, the following statement

$$\frac{\|\widetilde{M} - M\|}{\sigma_r(M)} \leq \frac{C_3\|M\|_\infty}{\sigma_r(M)}\mathcal{B} \tag{7}$$

holds with probability at least $1 - \delta$, for some universal constants $c_3, C_3 > 0$ large enough. Note that under the condition $T \geq c_4\mu^4\kappa^2r^2 \log^3 (e(n+m)/\delta)$ for some universal constant $c_4$ large enough, the high probability statement in (7) holds and in addition we also have $\|\widetilde{M} - M\| \leq c_1\sigma_r(M)$. There, we used the result of Lemma 17. The statement (ii) in Theorem 1 is obtained by first substituting in (6), the upper bound we get in (i) and that we get in (7), and then, using $\sigma_1(M) \leq \sqrt{mn}\|M\|_\infty$ and the bound $\|M\|_\infty/\sigma_r(M) \leq (\mu^2\kappa r)/\sqrt{mn}$ from Lemma 17.

**Proof of (iii).** To establish the desired bound, we use the decomposition error established in Lemma 36. Namely, under the event that $\|\widetilde{M} - M\| \leq c_1\sigma_r(M)$ for some universal constant $c_1 > 0$ sufficiently small, there exists a universal constant $c_2 > 0$ such that

$$\|\widehat{M} - M\|_\infty \leq c_2\|M\|_{2\to\infty} \left( \frac{\|M - \widetilde{M}\|}{\sigma_r(M)}\|V\|_{2\to\infty} + \|V - \widehat{V}W_{\widehat{V}}\|_{2\to\infty} \right)$$

$$+ c_2\|M - \widehat{M}\|_{2\to\infty}(\|V\|_{2\to\infty} + \|V - \widehat{V}W_{\widehat{V}}\|_{2\to\infty}). \tag{8}$$

To upper bound the above error, we need to control: (a) $\|V - \widehat{V}W_{\widehat{V}}\|_{2\to\infty}$, which we have already done in (i); (b) $\|M - \widetilde{M}\|$, which follows from Lemma 20 as established in the proof of (ii) (see the high probability statement (7)); and (c) $\|M - \widehat{M}\|_{2\to\infty}$, which again we have already done in (ii). The statement (iii) in Theorem 1 follows from first substituting in (8), the upper bounds we get from (a), (b) and (c), and then using $\|M\|_\infty/\sigma_r(M) \le (\mu^2\kappa r)/\sqrt{mn}$, $\mu = \max(\sqrt{m/r}\|U\|_{2\to\infty}, \sqrt{n/r}\|V\|_{2\to\infty})$, and the basic inequality $\|M\|_{2\to\infty} \le \sqrt{m}\|M\|_\infty \le \sqrt{m+n}\|M\|_\infty$. $\qquad\square$

## B.2 Transition matrix estimation under the generative model – Model II(a)

In this subsection, we present a complete, non-simplified version of Theorem 3, from which one can deduce Theorem 3 and Corollary 4 given in the main text.

First, let us define the function $g_\delta : \mathbb{R}^{n\times n} \to \mathbb{R}_+$ as

$$g_\delta(M) = \mathbb{1}_{\{\exists\ell:\|M_{\ell,:}\|_\infty\le 1\}} \log\left(\frac{ne}{\delta}\right) \log^{-1}\left(1 + \frac{1}{\|M\|_\infty}\right)$$
$$+ \mathbb{1}_{\{\forall\ell:\|M_{\ell,:}\|_\infty>1\}} \log\left(\frac{\|M\|_\infty ne}{\delta}\right) \sqrt{\|M\|_\infty}. \qquad (9)$$

We also use the following notation:

$$\begin{cases} \mathcal{A} = \frac{1}{\sqrt{T}}\sqrt{\|M\|_{1\to\infty} + \|M^\top\|_{1\to\infty}}, \\ \mathcal{B}' = \mu\kappa\sqrt{\frac{r}{n}}\left(\mathcal{A} + \frac{1}{T}g_{\delta/\sqrt{T}}(TM)\log\left(\frac{n\sqrt{T}}{\delta}\right)\right) + \sqrt{\frac{r\|M\|_\infty}{T}\log\left(\frac{n\sqrt{T}}{\delta}\right)}. \end{cases}$$

We first recall a standard result quantifying how well $\widetilde{M}$ approximates $M$.

**Lemma 13.** $\forall\delta \in (0,1)$, w.p. at least $1 - \delta$, $\|\widetilde{M} - M\| \le C\mathcal{A} + \frac{C}{T}g_{\delta/\sqrt{T}}(TM)\sqrt{\log(\frac{n\sqrt{T}}{\delta})}$.

*Proof.* The lemma follows directly from Lemma 22 (replacing $T_\tau$ by $T$) and Lemma 28. $\qquad\square$

**Theorem 14.** *Assume that* $(\nu_0)_{\min} = \min_{i\in[n]}(\nu_0)_i > 0$. *For any* $\delta > 0$, *if* $\|\widetilde{M} - M\| \le c\sigma_r(M)$, $g_{\delta/\sqrt{T}}(TM)\log(n\sqrt{T}/\delta) \le cT\sigma_r(M)$ *and* $\|M\|_\infty\log(n\sqrt{T}/\delta) \le cT\sigma_r^2(M)$ *for some universal constant* $c > 0$, *then there exists a universal constant* $C > 0$ *such that with probability at least* $1 - \delta$ *holds:*

$$(i) \quad \max\left\{\|U - \widehat{U}(\widehat{U}^\top U)\|_{2\to\infty}, \|V - \widehat{V}(\widehat{V}^\top V)\|_{2\to\infty}\right\} \le C\frac{\mathcal{B}'}{\sigma_r(M)},$$

$$(ii) \quad \|\widehat{M} - M\|_{2\to\infty} \le C\kappa\mathcal{B}', \quad \|\widehat{P} - P\|_{1\to\infty} \le C\frac{\kappa\sqrt{n}}{(\nu_0)_{\min}}\mathcal{B}',$$

$$(iii) \quad \|\widehat{M} - M\|_\infty \le C\left(\frac{\|M\|_{2\to\infty} + \kappa\mathcal{B}'}{\sigma_r(M)} + \kappa\mu\sqrt{\frac{r}{n}}\right)\mathcal{B}',$$

$$(iv) \quad \|\widehat{P} - P\|_\infty \le C\frac{\mathcal{B}'}{(\nu_0)_{\min}}\left[\sqrt{n}\kappa\frac{\|M\|_\infty}{(\nu_0)_{\min}} + \frac{\|M\|_{2\to\infty} + \kappa\mathcal{B}'}{\sigma_r(M)} + \kappa\mu\sqrt{\frac{r}{n}}\right],$$

*where (iv) holds if in addition* $\|\widehat{M} - M\|_{1\to\infty} \le \frac{1}{2}(\nu_0)_{\min}$.

*Proof.* The first statement of the theorem follows from Lemma 22 (with $T$ instead of $T_\tau$), Lemmas 28 and 32. The remaining bounds are consequences of $(i)$ and of the results presented in Appendix F. $\qquad\square$

*Proof of Theorem 3.* Theorem 3 follows from Lemma 13 and Theorem 14 by simplifying the term $\mathcal{B}'$ using $\mathcal{B}$ given in Theorem 3. As a result of this simplification, as well as of the assumptions given in statement of Theorem 14, we obtain bounds on $n, T$ required in Theorem 3. Furthermore, we use simple inequalities (check Lemma 17) to rewrite all terms depending on $M$ as functions of $\|M\|_\infty$ and $(\nu_0)_{\min}$. $\qquad\square$

**Remark 1.** *It is worth noting that Corollary 4 is a corollary of Theorem 14, and that the lower bound on $n$ required in Theorem 3 is not required for this corollary. Moreover, results presented in this corollary are valid for almost all $T \ge cn\log(nT)$ - in the case when $T \asymp [n^{2-\epsilon}, n^2]$ for arbitrarily*

small $\epsilon > 0$, bounds in Corollary 4 contain additional log *term, which is an artifact of our analysis (and splitting concentration into cases $T \lesssim n^2$ and $T \gtrsim n^2$). This discontinuity in the range of $T$ can be resolved, but at the price of a reduced readability.*

## B.3 Transition matrix estimation under the forward model – Model II(b)

In this subsection, we present a complete, non-simplified version of Theorem 5, from which one can deduce Theorem 5 and Corollary 6 given in the main text.

Again, we use function the funcion $g_\delta$ defined in (9), and we introduce:

$$\mathcal{B}' = \mu\kappa\sqrt{\frac{r}{n}}\left(\sqrt{\frac{\|\nu\|_\infty \tau^\star}{T}}\log\left(\frac{ne}{\delta}\right)\log(T\nu_{\min}^{-1}) + \frac{\tau^\star}{T}g_{\delta/\sqrt{T_\tau}}(T_\tau M)\log\left(\frac{n\sqrt{T_\tau}}{\delta}\right)\log(T\nu_{\min}^{-1})\right)$$
$$+ \sqrt{\frac{r\tau^\star\|M\|_\infty}{T}}\log\left(\frac{n\sqrt{T_\tau}}{\delta}\right)\log(T\nu_{\min}^{-1}).$$

Our analysis starts from the following lemma stating how well $\widetilde{M}$ approximates $M$.

**Lemma 15.** *(Lemma 7 in [63]) For $\tau \geq 2\tau^\star\log(T\nu_{\min}^{-1})$ and for any $\delta \in (0,1)$, we have with probability at least $1 - \delta$: $\|\widetilde{M} - M\| \leq C\sqrt{\frac{\|\nu\|_\infty \tau}{T}\log\left(\frac{ne}{\delta}\right)} + C\frac{\tau}{T}\log\left(\frac{ne}{\delta}\right)$.*

**Theorem 16.** *Assume that $\nu_{\min} = \min_{i \in [n]} \nu_i > 0$ and that $\tau/(\tau^\star\log(T\nu_{\min}^{-1})) \in [c_1, c_2]$ for some universal constants $c_2 > c_1 \geq 2$. For any $\delta > 0$, if $\|\widetilde{M} - M\| \leq c\sigma_r(M)$, $g_{\delta/\sqrt{T}}(T_\tau M)\log(n\sqrt{T_\tau}/\delta) \leq cT_\tau\sigma_r(M)$ and $\|M\|_\infty \log(n\sqrt{T_\tau}/\delta) \leq cT_\tau\sigma_r^2(M)$ for some universal constant $c > 0$, then there exists a universal constant $C > 0$ such that with probability at least $1 - \delta$,*

$(i)$    $\max\left\{\|U - \widehat{U}(\widehat{U}^\top U)\|_{2\to\infty}, \|V - \widehat{V}(\widehat{V}^\top V)\|_{2\to\infty}\right\} \leq C\frac{\mathcal{B}'}{\sigma_r(M)}$,

$(ii)$    $\|\widehat{M} - M\|_{2\to\infty} \leq C\kappa\mathcal{B}'$,    $\|\widehat{P} - P\|_{1\to\infty} \leq C\frac{\kappa\sqrt{n}}{\nu_{\min}}\mathcal{B}'$,

$(iii)$    $\|\widehat{M} - M\|_\infty \leq C\left(\frac{\|M\|_{2\to\infty} + \kappa\mathcal{B}'}{\sigma_r(M)} + \kappa\mu\sqrt{\frac{r}{n}}\right)\mathcal{B}'$,

$(iv)$    $\|\widehat{P} - P\|_\infty \leq C\frac{\mathcal{B}'}{\nu_{\min}}\left[\sqrt{n}\kappa\frac{\|M\|_\infty}{\nu_{\min}} + \frac{\|M\|_{2\to\infty} + \kappa\mathcal{B}'}{\sigma_r(M)} + \kappa\mu\sqrt{\frac{r}{n}}\right]$,

*where (iv) holds if in addition $\|\widehat{M} - M\|_{1\to\infty} \leq \frac{1}{2}\nu_{\min}$.*

*Proof.* The first statement of the theorem follows from Lemmas 21, 22 and 32, whereas the next four bounds follow from $(i)$ and the bounds presented in Appendix F. □

As for the generative model, Theorem 5 is a direct consequence of Theorem 16, and it is obtained by simplifying the term $\mathcal{B}'$ to $\mathcal{B}$. Corollary 6 is also easily derived from Theorem 16.

## B.4 An additional lemma

**Lemma 17.** *Let $M$ be matrix and $m \times n$ matrix with rank $r$, incoherence parameter $\mu > 0$, and condition number $\kappa > 0$. Then, we have*

$$\|M\|_\infty \leq \sigma_1(M)\frac{\mu^2 r}{\sqrt{nm}} \leq \sigma_r(M)\frac{\mu^2\kappa r}{\sqrt{nm}}.$$

*Proof of Lemma 17.* For all $(i, j) \in [m] \times [n]$, we have

$$|M_{i,j}| = \left|\sum_{\ell=1}^r \sigma_\ell(M)u_{i,\ell}v_{j,\ell}\right| \leq \sigma_1(M)\sum_{\ell=1}^r |u_{i,\ell}v_{j,\ell}|$$
$$\leq \sigma_1(M)\|U\|_{2\to\infty}\|V\|_{2\to\infty} \leq \sigma_1(M)\frac{\mu r}{\sqrt{nm}} \leq \sigma_r(M)\frac{\mu\kappa r}{\sqrt{nm}}.$$

The first inequality follows from the triangular inequality and the fact that $\sigma_1(M) \geq \sigma_2(M) \geq \cdots \geq \sigma_r(M)$. The second inequality follows from Cauchy-Schwarz inequality. The last inequalities follow by definition of the incoherence parameter and of the condition number. □

# C Comparison inequalities and the Poisson approximation argument

In this section, we state and prove the results related to the Poisson approximations used to handle the noise correlations in the data. We start by presenting some of the key tools behind the Poisson approximation argument. This argument comes in the form of comparison inequalities. The latter are applied and specified first to Model I (reward matrix estimation), and then to Model II (transition matrix estimation).

## C.1 Preliminaries on Poisson approximation

The Poisson approximation argument comes in the form of an inequality, which is presented in Lemma 19. However, the key idea behind the argument is, roughly speaking, the equality in distribution between a multinomial distribution with $t$ trials and $n$ outcomes, and the joint distribution of $n$ in dependent Poisson random variables with properly chosen parameters, conditioned on some particular event. This equality of distribution is powerful for our purposes precisely because of the independence between the Poisson random variables. Below, we present Lemma 18 that represents this idea.

**Lemma 18.** *(Heterogeneous analogue of Theorem 5.2 in [45]) Let $Y_i^{(t)} \sim \text{Poisson}(tp_i)$, $i = 1, \ldots, n$, be independent random variables with $\sum_{i=1}^n p_i = 1$. Moreover, let $(Z_1^{(t)}, Z_2^{(t)}, \ldots, Z_n^{(t)}) \sim \text{Multinomial}(t, (p_1, \ldots, p_n))$. Then distribution of $(Y_1^{(t)}, \ldots, Y_n^{(t)})$ conditioned on $\sum_{i=1}^n Y_i^{(t)} = s$ is the same as $(Z_1^{(s)}, \ldots, Z_n^{(s)})$ irrespective of $t$.*

*Proof.* The proof follows similar steps as the proof of Theorem 5.2 in [45], but we provide it here for the sake of completeness. First, note that from the definition of multinomial distributions:

$$\mathbb{P}((Z_1^{(s)}, \ldots, Z_n^{(s)}) = (a_1, \ldots, a_n)) = \frac{s!}{a_1! \cdots a_n!} p_1^{a_1} \cdots p_n^{a_n} \tag{10}$$

if $\sum_{i=1}^n a_i = s$, and 0 otherwise. Since the sum of Poisson random variables is a Poisson random variable with parameter equal to the sum of parameters of the initial random variables, we get that the random variable $\sum_{i=1}^n Y_i^{(t)} \sim \text{Poisson}(\sum_{i=1}^n tp_i) = \text{Poisson}(t)$. Hence we have:

$$\mathbb{P}\left((Y_1^{(t)}, \ldots, Y_n^{(t)}) = (a_1, \ldots, a_n) \,\middle|\, \sum_{i=1}^n Y_i^{(t)} = s\right) = \frac{\mathbb{P}((Y_1^{(t)}, \ldots, Y_n^{(t)}) = (a_1, \ldots, a_n))}{\mathbb{P}(\sum_{i=1}^n Y_i^{(t)} = s)}$$

$$= \frac{s!}{\exp(-t)t^s} \prod_{i=1}^n \frac{(tp_i)^{a_i} \exp(-tp_i)}{a_i!}$$

$$= \frac{s!}{a_1! \cdots a_n!} p_1^{a_1} \cdots p_n^{a_n} \tag{11}$$

where in the last step we used the independence of $Y_i^{(t)}$'s and $\sum_{i=1}^n a_i = s$. Note that equations (10) and (11) are exactly the same, which concludes the proof. $\square$

**Lemma 19.** *Consider the setting of Lemma 18 and let $f : \mathbb{R}^p \to \mathbb{R}_+$ be any non-negative function. Then:*

$$\mathbb{E}\left[f(Z_1^{(t)}, \ldots, Z_p^{(t)})\right] \leq e\sqrt{t}\,\mathbb{E}\left[f(Y_1^{(t)}, \ldots, Y_p^{(t)})\right]$$

*Proof.* The proof is essentially the same as that of Theorem 5.7 in [45] with the exception that we use Lemma 18 instead of Theorem 5.6 in [45] and we repeat it here for the sake of completeness.

$$\mathbb{E}[f(Y_1^{(t)}, \ldots, Y_p^{(t)})] = \sum_{k=0}^\infty \mathbb{E}\left[f(Y_1^{(t)}, \ldots, Y_p^{(t)})\,\middle|\, \sum_{i=1}^p Y_i^{(t)} = k\right] \mathbb{P}\left(\sum_{i=1}^p Y_i^{(t)} = k\right)$$

$$\geq \mathbb{E}\left[f(Y_1^{(t)}, \ldots, Y_p^{(t)})\,\middle|\, \sum_{i=1}^p Y_i^{(t)} = t\right] \mathbb{P}\left(\sum_{i=1}^p Y_i^{(t)} = t\right)$$

$$= \mathbb{E}[f(Z_1^{(t)}, \ldots, Z_p^{(t)})] \mathbb{P}\left(\sum_{i=1}^p Y_i^{(t)} = t\right) \tag{12}$$

where in the second line we used non-negativeness of $f$, and in the last line we used Lemma 18. Now, since $\sum_{i=1}^{p} Y_i^{(t)}$ is a Poisson random variable with mean $t$ we have $\mathbb{P}(\sum_{i=1}^{p} Y_i^{(t)} = t) = \frac{t^t \exp(-t)}{t!}$ and using simple inequality $t! < e\sqrt{t}(\frac{t}{e})^t$ we can rewrite inequality (12) as follows:

$$\mathbb{E}[f(Y_1^{(t)}, \ldots, Y_p^{(t)})] \geq \mathbb{E}[f(Z_1^{(t)}, \ldots, Z_p^{(t)})] \frac{1}{e\sqrt{t}} \tag{13}$$

which gives statement of the lemma. $\qquad\square$

## C.2 Poisson approximation for reward matrices – Model I

We recall from Section 4 that the definition of the empirical reward matrix $\widetilde{M}$ is given as follows

$$\forall (i,j) \in [n] \times [m], \qquad \widetilde{M}_{i,j} = \frac{nm}{T} \sum_{t=1}^{T} (M_{i_t,j_t} + \xi_t) \mathbb{1}_{\{(i_t,j_t)=(i,j)\}} \tag{14}$$

where $(i_t, j_t)$ are sampled uniformly at random from $[n] \times [m]$. Due to independence between $(i_1, j_1), \ldots, (i_T, j_T)$ and $\xi_1, \ldots, \xi_T$, we note that the observation model (14) is equivalent in distribution to the following one

$$\forall (i,j) \in [n] \times [m], \qquad \widetilde{M}_{i,j} = \frac{nm}{T} \sum_{t=1}^{Z_{i,j}} (M_{i,j} + \xi'_{i,j,t})$$

where we for all $(i,j) \in [n] \times [j]$, $(\xi'_{i,j,t})_{t \geq 1}$ is a sequence of i.i.d. random variables copies, say of $\xi_1$, and

$$Z_{i,j} = \sum_{t=1}^{T} \mathbb{1}_{\{(i_t,j_t)=(i,j)\}}.$$

Observe that $Z = (Z_{i,j})_{(i,j)}$ is a multinomial random variable whose parameters are defined by the fact that for all $t \in [T]$, $\mathbb{P}((i_t, j_t) = (i,j)) = 1/nm)$. We denote $\mathbb{P}$ the joint probability of the entries of $Z$ and sequences $(\xi_{i,j,t})_{t \geq 1}$, $(i,j) \in [n] \times [m]$.

**Compound Poisson random matrix model.** We define a random matrix $Y \in \mathbb{R}^{n \times m}$ generated by a Poisson model as follows:

$$Y_{i,j} \sim \text{Poisson}(T/nm), \qquad (i,j) \in [n] \times [m]$$

and denote $\mathbb{P}'$ the joint probability of the entries of $Y$ and the sequences $(\xi'_{i,j,t})_{t \geq 1}$, for $(i,j) \in [n] \times [m]$. We may then consider the matrix model

$$X_{i,j} = \sum_{t=1}^{Y_{i,j}} (M_{i,j} + \xi'_{i,j,t}). \tag{15}$$

We note that the entries of the matrix $X$ are distributed according to compound Poisson distributions. Below, we precise the Poisson approximation argument for the reward matrix model.

**Lemma 20** (Poisson Approximation). *Let $(\Omega, \mathcal{F}, \mathbb{P})$ (resp. $(\Omega, \mathcal{F}, \mathbb{P}')$) be the probability space under the matrix-plus-noise model (14) (resp. (15)). Then for any event $\mathcal{E} \in \mathcal{F}$, we have*

$$\mathbb{P}(\mathcal{E}) \leq e\sqrt{T} \mathbb{P}'(\mathcal{E}).$$

*Proof of Lemma 20.* For convenience, we denote $X = ((\xi_{i,j,t})_{t \geq 1})_{i,j \in [n] \times [m]}$. We set $f(Z, X) = \mathbb{1}_{\{\mathcal{E}\}}$. Thanks to Lemma 19, given that $Z$ is independent of $X$, we have

$$\mathbb{E}[f(Z, X)|X] \leq e\sqrt{T} \mathbb{E}[f(Y, X)|X].$$

We further take the expectation on $X$ and write

$$\mathbb{P}(\mathcal{E}) = \mathbb{E}[f(Z, X)] \leq e\sqrt{T} \mathbb{E}[f(Y, X)] = e\sqrt{T} \mathbb{P}'(\mathcal{E}).$$

$\qquad\square$

## C.3 Approximations for transition matrices – Model II

We restrict our attention to the forward model, Model II(b). The results for the generative model are simpler and can be easily deduced from those for the forward model. Recall from Section 3.3 definition of matrix $\widetilde{M}^{(k)}$ and in the following discussion we fix value of $k \in [\tau]$. Define a matrix $N = T_\tau \widetilde{M}^{(k)}$ and note that it is equal to:

$$N_{i,j} = \sum_{l=0}^{T_\tau - 1} \mathbb{1}_{\{(x_{k+l\tau}, x_{k+1+l\tau}) = (i,j)\}}, \qquad i, j = 1, 2, \ldots, n \tag{16}$$

Furthermore, let $\mathbb{P}_1$ be joint probability distribution of entries of $N$.

### C.3.1 Multinomial approximation

Here we define a matrix $Z \in \mathbb{R}^{n \times n}$ with entries:

$$Z_{i,j} = \sum_{t=0}^{T_\tau - 1} \mathbb{1}_{\{(i_t, j_t) = (i,j)\}}, \qquad i, j = 1, 2, \ldots, n, \tag{17}$$

where $\mathbb{P}((i_t, j_t) = (i,j)) = \nu_i P_{i,j}$ independently over $i, j \in [n]$ and $t \in [T_\tau]$. Denote by $\mathbb{P}_2$ joint probability distribution of entries of $Z$. Then we have:

**Lemma 21.** *Let $N$ and $Z$ be matrices obtained under the models* (16) *and* (17)*, respectively. Then, for any subset $\mathcal{Z}$ of $\{z \in \mathbb{N}^{n^2} : \sum_{(i,j)} z_{i,j} = T_\tau\}$, we have $\mathbb{P}(N \in \mathcal{Z}) \leq 3\mathbb{P}(Z \in \mathcal{Z})$.*

*Proof.* Note that by subsampling as explained in Section 3.3, for each $k$ we obtain a Markov chain with transition kernel

$$P_\tau((y, y')|(x, x')) = P^\tau(y|x')P(y'|y)$$

and initial distribution $\nu_0^{(k)}(x, x') = \nu_0^{(k)}(x)P(x'|x)$ with

$$\nu_0^{(1)}(x) = \nu_0(x) \qquad \text{and} \qquad \nu_0^{(k)}(x) = \sum_{y \in [n]} \nu_0(y) P^{k-1}(x|y) \quad \text{for } k = 2, \ldots, \tau.$$

Moreover, all chains share the same stationary distribution given by $\Pi \in \mathbb{R}^{n \times n}$ with $\Pi_{x,x'} = \nu(x)P(x'|x)$, $x, x' \in [n]$. Now, recall definition of $\tau$ from Theorem 5 and note that according to Lemma 10 with $\delta = \frac{1}{4}$ and $\varepsilon = \nu_{\min}/(eT)$ we have $\tau(\varepsilon) \leq \tau$ and thus:

$$\max_{1 \leq i \leq n} \|P_{i,:}^\tau - \nu^\top\|_1 \leq \frac{\nu_{\min}}{eT}. \tag{18}$$

Now, let $z = (z_{i,j})_{i,j=1}^n \in \{z \in \mathbb{N}^{n^2} : \sum_{i,j} z_{i,j} = T_\tau\}$ be a tuple of fixed integers. Define a set:

$$\mathcal{S}(z) := \{(a_{2l+1}, a_{2l+2})_{l=0}^{T_\tau - 1} \in ([n] \times [n])^{T_\tau} : \sum_{l=0}^{T_\tau - 1} \mathbb{1}_{\{(a_{2l+1}, a_{2l+2}) = (i,j)\}} = z_{i,j}, \forall i, j \in [n]\}$$

and note that $|\mathcal{S}(z)| = T_\tau! (\prod_{i,j=1}^n z_{i,j}!)^{-1}$. By definition of Markovian and multinomial models, we have:

$$\mathbb{P}(N = z) = \sum \nu_0^{(k)}(x_{k-1}, x_k) \prod_{l=1}^{T_\tau - 1} P_\tau((x_{k-1+l\tau}, x_{k+l\tau})|(x_{k-1+(l-1)\tau}, x_{k+(l-1)\tau}))$$

where the sum is over $(x_{k-1+l\tau}, x_{k+l\tau})_{l=0}^{T_\tau - 1} \in \mathcal{S}(z)$, and

$$\mathbb{P}(Z = z) = \frac{T_\tau!}{\prod_{i,j=1}^n z_{i,j}!} \prod_{i,j=1}^n \Pi_{i,j}^{z_{i,j}}.$$

Now we fix arbitrarily one of the summands in the expression for $\mathbb{P}(N = z)$ and note that:

$$\left| \nu_0^{(k)}(x_{k-1}, x_k) \prod_{l=1}^{T_\tau - 1} P_\tau((x_{k-1+l\tau}, x_{k+l\tau}) | (x_{k-1+(l-1)\tau}, x_{k+(l-1)\tau})) - \prod_{i,j=1}^{n} \Pi_{i,j}^{z_{i,j}} \right|$$

$$= \left( \prod_{l=0}^{T_\tau - 1} P(x_{k+l\tau} | x_{k-1+l\tau}) \right) \left| \nu_0^{(k)}(x_{k-1}) \prod_{l=1}^{T_\tau - 1} P^\tau(x_{k-1+l\tau} | x_{k+(l-1)\tau}) - \prod_{l=0}^{T_\tau - 1} \nu(x_{k-1+l\tau}) \right|$$

$$\leq \left( \prod_{l=0}^{T_\tau - 1} P(x_{k+l\tau} | x_{k-1+l\tau}) \right) \left( \prod_{l=0}^{T_\tau - 1} (\nu(x_{k-1+l\tau}) + \epsilon) - \prod_{l=0}^{T_\tau - 1} \nu(x_{k-1+l\tau}) \right)$$

$$\leq \left( \prod_{i,j=1}^{n} \Pi_{i,j}^{z_{i,j}} \right) \sum_{j=1}^{T_\tau} \left( \frac{\epsilon}{\nu_{\min}} \right)^j \binom{T_\tau}{j} \leq \left( \prod_{i,j=1}^{n} \Pi_{i,j}^{z_{i,j}} \right) \sum_{j=1}^{T_\tau} \left( \frac{eT_\tau \epsilon}{j \nu_{\min}} \right)^j \leq 2 \left( \prod_{i,j=1}^{n} \Pi_{i,j}^{z_{i,j}} \right)$$

where in first inequality we used Equation (18), where we then used the bound on binomial coefficients $\binom{T_\tau}{j} \leq (eT_\tau / j)^j$, and where in the last inequality, we used definition of $\epsilon$. Since this upper bound holds irrespective of the summand, we deduce that:

$$|\mathbb{P}(N = z) - \mathbb{P}(Z = z)| \leq 2 \frac{T_\tau!}{\prod_{i,j=1}^{n} z_{i,j}!} \left( \prod_{i,j=1}^{n} \Pi_{i,j}^{z_{i,j}} \right) = 2\mathbb{P}(Z = z).$$

Now, let $\mathcal{Z}$ be any subset of $\{z \in \mathbb{N}^{n^2} : \sum_{(i,j)} z_{i,j} = T_\tau\}$. Then we have:

$$\mathbb{P}(N \in \mathcal{Z}) = \sum_{z \in \mathcal{Z}} \mathbb{P}(N = z) \leq 3 \sum_{z \in \mathcal{Z}} \mathbb{P}(Z = z) = 3\mathbb{P}(Z \in \mathcal{Z})$$

as claimed in the lemma.

$\square$

### C.3.2   Poisson approximation

We define a matrix $Y \in \mathbb{R}^{n \times n}$ generated by the Poisson model as follows:

$$Y_{i,j} \sim \text{Poisson}(T_\tau M_{i,j}), \qquad i, j = 1, 2, \ldots, n. \tag{19}$$

We show that rare random events occur with approximately equal probability for the Poisson and multinomial models:

**Lemma 22.** *Let $Z$ and $Y$ be matrices obtained under the models* (17) *and* (19)*, respectively. Then for any $\mathcal{Z} \subset \mathbb{N}^{n^2}$, we have $\mathbb{P}(Z \in \mathcal{Z}) \leq e\sqrt{T_\tau}\mathbb{P}(Y \in \mathcal{Z})$.*

*Proof.* Proof of the lemma is a straightforward consequence of Lemma 19 with parameters $T_\tau$, $n^2$ and $f = \mathbb{1}_{\{\mathcal{Z}\}}$.  $\square$

# D Concentration of matrices with Poisson and compound Poisson entries

As mentioned in Appendix C, our analysis relies on a Poisson approximation argument. As a result, we will require tight concentration bounds for random matrices with entries distributed according to compound Poisson distributions (when estimating the reward matrix) and Poisson distributions (when estimating the transition matrices). In §D.1, we present a few simple facts about Poisson and compound Poisson random variables, together with some other useful tools. In §D.2, we present two concentration results, required for the model with compound Poisson entries. Similarly, in §D.3, we present two concentration results, required for the model with Poisson entries. These concentration results will be extensively used in the forthcoming analysis for the subspace recovery.

It is worth noting that our results in §D.3 are sharper than those in §D.2 thanks to Bennett's inequality. As a consequence, our results for estimating the reward matrix exhibit a dependence in $\log^3(n + m)$ while in the estimation of the transitions, our results exhibit a dependence in $\log^2(n)$ and even $\log(n)$ in some regimes.

## D.1 Preliminaries

We first present Theorem 23, which can be seen as a version of matrix Bernstein inequality. The theorem is borrowed from [26] and relies on a truncation trick. The proofs of our concentration results in §D.2 and §D.3 rely on this theorem.

**Theorem 23.** *(Proposition A.3 in [26]) Let $\{Z_t\}_{t=1}^T$ be a sequence of $m \times n$ independent zero-mean real random matrices. Suppose that for all $1 \leq t \leq T$,*

$$(i) \ \ \mathbb{P}\left(\|Z_t\| \geq \beta\right) \leq p, \qquad and \qquad (ii) \ \ \left\|\mathbb{E}[Z_t \mathbb{1}_{\{\|Z_t\|>\beta\}}]\right\| \leq q, \tag{20}$$

*hold for some quantities $p \in (0,1)$, and $q \geq 0$. Furthermore, assume there exists $v \geq 0$, such that*

$$(iii) \ \ \max\left\{\left\|\sum_{t=1}^T \mathbb{E}\left[Z_t Z_t^\top\right]\right\|, \left\|\sum_{t=1}^T \mathbb{E}\left[Z_t^\top Z_t\right]\right\|\right\} \leq v. \tag{21}$$

*Then, for all $u > 0$,*

$$\mathbb{P}\left(\left\|\sum_{t=1}^T Z_t\right\| \geq Tq + u\right) \leq Tp + (n+m)\exp\left(-\frac{u^2/2}{v + \beta u/3}\right). \tag{22}$$

To apply Theorem 23, we need control of the tails of the entries of the random matrix we study. In the case of Poisson entries, we will simply use the following standard fact about Poisson random variables. It is a simple consequence of Bennett's inequality [8].

**Lemma 24.** *Let $Y$ be a Poisson random variable with mean $\lambda$. Then for, all $\theta \in \mathbb{R}$, we have $\mathbb{E}[e^{\theta Y}] \leq \exp(\lambda(e^\theta - 1))$. Furthermore, we have for all $u > 0$*

$$\mathbb{P}(|Y - \lambda| > u) \leq 2\exp\left(-\lambda h(u/\lambda)\right) \leq 2\exp\left(-\frac{u^2/2}{\lambda + u/3}\right),$$

*where $h(u) = (1 + u)\log(1 + u) - u$.*

In the case of compound Poisson entries, we do not have any result similar to Bennett's inequality. Instead, we derive a Bernstein-type concentration result on these random variables.

**Lemma 25.** *Let $(\xi_t)_{t \geq 1}$ be a sequence of zero-mean, $\sigma^2$-subgaussian, i.i.d. random variables. Let $Y$ be a Poisson random variables with mean $\lambda$. Let $M$ be a positive constant. Then, the moment generating function of the compound Poisson random variable $Z = \sum_{i=1}^Y (M + \xi_i)$ satisfies the following:*

$$\forall u > 0, \qquad \mathbb{P}(|Z - \lambda M| > u) \leq 2\exp\left(-\min\left(\frac{u^2}{16e\lambda L^2}, \frac{u}{4L}\right)\right),$$

$$\mathbb{E}\left[|Z - \lambda M|^2\right] \leq 18\lambda L^2,$$

*where $L = \max(M, \sigma)$.*

*Proof of Lemma 25.* First, we upper bound the moment generating function of $\sum_{i=10}^{Y}(M + \xi_i)$. Let $\theta > 0$, we have

$$I_Z(\theta) \triangleq \mathbb{E}\left[e^{\theta(\sum_{i=1}^{Y}(M+\xi_i))}\right] \leq \sqrt{\mathbb{E}\left[e^{2\theta MY}\right]\mathbb{E}[e^{2\theta\sum_{i=0}^{Y}\xi_i}]}$$

$$\leq \exp\left(\frac{\lambda(e^{2\theta M}-1)}{2}\right)\sqrt{\mathbb{E}[e^{2\theta\sum_{i=0}^{Y}\xi_i}]}$$

$$\leq \exp\left(\frac{\lambda}{2}\left((2\theta M)^2 e^{2\theta M}+2\theta M\right)\right)\sqrt{\mathbb{E}[e^{2\theta\sum_{i=0}^{Y}\xi_i}]},$$

where in the first inequality, we use Cauchy-Schwarz inequality, in the second inequality, we use the well known bound on the moment generating function of a Poisson random variable (if $Y$ is a Poisson random variable with mean $\lambda$, then for all $\theta > 0$, $\mathbb{E}[e^{\theta Y}] \leq \exp(\lambda(e^\theta - 1)))$, and in the last inequality, we use the elementary fact that $e^x - 1 \leq x^2 e^x + x$ for all $x \in \mathbb{R}$. Next, we have

$$\mathbb{E}\left[e^{2\theta\sum_{i=1}^{Y}\xi_i}\right] = \mathbb{E}\left[\sum_{k=1}^{\infty}\mathbb{1}_{\{Y=k\}}\exp\left(2\theta\sum_{i=1}^{k}\xi_i\right)\right]$$

$$= \sum_{k=1}^{\infty}\mathbb{P}(Y=k)\mathbb{E}\left[\exp\left(2\theta\sum_{i=1}^{k}\xi_i\right)\right]$$

$$\leq \sum_{k=1}^{\infty}\mathbb{P}(Y=k)\exp(2k\theta^2\sigma^2)$$

$$\leq \exp\left(\lambda(e^{2\theta^2\sigma^2}-1)\right)$$

$$\leq \exp\left(\lambda\left(2\theta^2\sigma^2 e^{2\theta^2\sigma^2}\right)\right),$$

where we use the fact that the $\xi_i$ are $\sigma^2$-subgaussian r.v., and the elementary inequality $e^{x^2}-1 \leq x^2 e^{x^2}$ for all $x \in \mathbb{R}$. We conclude that for all $\theta > 0$,

$$I_Z(\theta) \leq \exp\left(\lambda\left(2\theta^2 M^2 e^{2\theta M}+2\theta^2\sigma^2 e^{2\theta^2\sigma^2}\right)+\lambda\theta M\right).$$

Next, we introduce $L = \max(M, \sigma)$. Then, for all $\alpha > 0$, we deduce that

$$I_Z(\theta) \leq \exp\left(2\lambda\theta^2 L^2\left(e^\alpha + e^{\alpha^2}\right)+\lambda\theta M\right), \qquad \forall|\theta| \leq \frac{\alpha}{2L}.$$

By Markov inequality, and fixing $\alpha = 1$, we have

$$\mathbb{P}(Z - \lambda M > u) \leq \inf_{|\theta|\leq 1/(2L)}I_Z(\theta)e^{-\lambda\theta M-\theta u} \leq \exp\left(-\min\left(\frac{u^2}{16e\lambda L^2}, \frac{u}{4L}\right)\right).$$

Similarly, we have

$$\mathbb{P}(\lambda M - Z > u) \leq \exp\left(-\min\left(\frac{u^2}{16e\lambda L^2}, \frac{u}{4L}\right)\right).$$

The final tail bound follows from a union bound. Finally, straightforward computations yield an upper bound on $\mathbb{E}[|\lambda M - Z|^2]$. Indeed, we have

$$\mathbb{E}[|\lambda M - Z|^2] \leq 2\mathbb{E}[|Y-\lambda|^2]M^2+2\mathbb{E}\left[\left(\sum_{i=1}^{Y}\xi_i\right)^2\right] \leq 2\lambda M^2+16\lambda\sigma^2 \leq 18\lambda L^2.$$

$\square$

### D.2 Random matrices with compound Poisson entries

We list below the two main concentration results that we need for the forthcoming analysis. In Proposition 26, we provide a high probability guarantee on the error between the empirical mean reward matrix and the true matrix in operator norm. In Proposition 27, we establish another concentration result that will be instrumental in the subspace recovery analysis. The proofs of the two results are similar with slight differences and they both rely on Theorem 23. The proofs are presented at the end of this subsection.

**Proposition 26.** *Under the random matrix model* (15) *with compound Poisson entries, for all* $\delta \in (0,1)$, *for all* $T \geq 13(n+m)\log^3((n+m)/\delta)$, *the following statement*

$$\|\widetilde{M} - M\| \leq 36\sqrt{2}L\sqrt{\frac{nm}{T}}\left(\sqrt{(n+m)\log\left(\frac{n+m}{\delta}\right)} + \log^{3/2}\left(\frac{n+m}{\delta}\right)\right)$$

*holds with probability at least* $1 - \delta$, *where* $L = \max(\|M\|_\infty, \sigma)$.

**Proposition 27.** *Let $A$ be a $m \times 2r$ nonrandom matrix, and $B$ be a $n \times 2r$ nonrandom matrix. Then, under the random matrix model* (15) *with compound Poisson entries, and denoting* $L = \max(\|M\|_\infty, \sigma)$, *we have:*

*(i) for all $\ell \in [m]$, for all $\delta \in (0,1)$, for all $T \geq m\log^3(en/\delta)$, the following event*

$$\|(\widetilde{M}_{\ell,:} - M_{\ell,:})A\| \leq 73\sqrt{2}L\|A\|_{2\to\infty}\sqrt{\frac{nm}{T}}\left(\sqrt{n\log\left(\frac{en}{\delta}\right)} + \log^{3/2}\left(\frac{en}{\delta}\right)\right) \quad (23)$$

*holds with probability at least* $1 - \delta$;

*(ii) for all $k \in [n]$, for all $\delta \in (0,1)$, for all $T \geq n\log^3(em/\delta)$, the following event*

$$\|(\widetilde{M}_{:,k} - M_{:,k})^\top B\| \leq 73\sqrt{2}L\|B\|_{2\to\infty}\sqrt{\frac{nm}{T}}\left(\sqrt{m\log\left(\frac{em}{\delta}\right)} + \log^{3/2}\left(\frac{em}{\delta}\right)\right) \quad (24)$$

*holds with probability at least* $1 - \delta$.

*Proof of Proposition 26.* To simplify the notation, introduce the matrices $Z_{i,j} = (\widetilde{M}_{i,j} - M_{i,j})e_i e_j^\top$, for all $(i,j) \in [m] \times [n]$, $\lambda = T/nm$, and $L = \max(\|M\|_\infty, \sigma)$. We remark that we can write

$$\widetilde{M} - M = \sum_{(i,j)\in[m]\times[n]} Z_{i,j}.$$

Starting from the above expression, we will apply Theorem 23 to obtain the desired result. First, we note that for all $(i,j) \in [m] \times [n]$, $\|Z_{i,j}\| = |\widetilde{M}_{i,j} - M_{i,j}|$ and $\widetilde{M}_{i,j} - M_{i,j}$ is a centered and normalized compound Poisson random variable. Thus, we have by Lemma 25, for all $\delta \in (0,1)$, $\mathbb{P}(\|Z_{i,j}\| > \beta) \leq \delta/(2n^2m^2)$, where we define

$$\beta = 4L\max\left(\sqrt{\frac{e}{\lambda}\log\left(\frac{4n^2m^2}{\delta}\right)}, \frac{1}{\lambda}\log\left(\frac{4n^2m^2}{\delta}\right)\right),$$

$$\leq 4L\max\left(\sqrt{\frac{4e}{\lambda}\log\left(\frac{n+m}{\delta}\right)}, \frac{4}{\lambda}\log\left(\frac{n+m}{\delta}\right)\right).$$

Moreover, we have

$$\mathbb{E}\left[\|Z_{i,j}\|\mathbb{1}_{\{\|Z_{i,j}\|>\beta\}}\right] \leq \sqrt{\mathbb{E}\left[\|Z_{i,j}\|^2\right]\mathbb{E}\left[\mathbb{1}_{\{\|Z_{i,j}\|>\beta\}}\right]}$$

$$\leq \sqrt{\mathbb{E}[|\widetilde{M}_{i,j} - M_{i,j}|^2]\mathbb{P}(\|Z_{i,j}\| > \beta)}$$

$$\leq \sqrt{\frac{9L^2\delta}{\lambda n^2 m^2}},$$

where the first inequality follows from Cauchy-Schwarz inequality, the second inequality follows from the expression of $Z_{i,j}$, and the third inequality follows from Lemma 25. Next, we have

$$\left\|\sum_{(i,j)\in[m]\times[n]} \mathbb{E}\left[Z_{i,j}Z_{i,j}^\top\right]\right\| = \left|\sum_{i\in[m]}\left(\sum_{j\in[n]} \mathbb{E}\left[\left(\widetilde{M}_{i,j} - M_{i,j}\right)^2\right]\right)e_i e_i^\top\right|$$

$$= \max_{i\in[m]}\sum_{j\in[n]} \mathbb{E}\left[\left(\widetilde{M}_{i,j} - M_{i,j}\right)^2\right]$$

$$\leq \frac{18nL^2}{\lambda}.$$

By symmetry, we obtain similarly

$$\left\| \sum_{(i,j)\in[m]\times[n]} \mathbb{E}\left[Z_{i,j}^{\top} Z_{i,j}\right] \right\| \leq \frac{18mL^2}{\lambda}.$$

Let us set $v = 18(n \wedge m)L^2/\lambda$. We conclude using Theorem 23 that, for all $u > 0$,

$$\mathbb{P}\left(\|\widetilde{M} - M\| > \sqrt{\frac{9L^2\delta}{\lambda}} + u\right) \leq \frac{\delta}{2(nm)} + (n+m)\exp\left(-\frac{u^2/2}{v + \beta u/3}\right)$$

$$\leq \frac{\delta}{2(nm)} + (n+m)\exp\left(-\frac{1}{4}\min\left(\frac{u^2}{v}, \frac{3u}{\beta}\right)\right).$$

We re-parametrize by choosing $\delta = 2(n+m)\exp(-(1/4)\min(u^2/v, 3u/\beta))$, and write

$$\mathbb{P}\left(\|\widetilde{M} - M\| > \frac{3L\sqrt{\delta}}{\sqrt{\lambda}} + u\right) \leq \delta \tag{25}$$

with

$$u = \max\left(\sqrt{4v\log\left(\frac{2(n+m)}{\delta}\right)}, \frac{4\beta}{3}\log\left(\frac{2(n+m)}{\delta}\right)\right)$$

$$\leq \max\left(\sqrt{8v\log\left(\frac{n+m}{\delta}\right)}, \frac{8\beta}{3}\log\left(\frac{n+m}{\delta}\right)\right).$$

By inspecting the definition of $\beta$ and $v$, we note that under the condition

$$\lambda = \frac{T}{nm} \geq \frac{4^5}{3^4}\frac{1}{n \wedge m}\log^3\left(\frac{n+m}{\delta}\right) \tag{26}$$

then

$$u \leq \max\left(\sqrt{8v\log\left(\frac{n+m}{\delta}\right)}, \frac{16L\sqrt{2e}}{3\sqrt{\lambda}}\log^{3/2}\left(\frac{2(n+m)}{\delta}\right)\right)$$

$$\leq \frac{L}{\sqrt{\lambda}}\max\left(\sqrt{3^2 4^2 (n \wedge m)\log\left(\frac{n+m}{\delta}\right)}, \frac{4^2 2\sqrt{e}}{3}\log\left(\frac{n+m}{\delta}\right)\right).$$

After using the upper bound on $u$ in (25), and after upper bounding $\delta$ by 1, we obtain, under the condition (26),

$$\|\widetilde{M} - M\| > \frac{L}{\sqrt{\lambda}}\left(3 + 12\sqrt{2(n+m)\log\left(\frac{n+m}{\delta}\right)} + \frac{4^3\sqrt{e}}{3}\log^{3/2}\left(\frac{n+m}{\delta}\right)\right)$$

$$> \frac{L}{\sqrt{\lambda}}\left(36\sqrt{2(n+m)\log\left(\frac{n+m}{\delta}\right)} + 36\log^{3/2}\left(\frac{n+m}{\delta}\right)\right)$$

$$> \frac{36\sqrt{2}L}{\sqrt{\lambda}}\left(\sqrt{(n+m)\log\left(\frac{n+m}{\delta}\right)} + \log^{3/2}\left(\frac{n+m}{\delta}\right)\right)$$

with probability at most $\delta$. Noting that a stricter condition than (26) is

$$T \geq 13(n+m)\log^3\left(\frac{n+m}{\delta}\right),$$

we complete the proof. $\qquad\square$

*Proof of Proposition 27.* To simplify the notation, let us denote $Z_j = (\widetilde{M}_{\ell,j} - M_{\ell,j})A_{j,:}$, $\lambda = mn/T$, and $L = \max(\|M\|_\infty, \sigma)$. We remark that we can write

$$(\widetilde{M}_{\ell,:} - M_{\ell,:})A = \sum_{j \in [n]} (\widetilde{M}_{\ell,j} - M_{\ell,j})A_{j,:} = \sum_{j \in [n]} Z_j.$$

Starting from the above expression, we will apply 23 to obtain the desired result. First, we note that for all $j \in [n]$, $\|Z_j\| = |\widetilde{M}_{\ell,j} - M_{\ell,j}|\|A_{j,:}\|$, and $\widetilde{M}_{\ell,j} - M_{\ell,j}$ is a centered and normalized compound Poisson random variable. Thus, we have by Lemma 25, for all $\delta \in (0,1)$, $\mathbb{P}(\|Z_j\| > \|A\|_{2\to\infty}\beta) \leq \mathbb{P}(\|Z_j\| > \|A_{j,:}\|\beta) \leq \delta/(2n^2)$, where we define

$$\beta = 4L \max\left(\sqrt{\frac{e}{\lambda} \log\left(\frac{4n^2}{\delta}\right)}, \frac{1}{\lambda} \log\left(\frac{4n^2}{\delta}\right)\right)$$

$$\leq 4L \max\left(\sqrt{\frac{2e}{\lambda} \log\left(\frac{en}{\delta}\right)}, \frac{2}{\lambda} \log\left(\frac{en}{\delta}\right)\right).$$

Moreover, we have

$$\mathbb{E}\left[\|Z_j\|\mathbb{1}_{\{\|Z_j\| > \|A\|_{2\to\infty}\beta\}}\right] \leq \sqrt{\mathbb{E}[\|Z_j\|^2]\mathbb{P}(\|Z_j\| > \|A\|_{2\to\infty}\beta)}$$

$$\leq \|A\|_{2\to\infty}\sqrt{\frac{\mathbb{E}[\|\widetilde{M}_{\ell,:} - M_{\ell,:}\|^2]\delta}{2n^2}}$$

$$\leq \|A\|_{2\to\infty}\sqrt{\frac{9L^2\delta}{\lambda n^2}},$$

where in the first inequality, we use Cauchy-Schwarz inequality, and in the third inequality, the result of Lemma 25 to upper bound the variances. Next, we have

$$\left\|\mathbb{E}\left[\sum_{j \in [n]} Z_j Z_j^\top\right]\right\| \leq \sum_{j \in [n]} \mathbb{E}\left[(\widetilde{M}_{\ell,j} - M_{\ell,j})^2\right]\|A_{j,:}\|^2$$

$$\leq \frac{18L^2\|A\|_F^2}{\lambda}$$

$$\leq \frac{18L^2 n\|A\|_{2\to\infty}^2}{\lambda},$$

where we simply used the expressions of $Z_j$, $j \in [n]$, the triangular inequality, and Lemma 25 to upper bound the variances. Similarly, we have

$$\left\|\mathbb{E}\left[\sum_{j \in [n]} Z_j^\top Z_j\right]\right\| \leq \frac{18L^2 n\|A\|_{2\to\infty}^2}{\lambda}.$$

We set $v = 18L^2 n\|A\|_{2\to\infty}^2/\lambda$. Now we are ready to apply Theorem 23. We get:

$$\mathbb{P}\left(\|(\widetilde{M}_{\ell,:} - M_{\ell,:})A\| > \|A\|_{2\to\infty}\sqrt{\frac{9L^2\delta}{\lambda}} + u\right) \leq \frac{\delta}{2n} + n\exp\left(-\frac{1}{4}\min\left(\frac{u^2}{v}, \frac{3u}{\|A\|_{2\to\infty}\beta}\right)\right).$$

We re-parametrize by choosing $\delta = 2n\exp(-(1/4)\min(u^2/v, 3u/(\|A\|_{2\to\infty}\beta)))$ and we write

$$\mathbb{P}\left(\|(\widetilde{M}_{\ell,:} - M_{\ell,:})A\| > \|A\|_{2\to\infty}\sqrt{\frac{9L^2\delta}{\lambda}} + u\right) \leq \delta$$

with

$$u = \max\left(\sqrt{4v \log\left(\frac{2n}{\delta}\right)}, \frac{4\|A\|_{2\to\infty}\beta}{3} \log\left(\frac{2n}{\delta}\right)\right)$$

$$\leq \max\left(\sqrt{4v \log\left(\frac{en}{\delta}\right)}, \frac{4\|A\|_{2\to\infty}\beta}{3} \log\left(\frac{en}{\delta}\right)\right).$$

By inspecting the definition of $\beta$ and $v$, we note that when the condition

$$\lambda = \frac{T}{mn} \geq \frac{4^3}{3^4}\frac{1}{n}\log^3\left(\frac{en}{\delta}\right) \tag{27}$$

holds, then

$$
u \leq \max\left(\sqrt{4v\log\left(\frac{en}{\delta}\right)}, \frac{16\sqrt{2e}L\|A\|_{2\to\infty}}{3\sqrt{\lambda}}\log^{3/2}\left(\frac{en}{\delta}\right)\right)
$$

$$
\leq \frac{L\|A\|_{2\to\infty}}{\sqrt{\lambda}}\max\left(\sqrt{2^3 3^2 n\log\left(\frac{en}{\delta}\right)}, \frac{16\sqrt{2e}}{3}\log^{3/2}\left(\frac{en}{\delta}\right)\right)
$$

$$
\leq \frac{36\sqrt{2}L\|A\|_{2\to\infty}}{\sqrt{\lambda}}\max\left(\sqrt{n\log\left(\frac{en}{\delta}\right)}, \log^{3/2}\left(\frac{en}{\delta}\right)\right). \tag{28}
$$

After using the upper bound in (28), and upper bounding $\delta$ by 1, we obtain that, under the condition (27),

$$
\|(\widetilde{M}_{\ell,:} - M_{\ell,:})A\| > \frac{73\sqrt{2}L\|A\|_{2\to\infty}}{\sqrt{\lambda}}\left(\sqrt{n\log\left(\frac{en}{\delta}\right)} + \log^{3/2}\left(\frac{en}{\delta}\right)\right)
$$

holds with probability at most $\delta$. We can also refine the condition (27) as follows

$$
T \geq m\log^3\left(\frac{en}{\delta}\right).
$$

This concludes the proof of the statement (23) in the proposition. The statement (24) follows similarly. Therefore, we omit it. □

### D.3  Random matrices with Poisson entries

Recall from Section B.2, the definition of the function $g_\delta$ from (9) and that $\mathcal{A} = \frac{1}{\sqrt{T}}\sqrt{\|M\|_{1\to\infty} + \|M^\top\|_{1\to\infty}}$. First we show the following lemma that provides an upper bound of the spectral norm. This lemma is used to derive Lemma 13.

**Lemma 28.** *Let $Y \in \mathbb{R}^{n\times n}$ be a matrix with independent entries $Y_{i,j} \sim T^{-1}\mathrm{Poisson}(TM_{ij})$, $i,j \in [n]$, and let $0 \leq \delta \leq 1$. Then, w.p. at least $1 - \delta$, $\|Y - M\| \leq C\mathcal{A} + \frac{C}{T}g_\delta(TM)\sqrt{\log(\frac{ne}{\delta})}$.*

*Proof.* The proof follows from that of Lemma 29 and that of Lemma 4 in [42], which is based on a spectral bound from [6]. We use that the random variables $|Y_{i,j} - M_{i,j}|$ concentrate well around $L = L_1 \mathbb{1}_{\{\exists \ell: T\|M_{\ell,:}\|_\infty \leq 1\}} + L_2 \mathbb{1}_{\{\forall \ell: T\|M_{\ell,:}\|_\infty > 1\}}$ where $L_1 = 4T^{-1}\log^{-1}(1 + (T\|M\|_\infty)^{-1} \wedge n\delta^{-1})\log(\frac{ne}{\delta})$ and $L_2 = 4\sqrt{T^{-1}\|M\|_\infty}\log\left(T\|M\|_\infty \frac{ne}{\delta}\right)$ using exactly the same argument as in the first step of Lemma 29. Moreover, we use upper bound on $|\mathbb{E}[(Y_{i,j} - M_{i,j})\mathbb{1}_{\{|Y_{i,j}-M_{i,j}|<L\}}]|$ derived in the second step of Lemma 29. □

We also derive upper bounds in the $\ell_{2\to\infty}$ norm. These bounds are used in the analysis of the singular subspace recovery in Lemma 32, and therefore in the proofs of Theorems 3 and 5.

**Lemma 29.** *Let $Y \in \mathbb{R}^{n\times n}$ be a matrix with independent entries $Y_{i,j} \sim T^{-1}\mathrm{Poisson}(TM_{ij})$, $i,j \in [n]$, for an arbitrary integer $T > 0$. Let $0 \leq \delta \leq 1$. Then, for any $1 \leq l \leq n$ and any matrix $A \in \mathbb{R}^{n\times p}$, with $p \leq n$, and independent of $Y_{l,:}$ we have, if $T\|M_{l,:}\|_\infty \leq 1$,*

$$
\|(Y_{l,:} - M_{l,:})A\| \lesssim \|A\|_F \frac{\sqrt{\|M_{l,:}\|_\infty \log\left(\frac{ne}{\delta}\right)}}{\sqrt{T}} + \|A\|_{2\to\infty}\frac{\log^2\left(\frac{ne}{\delta}\right)}{T\log(1 + (T\|M_{l,:}\|_\infty)^{-1} \wedge n\delta^{-1}))}
$$

*else if $T\|M_{l,:}\|_\infty > 1$,*

$$
\|(Y_{l,:} - M_{l,:})A\| \lesssim \|A\|_F \frac{\sqrt{\|M_{l,:}\|_\infty \log\left(\frac{ne}{\delta}\right)}}{\sqrt{T}} + \|A\|_{2\to\infty}\frac{\sqrt{\|M_{l,:}\|_\infty}}{\sqrt{T}}\log\left(T\|M_{l,:}\|_\infty\frac{ne}{\delta}\right)\log\left(\frac{ne}{\delta}\right)
$$

*with probability at least $1 - \delta/n$.*

*Proof of Lemma 29.* The lemma is an application of the truncated matrix Bernstein theorem i.e. Theorem 23. In this theorem, $T$ corresponds to $n$ in Lemma 29, $n$ in Theorem 23 corresponds to 1 in Lemma 29, and $m$ in Theorem 23 corresponds to $n$ in Lemma 29. First note that for any $l$, we have $(Y_{l,:} - M_{l,:})A = \sum_{i=1}^{n}(Y_{l,i} - M_{l,i})A_{i,:}$. Moreover, since each of these $n$ summands are independent, zero-mean random vectors, we can identify $Z_i$'s from Theorem 23 with $(Y_{l,i} - M_{l,i})A_{i,:} \in \mathbb{R}^{1\times n}$ for $i \in [n]$. To apply Theorem 23, we need to verify its assumptions. This is done below.

**Step 1: Showing *(i)* in** (20) First, recall Bennett's concentration inequality from Lemma 24, which in our case implies that for any $i,j \in [n]$:

$$\mathbb{P}(|Y_{i,j} - M_{i,j}| \geq tM_{i,j}) \leq 2\exp\left(-h(t)TM_{i,j}\right). \tag{29}$$

Note that $\|Z_i\|$ in Theorem 23 in our case corresponds to:

$$\|(Y_{l,i} - M_{l,i})A_{i,:}\| = |Y_{l,i} - M_{l,i}|\|A_{i,:}\| \leq |Y_{l,i} - M_{l,i}|\|A\|_{2\to\infty}.$$

We consider two different cases:

1. $T\|M_{l,:}\|_\infty \leq 1$: We let $\beta_1 = 4T^{-1}\|A\|_{2\to\infty}\log^{-1}(1 + (T\|M_{l,:}\|_\infty)^{-1} \wedge n\delta^{-1})\log(\frac{ne}{\delta})$ and note that $h(t) \geq \frac{1}{2}t\log t$ for $t \geq 1$. Thus, from Equation (29), we have:

$$\mathbb{P}\left(|Y_{l,i} - M_{l,i}| \geq \frac{\beta_1}{\|A\|_{2\to\infty}}\right) \leq 2\exp\Bigg(-2\log(\frac{ne}{\delta})\log^{-1}(1 + (T\|M_{l,:}\|_\infty)^{-1} \wedge n\delta^{-1})$$
$$\cdot \log\left(\frac{4\log(\frac{ne}{\delta})}{T\|M_{l,:}\|_\infty\log(1 + (T\|M_{l,:}\|_\infty)^{-1} \wedge n\delta^{-1})}\right)\Bigg) \leq \frac{\delta}{2n^2}.$$

where, in the second inequality, we show using simple algebra that $\log^{-1}(1 + (T\|M_{l,:}\|_\infty)^{-1} \wedge n\delta^{-1})\log\left(\frac{4\log(\frac{ne}{\delta})}{T\|M_{l,:}\|_\infty\log(1+(T\|M_{l,:}\|_\infty)^{-1}\wedge n\delta^{-1})}\right) \geq 1$ for $\delta \leq 1$ and $T\|M_{l,:}\|_\infty \leq 1$.

2. $T\|M_{l,:}\|_\infty > 1$: Here we define $\beta_2 := 4\|A\|_{2\to\infty}\sqrt{T^{-1}\|M_{l,:}\|_\infty}\log\left(T\|M_{l,:}\|_\infty\frac{ne}{\delta}\right)$. Then, according to Equation (29) and the approximation $h(t) \geq \min\{t^2/4, t\}$ for $t \geq 0$, we have:

$$\mathbb{P}\left(|Y_{l,i} - M_{l,i}| \geq \frac{\beta_2}{\|A\|_{2\to\infty}}\right)$$
$$\leq 2\exp\left(-4\min\left\{\log^2(T\|M_{l,:}\|_\infty\frac{ne}{\delta}), \sqrt{T\|M_{l,:}\|_\infty}\log(T\|M_{l,:}\|_\infty\frac{ne}{\delta})\right\}\right)$$
$$\leq 2\exp\left(-4\log(T\|M_{l,:}\|_\infty\frac{ne}{\delta})\right) \leq \frac{1}{2T\|M_{l,:}\|_\infty}\frac{\delta}{n^2}.$$

where, in the second inequality, we used that $\delta \leq 1$ and $T\|M_{l,:}\|_\infty > 1$. Finally, we define $\beta = \beta_1\mathbb{1}_{\{T\|M_{l,:}\|_\infty\leq1\}} + \beta_2\mathbb{1}_{\{T\|M_{l,:}\|_\infty>1\}}$ and $p = \frac{\delta}{2n}\mathbb{1}_{\{T\|M_{l,:}\|_\infty\leq1\}} + \frac{1}{2T\|M_{l,:}\|_\infty}\frac{\delta}{n}\mathbb{1}_{\{T\|M_{l,:}\|_\infty>1\}}$ (since we took union bound over $i \in [n]$).

**Step 2: Showing *(ii)* in** (20) In our case the l.h.s. corresponds to $\|\mathbb{E}[(Y_{l,i} - M_{l,i})A_{i,:}\mathbb{1}_{\{\|(Y_{l,i}-M_{l,i})A_{i,:}\|>\beta\}}]\| = \|\mathbb{E}[(Y_{l,i}-M_{l,i})A_{i,:}\mathbb{1}_{\{\|(Y_{l,i}-M_{l,i})A_{i,:}\|\leq\beta\}}]\|$ which can be upper bounded by $\|A\|_{2\to\infty}|\mathbb{E}[(Y_{l,i} - M_{l,i})\mathbb{1}_{\{|Y_{l,i}-M_{l,i}|\leq\frac{\beta}{\|A_{i,:}\|}\}}]|$. For some integers $\kappa_{\min}, \kappa_{\max}$, let $Y_{l,i} \in \frac{1}{T}[\kappa_{\min}, \kappa_{\max}]$ be interval of $Y_{l,i}$ for which indicator $\mathbb{1}_{\{|Y_{l,i}-M_{l,i}|\leq\frac{\beta}{\|A_{i,:}\|}\}}$ is active and note that this is a superset of interval for which $\mathbb{1}_{\{|Y_{l,i}-M_{l,i}|\leq\frac{\beta}{\|A\|_{2\to\infty}}\}}$ is active. Then from the definition of Poisson random variables and the bounds derived previously, we obtain:

$$\left|\mathbb{E}[(Y_{l,i} - M_{l,i})\mathbb{1}_{\{|Y_{l,i}-M_{l,i}|\leq\frac{\beta}{\|A_{i,:}\|}\}}]\right| = \frac{1}{T}\left|\sum_{k=\kappa_{\min}}^{\kappa_{\max}}(k - TM_{l,i})\frac{\exp(-TM_{l,i})(TM_{l,i})^k}{k!}\right|$$
$$= M_{l,i}\left|\sum_{k=\kappa_{\min}-1}^{\kappa_{\max}-1}\frac{\exp(-TM_{l,i})(TM_{l,i})^k}{k!} - \sum_{k=\kappa_{\min}}^{\kappa_{\max}}\frac{\exp(-TM_{l,i})(TM_{l,i})^k}{k!}\right|$$
$$\leq M_{l,i}(\mathbb{P}(TY_{l,i} = \kappa_{\min} - 1) + \mathbb{P}(TY_{l,i} = \kappa_{\max})) \leq \frac{2\delta}{Tn^2}\min\{T\|M_{l,:}\|_\infty, 1\},$$

where we assumed that $\kappa_{\min} \geq 1$, otherwise we keep just the second probability term above. Thus, using previous two inequalities, we have:

$$\|\mathbb{E}[(Y_{l,i} - M_{l,i})A_{i,:}\mathbb{1}_{\{\|(Y_{l,i}-M_{l,i})A_{i,:}\|>\beta\}}]\| \leq \|A\|_{2\to\infty}\frac{2\delta}{Tn^2}\min\{T\|M_{l,:}\|_\infty, 1\}$$

**Step 3: Showing** *(iii)* **in** (21) Using our definition $Z_i = (Y_{l,i} - M_{l,i})A_{i,:} \in \mathbb{R}^{1\times n}$, we have that

$$Z_i Z_i^\top = (Y_{l,i} - M_{l,i})^2\|A_{i,:}\|^2,$$
$$Z_i^\top Z_i = (Y_{l,i} - M_{l,i})^2 A_{i,:}^\top A_{i,:}.$$

Since $A$ and $Y_{l,:}$ are independent, we have:

$$\|\sum_{i=1}^n \mathbb{E}[Z_i Z_i^\top]\| = \sum_{i=1}^n \mathbb{E}[Z_i Z_i^\top] = \sum_{i=1}^n \|A_{i,:}\|^2\mathbb{E}(Y_{l,i} - M_{l,i})^2 \leq \|A\|_F^2 \max_i \mathbb{E}(Y_{l,i} - M_{l,i})^2$$

and

$$\|\sum_{i=1}^n \mathbb{E}[Z_i^\top Z_i]\| = \|\sum_{i=1}^n \mathbb{E}(Y_{l,i} - M_{l,i})^2 A_{i,:}^\top A_{i,:}\| \leq \sum_{i=1}^n \mathbb{E}(Y_{l,i} - M_{l,i})^2\|A_{i,:}^\top A_{i,:}\|$$
$$\leq \|A\|_F^2 \max_i \mathbb{E}(Y_{l,i} - M_{l,i})^2.$$

Now note that for $Y_{l,i} \sim T^{-1}\text{Poisson}(TM_{l,i})$, $\text{Var}(Y_{l,i}) = \mathbb{E}(Y_{l,i} - M_{l,i})^2 = T^{-1}M_{l,i}$. Thus, by setting $v = T^{-1}\|A\|_F^2\|M_{l,:}\|_\infty$, we get *(iii)*.

Plugging in all obtained quantities into Equation (22) finishes proof of the lemma. $\qquad\square$

# E  Singular subspace recovery via the leave-one-out argument

In this section, we present Lemma 30 and Lemma 32 providing sharp guarantees for the singular subspace recovery in two-to-infinity norm. Obtaining such guarantees is not trivial and requires the use of a rather technical analysis, namely the leave-one-out technique [2, 14]. However, such technique heavily relies on independence between entries of the observed random matrix. We use the Poisson approximation argument to address this, which in turn requires to reproduce the leave-one-out analysis under a different random matrix observation models (see (15) and (19)).

We wish to highlight that Farias et al. [23], like us, have also used the leave-one-out argument to obtain entry-wise guarantees for matrix estimation with sub-exponential noise. In our case, we use this argument as a sub-step of our analysis after performing the Poisson approximation. However, we believe that, our final results are richer, more precise and actually needed for our RL applications. Indeed, we are able to obtain guarantees in the norms $\|\cdot\|_{2\to\infty}$ and $\|\cdot\|_{1\to\infty}$ (these are not provided in [23]). Moreover, the entry-wise guarantees in [23] are only expressed in terms of the matrix dimensions $m$ and $n$. Our guarantees on the other hand exhibit dependencies on the dimensions $m, n$, the number of observation $T$ and the confidence level $\delta$. Having guarantees with an explicit dependence for all $T \geq 1$ and $\delta \in (0, 1)$ is crucial in the design of our algorithm for low-rank bandits.

## E.1  Subspace recovery for reward matrices

**Lemma 30.** *Let $\delta \in (0, 1)$. Define:*

$$\mathcal{B} = \sqrt{\frac{nm}{T}} \left( \sqrt{(n+m) \log\left(\frac{e(n+m)T}{\delta}\right)} + \log^{3/2}\left(\frac{e(n+m)T}{\delta}\right) \right).$$

*For all $T \geq c(\mu^4 \kappa^2 r^2 + 1)(m+n) \log^3\left(e^2(m+n)T/\delta\right)$, the event*

$$\max(\|U - \widehat{U}(\widehat{U}^\top U)\|, \|V - \widehat{V}(\widehat{V}^\top V)\|) \leq C \frac{\|M\|\|M\|_\infty}{\sigma_r(M)^2} \max(\|V\|_{2\to\infty} \|U\|_{2\to\infty})\mathcal{B}$$

*holds with probability at least $1 - \delta$, for some universal constants $c, C > 0$.*

*Proof of Lemma 30.* The proof follows similar steps as that of Theorem 4.2 in [14], which is based on the leave-one-out analysis.

**Step 1: Dilation trick.** In order to apply the leave-one-out analysis, we first use a dilation trick [58] to reduce the problem to that of symmetric matrices. Define:

$$S = \begin{bmatrix} 0 & M \\ M^\top & 0 \end{bmatrix}$$

and note that for matrix $M$ with SVD $M = U\Sigma V^\top$, we have:

$$S = \frac{1}{\sqrt{2}} \begin{bmatrix} U & U \\ V & -V \end{bmatrix} \begin{bmatrix} \Sigma & 0 \\ 0 & -\Sigma \end{bmatrix} \frac{1}{\sqrt{2}} \begin{bmatrix} U & U \\ V & -V \end{bmatrix}^\top := QDQ^\top.$$

We define, in a similar way, $\widetilde{S}$ using $\widetilde{M}$, and let $\widehat{Q} \in \mathbb{R}^{(n+m)\times 2r}$ be the matrix of eigenvectors of the best $2r$-rank approximation of $\widetilde{S}$. Note that:

$$\|Q - \widehat{Q}(\widehat{Q}^\top Q)\|_{2\to\infty} = \max\left\{ \|U - \widehat{U}(\widehat{U}^\top U)\|_{2\to\infty}, \|V - \widehat{V}(\widehat{V}^\top V)\|_{2\to\infty} \right\}. \tag{30}$$

To keep the notation simple, we will define $W_{\widehat{Q}} = \widehat{Q}^\top Q$. Further note that

$$\|\widetilde{S} - S\| = \|\widetilde{M} - M\|, \qquad \sigma_1(S) = \sigma_1(M), \qquad \text{and} \qquad \sigma_{2r}(S) = \sigma_r(M). \tag{31}$$

We start the analysis under the model (15) and assume that $\widetilde{M}$ has independent entries with compound Poisson distributions. We will eventually invoke the Poisson approximation argument via Lemma 20 to deduce the final result.

**Step 2: Error decomposition.** We apply the decomposition in Lemma 33 to obtain:

$$\|Q - \widehat{Q}W_{\widehat{Q}}\|_{2\to\infty} \leq \frac{1}{\sigma_{2r}(S)} \left( \frac{4\|\widetilde{S}Q\|_{2\to\infty}\|E\|}{\sigma_{2r}(S)} + \|EQ\|_{2\to\infty} + 2\|\widetilde{S}(Q - \widehat{Q}W_{\widehat{Q}})\|_{2\to\infty} \right),$$

where we set $E = \widetilde{S} - S$. We observe that when $\|E\| \leq \sigma_{2r}(S)/2$, then

$$\|Q - \widehat{Q}W_{\widehat{Q}}\|_{2\to\infty} \leq \frac{1}{\sigma_{2r}(S)} \left( \frac{4\|SQ\|_{2\to\infty}\|E\|}{\sigma_{2r}(S)} + 3\|EQ\|_{2\to\infty} + 2\|\widetilde{S}(Q - \widehat{Q}W_{\widehat{Q}})\|_{2\to\infty} \right). \tag{32}$$

Furthermore, we also have

$$
\begin{aligned}
\|\widetilde{S}(Q - \widehat{Q}W_{\widehat{Q}})\|_{2\to\infty} &\leq \|E(Q - \widehat{Q}W_{\widehat{Q}})\|_{2\to\infty} + \|S(Q - \widehat{Q}W_{\widehat{Q}})\|_{2\to\infty} \\
&\leq \|E(Q - \widehat{Q}W_{\widehat{Q}})\|_{2\to\infty} + \|SQ\|_{2\to\infty}\|\sin(Q,\widehat{Q})\|^2 \\
&\leq \|E(Q - \widehat{Q}W_{\widehat{Q}})\|_{2\to\infty} + \frac{\|SQ\|_{2\to\infty}\|E\|^2}{\sigma_{2r}(S)^2} \\
&\leq \|E(Q - \widehat{Q}W_{\widehat{Q}})\|_{2\to\infty} + \frac{\|SQ\|_{2\to\infty}\|E\|}{2\sigma_{2r}(S)},
\end{aligned}
$$

where the first inequality follows from the triangular inequality, the second inequality follows by the relation between the two-to-infinity norm and the sin theorem (see e.g., [12]). The third inequality follows from Davis-Kahan's theorem. The fourth inequality follows under the condition $\|E\| \leq \sigma_{2r}(S)/2$. We finally obtain

$$\|Q - \widehat{Q}W_{\widehat{Q}}\|_{2\to\infty} \leq \frac{1}{\sigma_{2r}(S)} \left( \frac{5\|SQ\|_{2\to\infty}\|E\|}{\sigma_{2r}(S)} + 3\|EQ\|_{2\to\infty} + 2\|E(Q - \widehat{Q}W_{\widehat{Q}})\|_{2\to\infty} \right). \tag{33}$$

Note that in the above inequality, we can control $\|E\|$ using Proposition 26 and $\|EQ\|_{2\to\infty}$ using Proposition 27. However, the term $\|E(Q - \widehat{Q}W_{\widehat{Q}})\|_{2\to\infty}$ is not easy to control because $E$ and $(Q - \widehat{Q}W_{\widehat{Q}})$ are dependent on each other in a non-trivial way. To control this term, we use the leave-one-out analysis.

**Step 3: Leave-one-out analysis.** We define a matrix $\widetilde{S}^{(\ell)} \in \mathbb{R}^{(n+m)\times(n+m)}$ as follows:

$$\widetilde{S}^{(\ell)}_{i,j} = \begin{cases} \widetilde{S}_{i,j}, & \text{if } i \neq \ell \text{ or } j \neq \ell \\ S_{i,j}, & \text{otherwise} \end{cases}$$

Then define $\widehat{Q}^{(\ell)} \in \mathbb{R}^{n\times 2r}$ as a matrix of eigenvectors corresponding to the $2r$ greatest (in absolute value) eigenvalues of matrix $\widetilde{S}^{(\ell)}$. Define $W_{\widetilde{U}^{(\ell)}}$ accordingly. We have

$$\|E(Q - \widehat{Q}W_{\widehat{Q}})\|_{2\to\infty} \leq \max_{\ell\in[n+m]} \|E_{\ell,:}(Q - \widehat{Q}^{(\ell)}W_{\widehat{Q}^{(\ell)}})\|_2 + \|E\|_2\|\widehat{Q}W_{\widehat{Q}} - \widehat{Q}^{(\ell)}W_{\widehat{Q}^{(\ell)}}\|_F.$$

We have by Proposition 26 that

$$\mathbb{P}\left(\|E\| \lesssim \|M\|_\infty \mathcal{G}\right) \geq 1 - \delta$$

provided that

$$\textbf{(C1)} \qquad T \geq c_1 \frac{mn}{m+n} \log^3\left(\frac{e(m+n)}{\delta}\right)$$

and where we define

$$\mathcal{G} = \sqrt{\frac{mn}{T}} \left( \sqrt{(m+n)\log\left(\frac{e(m+n)}{\delta}\right)} + \log^{3/2}\left(\frac{e(m+n)}{\delta}\right) \right).$$

Let us now introduce the event $\mathcal{E}_1$ as follows

$$\mathcal{E}_1 = \{\|E\| \leq \|M\|_\infty \mathcal{G}\}.$$

Note that if the following condition holds

**(C2)** $$T \geq c_2(\mu\kappa r)^2 \left( (m+n)\log\left(\frac{e(m+n)}{\delta}\right) + \log^3\left(\frac{e(m+n)}{\delta}\right) \right)$$

for $c_2$ large enough then $16\|E\| \leq \sigma_r(M)$. Hence, under the event $\mathcal{E}_1$, using Lemma 31, we have

$$\|\widehat{Q}W_{\widehat{Q}} - \widehat{Q}^{(\ell)}W_{\widehat{Q}^{(\ell)}}\|_F \leq \frac{16\|E_{\ell,:}\widehat{Q}^{(\ell)}W_{\widehat{Q}^{(\ell)}}\|_2 + 16\|E\|\|\widehat{Q}W_{\widehat{Q}}\|_{2\to\infty}}{\sigma_{2r}(M)},$$

which further gives by triangular inequality

$$\|\widehat{Q}W_{\widehat{Q}} - \widehat{Q}^{(\ell)}W_{\widehat{Q}^{(\ell)}}\|_F \leq \frac{16\|E_{\ell,:}(Q - \widehat{Q}^{(\ell)}W_{\widehat{Q}^{(\ell)}})\|_2}{\sigma_r(M)}$$
$$+ \frac{16\left(\|E_{\ell,:}Q\|_2 + \|E\|\|Q - \widehat{Q}W_{\widehat{Q}}\|_{2\to\infty} + \|E\|\|Q\|_{2\to\infty}\right)}{\sigma_r(M)}.$$

Now, by Proposition 27,

$$\mathbb{P}\left( \|E_{\ell,:}(Q - Q^{(\ell)}W_{\widehat{Q}^{(\ell)}})\|_2 \lesssim \|M\|_\infty\|Q - Q^{(\ell)}W_{\widehat{Q}^{(\ell)}}\|_{2\to\infty}\mathcal{G} \right) \geq 1 - \delta$$

as long as the same condition **(C1)** holds with $c_1$ large enough. So let us introduce the event

$$\mathcal{E}_2 = \left\{ \|E_{\ell,:}(Q - Q^{(\ell)}W_{\widehat{Q}^{(\ell)}})\|_2 \lesssim \|M\|_\infty\|Q - Q^{(\ell)}W_{\widehat{Q}^{(\ell)}}\|_{2\to\infty}\mathcal{G} \right\}.$$

We further upper bound under the event $\mathcal{E}_1 \cap \mathcal{E}_2$,

$$\|E_{\ell,:}(Q - \widehat{Q}^{(\ell)}W_{\widehat{Q}^{(\ell)}})\|_2 \lesssim \|M\|_\infty \left( \|Q - \widehat{Q}W_{\widehat{Q}}\|_{2\to\infty} + \|\widehat{Q}W_{\widehat{Q}} - \widehat{Q}^{(\ell)}W_{\widehat{Q}^{(\ell)}}\|_F \right)\mathcal{G}.$$

Note that, under the condition **(C2)** with $c_2$ large enough, we can also obtain

$$\frac{16\|M\|_\infty}{\sigma_r(M)}\mathcal{G} \leq \frac{1}{2}$$

which entails that

$$\|\widehat{Q}W_{\widehat{Q}} - \widehat{Q}^{(\ell)}W_{\widehat{Q}^{(\ell)}}\|_F \leq \frac{32\|M\|_\infty\|Q - \widehat{Q}W_{\widehat{Q}}\|_{2\to\infty}}{\sigma_r(M)}\mathcal{G}$$
$$+ \frac{32(\|E_{\ell,:}Q\| + \|E\|\|Q - \widehat{Q}W_{\widehat{Q}}\|_{2\to\infty} + \|E\|\|Q\|_{2\to\infty})}{\sigma_r(M)}.$$

To simplify the notation, let us define the three errors as

$$x = \|Q - \widehat{Q}W_{\widehat{Q}}\|_{2\to\infty},$$
$$y = \|EQ\|_{2\to\infty} \geq \|E_{\ell,:}Q\|_2,$$
$$z = \|E\|\|Q\|_{2\to\infty}.$$

We have

$$\|\widehat{Q}W_{\widehat{Q}} - \widehat{Q}^{(\ell)}W_{\widehat{Q}^{(\ell)}}\|_F \lesssim \left( \frac{\|M\|_\infty\mathcal{G}}{\sigma_r(M)} + \frac{\|E\|}{\sigma_r(M)} \right)x + \frac{1}{\sigma_r(M)}(y+z).$$

By plugging the above in the previous inequality, we get

$$\|E_{\ell,:}(Q - \widehat{Q}^{(\ell)}W_{\widehat{Q}^{(\ell)}})\|_2 \lesssim \|M\|_\infty\mathcal{G}\left( \left(1 + \frac{\|M\|_\infty\mathcal{G}}{\sigma_r(M)} + \frac{\|E\|}{\sigma_r(M)} \right)x + \frac{1}{\sigma_r(M)}(y+z) \right)$$

which entails finally

$$\|E(Q - \widehat{Q}W_{\widehat{Q}})\|_{2\to\infty} \lesssim \left( \frac{\|E\|}{\sigma_r(M)} + \frac{\mathcal{G}\|M\|_\infty}{\sigma_1(M)} \right)(y+z)$$
$$+ (\|E\| + \mathcal{G}\|M\|_\infty)\left( 1 + \frac{\mathcal{G}\|M\|_\infty}{\sigma_r(M)} + \frac{\|\mathcal{E}\|}{\sigma_r(M)} \right)x. \qquad (34)$$

**Step 4: Putting everything together.** Combining the inequalities (33) and (34) gives

$$x \leq C_1 \left( \frac{\|E\|}{\sigma_r(M)} + \frac{\|M\|_\infty \mathcal{G}}{\sigma_r(M)} \right) \left( 1 + \frac{\|M\|_\infty \mathcal{G}}{\sigma_r(M)} + \frac{\|E\|}{\sigma_r(M)} \right) x$$
$$+ \frac{C_2}{\sigma_r(M)} \left( 1 + \frac{\|E\|}{\sigma_r(M)} + \frac{\mathcal{G}\|M\|_\infty}{\sigma_r(M)} \right) y$$
$$+ \frac{C_3}{\sigma_r(M)} \left( \frac{\|M\|}{\sigma_r(M)} + \frac{\|E\|}{\sigma_r(M)} + \frac{\mathcal{G}\|M\|_\infty}{\sigma_r(M)} \right) z.$$

Under the events $\mathcal{E}_1$ and $\mathcal{E}_2$ and provided that the conditions **(C1)** and **(C2)** hold, for $c_1$ and $c_2$ are large enough, we have

$$C_1 \left( \frac{\|E\|}{\sigma_r(M)} + \frac{\|M\|_\infty \mathcal{G}}{\sigma_r(M)} \right) \left( 1 + \frac{\|M\|_\infty \mathcal{G}}{\sigma_r(M)} + \frac{\|E\|}{\sigma_r(M)} \right) \leq \frac{1}{2},$$
$$\left( 1 + \frac{\|E\|}{\sigma_r(M)} + \frac{\mathcal{G}\|M\|}{\sigma_r(M)} \right) \leq 3,$$
$$\left( \frac{\|M\|}{\sigma_r(M)} + \frac{\|E\|}{\sigma_r(M)} + \frac{\mathcal{G}\|M\|_\infty}{\sigma_r(M)} \right) \leq \left( \frac{\|M\|}{\sigma_r(M)} + 2 \right).$$

Thus, we obtain

$$x \leq \frac{1}{\sigma_r(M)} \left( y + \frac{\|M\|}{\sigma_r(M)} z \right).$$

We note that, under a similar conditions as before , we also have by Proposition 26 and Proposition 27 that

$$y \lesssim \|M\|_\infty \|Q\|_{2\to\infty} \mathcal{G}$$
$$z \lesssim \|M\|_\infty \|Q\|_{2\to\infty} \mathcal{G}$$

with probability at least $1 - \delta$. Thus, we conclude after further simplifications that for some $C > 0$ large enough, we have

$$\mathbb{P} \left( \|Q - \widehat{Q} W_{\widehat{Q}}\|_{2\to\infty} \leq C \frac{\|M\|\|M\|_\infty}{\sigma_r(M)^2} \|Q\|_{2\to\infty} \mathcal{G} \right) \geq 1 - \delta$$

provided

$$T \geq c(\mu^4 \kappa^2 r^2 + 1)(m + n) \log^3 \left( \frac{e(m+n)}{\delta} \right),$$

with

$$\mathcal{G}(n, m, T, \delta) = \sqrt{\frac{nm}{T}} \left( \sqrt{(n+m) \log \left( \frac{e(n+m)}{\delta} \right)} + \log^{3/2} \left( \frac{e(n+m)}{\delta} \right) \right).$$

**Step 5: Poisson approximation.** To conclude, we now invoke Lemma 20 which entails that under the true model (14), we have

$$\mathbb{P} \left( \|Q - \widehat{Q} W_{\widehat{Q}}\|_{2\to\infty} > C \frac{\|M\|\|M\|_\infty}{\sigma_r(M)^2} \|Q\|_{2\to\infty} \mathcal{G}(n, m, T, \delta) \right) \leq e\sqrt{T}\delta$$

provided $T \geq c(\mu^4 \kappa^2 r^2 + 1)(m + n) \log^3 (e(m+n)/\delta)$. By re-parametrizing with $\delta' = e\sqrt{T}\delta$, we obtain

$$\mathbb{P} \left( \|Q - \widehat{Q} W_{\widehat{Q}}\|_{2\to\infty} > C \frac{\|M\|\|M\|_\infty}{\sigma_r(M)^2} \|Q\|_{2\to\infty} \mathcal{G}(n, m, T, \delta'/e\sqrt{T}) \right) \leq \delta',$$

again provided that $T \geq c(\mu^4 \kappa^2 r^2 + 1)(m+n) \log^3 \left( e^2(m+n)\sqrt{T}/\delta' \right)$. Recalling that $\|Q\|_{2\to\infty} = \max(\|V\|_{2\to\infty}, \|U\|_{2\to\infty})$, we immediately obtain the final result. $\qquad\square$

**Lemma 31.** *Under the notation used in the proof of Lemma 30, provided the condition $\|E\| \leq \sigma_{2r}(S)/16$, the following inequality holds:*

$$\|\widehat{Q}W_{\widehat{Q}} - \widehat{Q}^{(\ell)}W_{\widehat{Q}^{(\ell)}}\|_F \leq \frac{16\|E_{\ell,:}\widehat{Q}^{(\ell)}W_{\widehat{Q}^{(\ell)}}\|_2 + 16\|E\|\|\widehat{Q}W_{\widehat{Q}}\|_{2\to\infty}}{\sigma_{2r}(S)}$$

*Proof of Lemma 31.* We have

$$\|\widehat{Q}W_{\widehat{Q}} - \widehat{Q}^{(\ell)}W_{\widehat{Q}^{(\ell)}}\|_F \leq \|\widehat{Q}\widehat{Q}^\top - \widehat{Q}^{(\ell)}(\widehat{Q}^{(\ell)})^\top\|_F \|Q\| \leq \frac{2\|(\widetilde{S} - \widetilde{S}^{(\ell)})\widehat{Q}^{(\ell)}\|_F}{|\sigma_{2r}(\widetilde{S}^{(\ell)}) - \sigma_{2r+1}(\widetilde{S}^{(\ell)})|}$$

where the first inequality follows the elementary fact that $\|AB\|_F \leq \|A\|_F\|B\|$, and the second inequality follows by Davis-Kahan. Now, by Weyl's inequality, we have for all $k \in [n + m]$, $|\sigma_k(\widetilde{S}^{(\ell)}) - \sigma_k(S)| \leq \|E^{(\ell)}\| \leq \|E\|$, where the error matrix $E^{(\ell)} = \widetilde{S}^\ell - S$, and more precisely is defined as follows:

$$E_{i,j}^{(\ell)} = \begin{cases} E_{i,j} & \text{if } i \neq \ell \text{ or } j \neq \ell, \\ 0 & \text{otherwise.} \end{cases}$$

The crude inequality $\|E^{(\ell)}\| \leq \|E\|$ follows from the fact that $\|E^{(\ell)}\|$ is equal to the operator norm of a submatrix of $E$ which will always be smaller than $\|E\|$. Therefore, under the condition that $\|E\| \leq \sigma_{2r}(S)/4$, we have $|\sigma_{2r}(\widetilde{S}^{(\ell)}) - \sigma_{2r+1}(\widetilde{S}^{(\ell)})| \geq \sigma_{2r}(S)/2$. In summary, we obtain that

$$\|\widehat{Q}W_{\widehat{Q}} - \widehat{Q}^{(\ell)}W_{\widehat{Q}^{(\ell)}}\|_F \leq \frac{4\|(\widetilde{S} - \widetilde{S}^{(\ell)})\widehat{Q}^{(\ell)}\|_F}{\sigma_{2r}(S)}.$$

Now, we further have by triangular inequality and by definition of $\widetilde{S}^{(\ell)}$:

$$\begin{aligned}\|(\widetilde{S} - \widetilde{S}^{(\ell)})\widehat{Q}^{(\ell)}\|_F &= \|(e_\ell E_{\ell,:} + (E_{:,\ell} - E_{\ell,\ell}e_\ell)e_\ell^\top)\widehat{Q}^{(\ell)}\|_F \\ &\leq \|E_{\ell,:}\widehat{Q}^{(\ell)}\|_2 + \|E_{:,\ell} - E_{\ell,\ell}e_\ell\|_2\|\widehat{Q}^{(\ell)}\|_{2\to\infty} \\ &\leq \|E_{\ell,:}\widehat{Q}^{(\ell)}\|_2 + \|E\|_2\|\widehat{Q}^{(\ell)}\|_{2\to\infty} \\ &\leq \|E_{\ell,:}\widehat{Q}^{(\ell)}\|_2 + 2\|E\|_2\|\widehat{Q}^{(\ell)}(\widehat{Q}^{(\ell)})^\top Q\|_{2\to\infty}\end{aligned}$$

where the last inequality follows under the condition that $\|E\| \leq 2\sigma_{2r}(S)$. Indeed, we have under such condition that $\|\widehat{Q}^{(\ell)}\|_{2\to\infty} = \|\widehat{Q}^{(\ell)}(\widehat{Q}^{(\ell)})^\top Q\|_{2\to\infty} + \|\widehat{Q}^{(\ell)}(\text{sgn}((\widehat{Q}^{(\ell)})^\top Q^\top) - (\widehat{Q}^{(\ell)})^\top Q)\|_{2\to\infty}$, and by Davis-Kahan's inequality $\|\text{sgn}((\widehat{Q}^{(\ell)})^\top Q^\top) - (\widehat{Q}^{(\ell)})^\top Q\| \leq \frac{2\|E^{(\ell)}\|^2}{(\sigma_{2r}(S))^2} \leq \frac{2\|E\|^2}{(\sigma_{2r}(S))^2} \leq \frac{1}{2}$. Similarly, we also have $\|E_{\ell,:}\widehat{Q}^{(\ell)}\|_2 \leq 2\|E_{\ell,:}\widehat{Q}^{(\ell)}(\widehat{Q}^{(\ell)})^\top Q\|_2$.

Hence, we obtain:

$$\|(\widetilde{S} - \widetilde{S}^{(\ell)})\widehat{Q}^{(\ell)}\|_F \leq 2\|E_{\ell,:}\widehat{Q}^{(\ell)}W_{\widehat{Q}^{(\ell)}}\|_2 + 2\|E\|(\|\widehat{Q}W_{\widehat{Q}}\|_{2\to\infty} + \|\widehat{Q}W_{\widehat{Q}} - \widehat{Q}^{(\ell)}W_{\widehat{Q}^{(\ell)}}\|_{2\to\infty})$$

Which entails under the condition that $\|E\| \leq \sigma_{2r}(S)/16$ that

$$\|\widehat{Q}W_{\widehat{Q}} - \widehat{Q}^{(\ell)}W_{\widehat{Q}^{(\ell)}}\|_F \leq \frac{8\|E_{\ell,:}\widehat{Q}^{(\ell)}W_{\widehat{Q}^{(\ell)}}\|_2 + 8\|E\|\|\widehat{Q}W_{\widehat{Q}}\|_{2\to\infty}}{\sigma_{2r}(S)} + \frac{\|\widehat{Q}W_{\widehat{Q}} - \widehat{Q}^{(\ell)}W_{\widehat{Q}^{(\ell)}}\|_{2\to\infty}}{2}$$

After rearranging, we obtain

$$\|\widehat{Q}W_{\widehat{Q}} - \widehat{Q}^{(\ell)}W_{\widehat{Q}^{(\ell)}}\|_F \leq \frac{16\|E_{\ell,:}\widehat{Q}^{(\ell)}W_{\widehat{Q}^{(\ell)}}\|_2 + 16\|E\|\|\widehat{Q}W_{\widehat{Q}}\|_{2\to\infty}}{\sigma_{2r}(S)}$$

$\square$

### E.2 Subspace recovery for transition matrices

**Lemma 32.** *Let $Y \in \mathbb{R}^{n\times n}$ be a matrix of independent Poisson entries with $Y_{i,j} \sim \frac{1}{T}\text{Poisson}(TM_{i,j})$, and let $\widehat{U}, \widehat{V}$ be the matrices of left and right singular vectors of best $r$-rank*

*approximation of $Y$. Let $g_\delta$ be the function defined in (9). Conditioned on the events where $\|Y - M\| \leq c_1 \sigma_r(M), g_\delta(TM) \log(ne/\delta) \leq c_2 T\sigma_r(M), \sqrt{\|M\|_\infty \log(ne/\delta)} \leq c_3\sqrt{T}\sigma_r(M)$ for some sufficiently small universal constants $c_1, c_2, c_3 > 0$, we have, with probability at least $1 - \delta$,*

$$\max\left\{\|U - \widehat{U}(\widehat{U}^\top U)\|_{2\to\infty}, \|V - \widehat{V}(\widehat{V}^\top V)\|_{2\to\infty}\right\}$$

$$\lesssim \frac{1}{\sigma_r(M)}\left[\mu\sqrt{\frac{r}{n}}\left(\frac{\sigma_1(M)}{\sigma_r(M)}\|Y - M\| + \frac{1}{T}g_\delta(TM)\log\left(\frac{ne}{\delta}\right)\right) + \sqrt{\frac{r\|M\|_\infty}{T}\log\left(\frac{ne}{\delta}\right)}\right].$$

*Proof.* The proof follows similar steps as the proof of Theorem 4.2 in [14]. In order to apply the leave-one-out technique, we first repeat the symmetric dilation trick as in Step 1 of proof of Lemma 30. We define

$$S = \begin{bmatrix} 0 & M \\ M^\top & 0 \end{bmatrix} \tag{35}$$

and note that for matrix $M$ with SVD $M = U\Sigma V^\top$, we have:

$$S = \frac{1}{\sqrt{2}}\begin{bmatrix} U & U \\ V & -V \end{bmatrix}\begin{bmatrix} \Sigma & 0 \\ 0 & -\Sigma \end{bmatrix}\frac{1}{\sqrt{2}}\begin{bmatrix} U & U \\ V & -V \end{bmatrix}^\top := QDQ^\top.$$

We define $\widetilde{S}$ as the symmetrized version of matrix $Y$, and let $\widehat{Q} \in \mathbb{R}^{n\times 2r}$ be the matrix of eigenvectors of the best $2r$-rank approximation of $\widetilde{S}$. Note that:

$$\|Q - \widehat{Q}(\widehat{Q}^\top Q)\|_{2\to\infty} = \max\left\{\|U - \widehat{U}(\widehat{U}^\top U)\|_{2\to\infty}, \|V - \widehat{V}(\widehat{V}^\top V)\|_{2\to\infty}\right\}.$$

We will also repeatedly use the properties (31). To keep the notation simple, define $W_{\widehat{Q}} = \widehat{Q}^\top Q$. Thus, proving Lemma 32 is equivalent to showing:

$$\|Q - \widehat{Q}W_{\widehat{Q}}\|_{2\to\infty}$$

$$\lesssim \frac{1}{\sigma_r(M)}\left[\|Q\|_{2\to\infty}(\frac{\sigma_1(M)}{\sigma_r(M)}\|\widetilde{S} - S\| + \frac{1}{T}g_\delta(TM)\log\left(\frac{ne}{\delta}\right)) + \sqrt{\frac{r\|M\|_\infty}{T}\log\left(\frac{ne}{\delta}\right)}\right]$$

with high probability. Define $E = \widetilde{S} - S$. Now, as in Lemma 33, we have:

$$\|Q - \widehat{Q}W_{\widehat{Q}}\|_{2\to\infty} \leq \frac{1}{\sigma_r(M)}\left(\frac{4\|\widetilde{S}Q\|_{2\to\infty}\|E\|}{\sigma_r(M)} + \|EQ\|_{2\to\infty} + 2\|\widetilde{S}(Q - \widehat{Q}W_{\widehat{Q}})\|_{2\to\infty}\right) \tag{36}$$

under the assumption that $\|E\| \leq c_1\sigma_r(M)$. Indeed, it is straightforward to show the same bounds as in Lemma 4.14 in [14] - note that the boundedness assumption is not used in these lemmas. We bound the three terms in Equation (36) as follows:

1. To bound the first term, we use:

$$\|\widetilde{S}Q\|_{2\to\infty} \leq \|SQ\|_{2\to\infty} + \|EQ\|_{2\to\infty} \leq \|Q\|_{2\to\infty}\|S\| + \|EQ\|_{2\to\infty} \tag{37}$$

where we first used the triangle inequality and then $\|SQ\|_{2\to\infty} = \|QD\|_{2\to\infty} \leq \|Q\|_{2\to\infty}\|S\|$.

2. For the second term, according to Lemma 29, we obtain with probability at least $1 - \delta$:

$$\|EQ\|_{2\to\infty} \lesssim \frac{1}{T}\left[\|Q\|_F\sqrt{T\|M\|_\infty\log(ne/\delta)} + g_\delta(TM)\log(ne/\delta)\|Q\|_{2\to\infty}\right]. \tag{38}$$

Moreover, we will use $\|Q\|_F \leq \sqrt{2r}$ and $\|Q\|_{2\to\infty} \leq \mu\sqrt{\frac{r}{n}}$, which follow from the low-rank and incoherence assumptions.

3. Finally, regarding the last term in Equation (36), we split it using the triangle inequality as follows:

$$\|\widetilde{S}(Q - \widehat{Q}W_{\widehat{Q}})\|_{2\to\infty} \leq \|S(Q - \widehat{Q}W_{\widehat{Q}})\|_{2\to\infty} + \|E(Q - \widehat{Q}W_{\widehat{Q}})\|_{2\to\infty},$$

and from Step 3 of proof of Theorem 4.2 in [14] we have:

$$\|S(Q - \widehat{Q}W_{\widehat{Q}})\|_{2\to\infty} \leq \|Q\|_{2\to\infty}\|S\|\|Q^\top(Q - \widehat{Q}W_{\widehat{Q}})\| \lesssim \|Q\|_{2\to\infty}\|S\|\frac{\|E\|^2}{\sigma_r^2(M)}, \tag{39}$$

where we used $\|Q^\top(Q - \widehat{Q}W_{\widehat{Q}})\| = \|\sin\Theta(Q,\widehat{Q})\|^2$. The remaining of the proof consists in bounding $\|E(Q - \widehat{Q}W_{\widehat{Q}})\|_{2\to\infty} = \max_{\ell=1,\dots,n}\|E_{\ell,:}(Q - \widehat{Q}W_{\widehat{Q}})\|$. First note that the matrix $Q - \widehat{Q}W_{\widehat{Q}}$ depends on $E$ and thus we cannot apply Lemma 29 immediately. Instead, we will use the leave-one-out method, and define a matrix $\widetilde{S}^{(\ell)} \in \mathbb{R}^{n\times n}$ as follows:

$$\widetilde{S}^{(\ell)}_{i,j} = \begin{cases} \widetilde{S}_{i,j}, & \text{if } i \neq \ell \text{ or } j \neq \ell \\ S_{i,j}, & \text{otherwise} \end{cases}$$

Then define $\widehat{Q}^{(\ell)} \in \mathbb{R}^{n\times 2r}$ as a matrix of eigenvectors corresponding to $2r$ greatest (in absolute value) eigenvalues of matrix $\widetilde{S}^{(\ell)}$. Define $W_{\widehat{Q}^{(\ell)}}$ accordingly. Then we have:

$$\|E(Q - \widehat{Q}W_{\widehat{Q}})\|_{2\to\infty} \leq 2 \max_{1\leq\ell\leq n}\left\{\|E_{\ell,:}(Q - \widehat{Q}^{(\ell)}W_{\widehat{Q}^{(\ell)}})\|, \|E_{\ell,:}(\widehat{Q}^{(\ell)}W_{\widehat{Q}^{(\ell)}} - \widehat{Q}W_{\widehat{Q}})\|\right\}. \quad (40)$$

(3a) Since $E_{\ell,:}$ is statistically independent of $Q - \widehat{Q}^{(\ell)}W_{\widehat{Q}^{(\ell)}}$, the first term from (40) can be bounded according to Lemma 29 for any $1 \leq \ell \leq n$ as follows:

$$\|E_{\ell,:}(Q - \widehat{Q}^{(\ell)}W_{\widehat{Q}^{(\ell)}})\| \lesssim \frac{1}{T}\Big[\|Q - \widehat{Q}^{(\ell)}W_{\widehat{Q}^{(\ell)}}\|_F\sqrt{T\|M\|_\infty\log\left(\frac{ne}{\delta}\right)}$$
$$+ g_\delta(TM)\log\left(\frac{ne}{\delta}\right)\|Q - \widehat{Q}^{(\ell)}W_{\widehat{Q}^{(\ell)}}\|_{2\to\infty}\Big]$$

with probability at least $1 - \delta$. After applying the triangle inequality to the second term and using $\|\cdot\|_{2\to\infty} \leq \|\cdot\|_F$, we get:

$$\|Q - \widehat{Q}^{(\ell)}W_{\widehat{Q}^{(\ell)}}\|_{2\to\infty} \leq \|Q - \widehat{Q}W_{\widehat{Q}}\|_{2\to\infty} + \|\widehat{Q}W_{\widehat{Q}} - \widehat{Q}^{(\ell)}W_{\widehat{Q}^{(\ell)}}\|_F. \quad (41)$$

Thus, combining the last two inequalities yields:

$$\|E_{\ell,:}(Q - \widehat{Q}^{(\ell)}W_{\widehat{Q}^{(\ell)}})\|$$
$$\lesssim \|Q - \widehat{Q}W_{\widehat{Q}}\|_F\sqrt{T^{-1}\|M\|_\infty\log\left(\frac{ne}{\delta}\right)} + T^{-1}g_\delta(TM)\log\left(\frac{ne}{\delta}\right)\|Q - \widehat{Q}W_{\widehat{Q}}\|_{2\to\infty}$$
$$+ \|\widehat{Q}W_{\widehat{Q}} - \widehat{Q}^{(\ell)}W_{\widehat{Q}^{(\ell)}}\|_F\left(\sqrt{T^{-1}\|M\|_\infty\log\left(\frac{ne}{\delta}\right)} + T^{-1}g_\delta(TM)\log\left(\frac{ne}{\delta}\right)\right) \quad (42)$$

for all $1 \leq \ell \leq n$.

(3b) The second term from (40) can be bounded very roughly as follows:

$$\|E_{\ell,:}(\widehat{Q}^{(\ell)}W_{\widehat{Q}^{(\ell)}} - \widehat{Q}W_{\widehat{Q}})\| \leq \|E\|\|\widehat{Q}^{(\ell)}W_{\widehat{Q}^{(\ell)}} - \widehat{Q}W_{\widehat{Q}}\|_F. \quad (43)$$

Similar to the Step 2.2 in the proof of Theorem 4.2 in [14], we have:

$$\|\widehat{Q}^{(\ell)}W_{\widehat{Q}^{(\ell)}} - \widehat{Q}W_{\widehat{Q}}\|_F \leq \frac{16}{\sigma_r(M)}(\|E_{\ell,:}(Q - \widehat{Q}^{(\ell)}W_{\widehat{Q}^{(\ell)}})\| + \|EQ\|_{2\to\infty}$$
$$+ \|E\|\|Q - \widehat{Q}^{(\ell)}W_{\widehat{Q}^{(\ell)}}\|_{2\to\infty} + \|E\|\|Q\|_{2\to\infty}).$$

Applying again the inequality (41) and moving the term $\|\widehat{Q}^{(\ell)}W_{\widehat{Q}^{(\ell)}} - \widehat{Q}W_{\widehat{Q}}\|_F$ to the left side of inequality, we get under assumption $\|E\| \leq c_1\sigma_r(M)$ that:

$$\|\widehat{Q}^{(\ell)}W_{\widehat{Q}^{(\ell)}} - \widehat{Q}W_{\widehat{Q}}\|_F \lesssim \frac{1}{\sigma_r(M)}(\|E_{\ell,:}(Q - \widehat{Q}^{(\ell)}W_{\widehat{Q}^{(\ell)}})\| + \|EQ\|_{2\to\infty}$$
$$+ \|E\|\|Q - \widehat{Q}W_{\widehat{Q}}\|_{2\to\infty} + \|E\|\|Q\|_{2\to\infty}). \quad (44)$$

After substitution of the results from (37), (38), (39), (40), (42), (43) and (44) into Equation (36) and using assumptions stated in the lemma, we obtain the statement of the lemma. $\qquad\square$

### E.3 Error decomposition in the two-to-infinity norm

Below, we present a decomposition for the error of subspace recovery in the norm $\| \cdot \|_{2\to\infty}$. We borrow this result from [14] and provide its proof for completeness.

**Lemma 33** (Lemma 4.16 in [14])*. Let $S, \widetilde{S} \in \mathbb{R}^{n\times n}$ be symmetric matrices and assume that $S$ and $\widetilde{S}$ are of rank $r$. Let $Q, \widehat{Q} \in \mathcal{O}_{n\times r}$ be the corresponding $r$ singular vectors of $S$ and $\widetilde{S}$, respectively. Denote $E = \widetilde{S} - S$. Under the condition $\|E\| \le \sigma_r(S)/2$, we have:*

$$\|Q - \widehat{Q}\widehat{Q}^\top Q\|_{2\to\infty} \le \frac{4\|\widetilde{S}Q\|_{2\to\infty}\|E\|}{(\sigma_r(S))^2} + \frac{\|EQ\|_{2\to\infty}}{\sigma_r(S)} + \frac{2\|\widetilde{S}(Q - \widehat{Q}\widehat{Q}^\top Q)\|_{2\to\infty}}{\sigma_r(S)}$$

*Proof of Lemma 33.* Since $S$ is a symmetric matrix of rank $r$, by SVD we write $S = Q\Sigma Q^\top$, where the matrix $\Sigma = \operatorname{diag}(\sigma_1(S), \ldots, \sigma_r(S))$. For ease of notations, let us further denote $W = \widehat{Q}Q^\top$. We have

$$
\begin{aligned}
\|Q - \widehat{Q}\widehat{Q}^\top Q\|_{2\to\infty} &= \|SQ\Sigma^{-1} - \widehat{Q}W\|_{2\to\infty} \\
&\le \|\widetilde{S}Q\Sigma^{-1} - \widehat{Q}W\|_{2\to\infty} + \|EQ\Sigma^{-1}\|_{2\to\infty} \\
&\le \frac{\|\widetilde{S}Q - \widehat{Q}W\Sigma\|_{2\to\infty}}{\sigma_r(S)} + \frac{\|EQ\|_{2\to\infty}}{\sigma_r(S)}
\end{aligned}
$$

Now, we focus on the term $\|\widetilde{S}Q - \widehat{Q}W\Sigma\|_{2\to\infty}$. To that end, we first establish the identity

$$
\begin{aligned}
\widehat{Q}W\Sigma &= \widehat{Q}\widehat{Q}^\top Q\Sigma \\
&= \widehat{Q}\widehat{Q}^\top SQ \\
&= \widehat{Q}\widehat{Q}^\top \widehat{S}Q + \widehat{Q}\widehat{Q}^\top EQ \\
&= \widehat{Q}\widehat{\Sigma}\widehat{Q}^\top Q + \widehat{Q}\widehat{Q}^\top EQ \\
&= \widetilde{S}\widehat{Q}\widehat{Q}^\top Q + \widehat{Q}\widehat{Q}^\top EQ
\end{aligned}
$$

where we use the identities $SQ = Q\Sigma$, $\widehat{Q}^\top \widetilde{S} = \widehat{\Sigma}\widehat{Q}^\top$, and $\widehat{Q}\widehat{\Sigma}\widehat{Q}^\top = \widetilde{S}\widehat{Q}\widehat{Q}^\top$. Then, we observe that

$$
\begin{aligned}
\|\widetilde{S}Q - \widehat{Q}W\Sigma\|_{2\to\infty} &= \|\widetilde{S}Q - \widetilde{S}\widehat{Q}\widehat{Q}^\top Q + \widehat{Q}\widehat{Q}^\top EQ\|_{2\to\infty} \\
&\le \|\widetilde{S}(Q - \widehat{Q}\widehat{Q}^\top Q)\|_{2\to\infty} + \|\widehat{Q}\widehat{Q}^\top EQ\|_{2\to\infty}
\end{aligned}
$$

Next, we note that when $\|E\| \le \sigma_r(S)/2$, we have

$$
\begin{aligned}
\|\widehat{Q}\widehat{Q}^\top EQ\|_{2\to\infty} &= \|\widetilde{S}\widehat{Q}\widehat{\Sigma}^{-1}\widehat{Q}^\top EQ\|_{2\to\infty} \\
&\le \|\widetilde{S}\widehat{Q}\|_{2\to\infty}\|\widehat{\Sigma}^{-1}\|\|\widehat{Q}^\top\|\|E\|\|Q\| \\
&\le \frac{\|\widetilde{S}\widehat{Q}\operatorname{sgn}(\widehat{Q}^\top Q)\|_{2\to\infty}\|E\|}{\sigma_r(\widetilde{S})}.
\end{aligned}
$$

At this point, we try to bound $\|\widetilde{S}\widehat{Q}\|_{2\to\infty}$ and $\sigma_r(\widetilde{S})$, under the condition $\|E\| \le \sigma_r(S)/2$. First, we can easily see by Weyl's inequality we have $|\sigma_r(\widetilde{S}) - \sigma_r(S)| \le \|E\|$, which entails under the assumed condition that $\sigma_r(\widetilde{S}) \ge \sigma_r(S)/2$. Next, we observe:

$$
\begin{aligned}
\|\widetilde{S}\widehat{Q}\|_{2\to\infty} &= \|\widetilde{S}\widehat{Q}\operatorname{sgn}(\widehat{Q}^\top Q)\|_{2\to\infty} \\
&\le \|\widetilde{S}\widehat{Q}\widehat{Q}^\top Q\|_{2\to\infty} + \|\widetilde{S}\widehat{Q}\|_{2\to\infty}\|\operatorname{sgn}(\widehat{Q}^\top Q) - \widehat{Q}^\top Q\| \\
&\le \|\widetilde{S}\widehat{Q}\widehat{Q}^\top Q\|_{2\to\infty} + \frac{2\|\widetilde{S}\widehat{Q}\|_{2\to\infty}\|E\|^2}{\sigma_r(S)^2} \\
&\le \|\widetilde{S}\widehat{Q}\widehat{Q}^\top Q\|_{2\to\infty} + \frac{\|\widetilde{S}\widehat{Q}\|_{2\to\infty}}{2}
\end{aligned}
$$

where we used the Davis-Kahan's inequality and properties of the $\mathrm{sgn}(\cdot)$, to upper bound $\|\mathrm{sgn}(\widehat{Q}^\top Q) - \widehat{Q}^\top Q\| \leq \|\sin(\widehat{Q}, Q)\|^2 \leq 2\|E\|^2/(\sigma_r(S))^2$. Thus, leading to:

$$\|\widetilde{S}\widehat{Q}\|_{2\to\infty} \leq 2\|\widetilde{S}\widehat{Q}\widehat{Q}^\top Q\|_{2\to\infty}$$

Moving forward we obtain:

$$
\begin{aligned}
\|\widehat{Q}\widehat{Q}^\top E Q\|_{2\to\infty} &\leq \frac{4\|\widetilde{S}\widehat{Q}\widehat{Q}^\top Q\|_{2\to\infty}\|E\|}{\sigma_r(S)} \\
&\leq \frac{4\left(\|\widetilde{S}(\widehat{Q}\widehat{Q}^\top Q - Q)\|_{2\to\infty} + \|\widetilde{S}Q\|_{2\to\infty}\right)\|E\|}{\sigma_r(S)} \\
&\leq 2\|\widetilde{S}(\widehat{Q}\widehat{Q}^\top Q - Q)\|_{2\to\infty} + \frac{4\left(\|\widetilde{S}Q\|_{2\to\infty}\right)\|E\|}{\sigma_r(S)}
\end{aligned}
$$

Now, putting everything together we conclude that:

$$\|Q - \widehat{Q}\widehat{Q}^\top Q\|_{2\to\infty} \leq \frac{2\|\widetilde{S}(\widehat{Q}\widehat{Q}^\top Q - Q)\|_{2\to\infty}}{\sigma_r(S)} + \frac{4\|\widetilde{S}Q\|_{2\to\infty}\|E\|}{(\sigma_r(S))^2} + \frac{\|EQ\|_{2\to\infty}}{\sigma_r(S)}$$

$\square$

# F  Row-wise and entry-wise matrix estimation errors

In this appendix, we provide a series of results about quantifying the matrix estimation error using different norms. It is important to note that all these results require a control of the error in the two-to-infinity norm, which in turn requires a subspace recovery guarantee in the two-to-infinity norm. Lemmas 35 and 37 are specific to our analysis for the estimation of the transition matrices. Lemmas 34 and 36 are common to the analysis of both the estimation of reward matrices and transition matrices. The results presented in this appendix are used in the proofs of the main results, presented in Appendix B.

## F.1  Bounding $\|M - \widehat{M}\|_{2\to\infty}$

**Lemma 34.** *Let $M, \widehat{M}$ be as in §2. Assume that there exists a sufficiently small universal constant $c_1 > 0$ such that $\|M - \widetilde{M}\| \leq c_1 \sigma_r(M)$. Then, there exists a universal constant $c_2 > 0$ such that*

$$\|\widehat{M} - M\|_{2\to\infty} \leq c_2 \sigma_1(M) \left[ \|U - \widehat{U}(\widehat{U}^\top U)\|_{2\to\infty} + \|U\|_{2\to\infty} \frac{\|\widetilde{M} - M\|}{\sigma_r(M)} \right].$$

*Proof.* We start by using definition of $\widehat{M}$ as a projection of matrix $\widetilde{M}$, and then use the triangle inequality and the inequality (3) to obtain:

$$
\begin{aligned}
\|\widehat{M} - M\|_{2\to\infty} &= \|\Pi_{\widehat{U}} \widetilde{M} - \Pi_U M\|_{2\to\infty} \\
&= \|(\Pi_{\widehat{U}} - \Pi_U)(\widetilde{M} - M) + \Pi_U(\widetilde{M} - M) + (\Pi_{\widehat{U}} - \Pi_U)M\|_{2\to\infty} \\
&\leq \|(\Pi_{\widehat{U}} - \Pi_U)(\widetilde{M} - M)\|_{2\to\infty} + \|\Pi_U(\widetilde{M} - M)\|_{2\to\infty} + \|(\Pi_{\widehat{U}} - \Pi_U)M\|_{2\to\infty} \\
&\leq \|\Pi_{\widehat{U}} - \Pi_U\|_{2\to\infty}(\|\widetilde{M} - M\| + \|M\|) + \|\Pi_U\|_{2\to\infty}\|\widetilde{M} - M\|. \qquad (45)
\end{aligned}
$$

Moreover, we note that $\|\Pi_U\|_{2\to\infty} = \|U\|_{2\to\infty}$ (refer to Proposition 6.6 in [12]). In the remaining of the proof, we upper bound $\|\Pi_{\widehat{U}} - \Pi_U\|_{2\to\infty}$ from (45). For any orthogonal matrix $R \in \mathcal{O}^{r\times r}$, we have

$$
\begin{aligned}
\|\Pi_{\widehat{U}} - \Pi_U\|_{2\to\infty} &= \|\widehat{U}\widehat{U}^\top - UU^\top\|_{2\to\infty} \\
&= \|\widehat{U}RR^\top\widehat{U}^\top - UR^\top\widehat{U}^\top + UR^\top\widehat{U}^\top - UU^\top\|_{2\to\infty} \\
&\leq \|\widehat{U}RR^\top\widehat{U}^\top - UR^\top\widehat{U}^\top\|_{2\to\infty} + \|UR^\top\widehat{U}^\top - UU^\top\|_{2\to\infty} \\
&\leq \|\widehat{U}R - U\|_{2\to\infty}\|R^\top\widehat{U}^\top\| + \|U\|_{2\to\infty}\|R^\top\widehat{U}^\top - U^\top\| \\
&\leq \|U - \widehat{U}R\|_{2\to\infty} + \|U\|_{2\to\infty}\|U - \widehat{U}R\|. \qquad (46)
\end{aligned}
$$

Recall the definition of $\mathrm{sgn}$ function given in the notation presented in §1 and choose the matrix $R$ as $R = \mathrm{sgn}(\widehat{U}^\top U)$. For this choice of $R$ we have according to Davis-Kahan's theorem (Corollary 2.8 in [14]):

$$\|U - \widehat{U}R\| \leq \sqrt{2}\|\sin\Theta(\widehat{U}, U)\| \leq \frac{2\|M - \widetilde{M}\|}{\sigma_r(M)}. \qquad (47)$$

Define the matrix $W_{\widehat{U}} = \widehat{U}^\top U$. We use the facts that

$$\|U - \widehat{U}R\|_{2\to\infty} \leq \|U - \widehat{U}W_{\widehat{U}}\|_{2\to\infty} + \|\widehat{U}\|_{2\to\infty}\|W_{\widehat{U}} - R\| \qquad (48)$$

and that $W_{\widehat{U}}$ is very close to $R$ according to the proof of Lemma 4.15 in [14] to show:

$$\|W_{\widehat{U}} - R\| = \|\widehat{U}^\top U - \mathrm{sgn}(\widehat{U}^\top U)\| = \|\sin\Theta(\widehat{U}, U)\|^2 \leq \frac{2\|M - \widetilde{M}\|^2}{\sigma_r^2(F)}. \qquad (49)$$

We also have $\sigma_i(R) = 1$ for $i \in [r]$ and according to Weyl's inequality $\sigma_{\min}(W_{\widehat{U}}) \geq \sigma_{\min}(R) - \|W_{\widehat{U}} - R\| = 1 - \|W_{\widehat{U}} - R\|$. Combining these results under assumption $\|M - \widetilde{M}\| < \sigma_r(M)/\sqrt{2}$ we obtain:

$$\|W_{\widehat{U}}^{-1}\| = \frac{1}{\sigma_{\min}(W_{\widehat{U}})} \leq \frac{1}{1 - \|W_{\widehat{U}} - R\|} \leq \frac{1}{1 - \frac{2\|M - \widetilde{M}\|^2}{\sigma_r^2(M)}}.$$

Thus:

$$\|\widehat{U}\|_{2\to\infty} \le \|\widehat{U}W_{\widehat{U}}\|_{2\to\infty}\|W_{\widehat{U}}^{-1}\| \le \frac{1}{1 - \frac{2\|M-\widetilde{M}\|^2}{\sigma_r^2(M)}}(\|U\|_{2\to\infty} + \|U - \widehat{U}W_{\widehat{U}}\|_{2\to\infty}). \quad (50)$$

Combining Equations (48), (49), (50) we get:

$$\|U - \widehat{U}R\|_{2\to\infty} \le \frac{1}{1 - \frac{2\|M-\widetilde{M}\|^2}{\sigma_r^2(M)}}(\|U - \widehat{U}W_{\widehat{U}}\|_{2\to\infty} + \frac{2\|M-\widetilde{M}\|^2}{\sigma_r^2(M)}\|U\|_{2\to\infty})$$

and combining the last equality with (46) and (47), we have

$$\|\Pi_{\widehat{U}} - \Pi_U\|_{2\to\infty} \le \frac{\|U - \widehat{U}W_{\widehat{U}}\|_{2\to\infty}}{1 - \frac{2\|M-\widetilde{M}\|^2}{\sigma_r^2(M)}} + \left(\frac{\frac{2\|M-\widetilde{M}\|^2}{\sigma_r^2(M)}}{1 - \frac{2\|M-\widetilde{M}\|^2}{\sigma_r^2(M)}} + \frac{2\|M-\widetilde{M}\|}{\sigma_r(M)}\right)\|U\|_{2\to\infty}. \quad (51)$$

Finally, substituting the obtained bound into Equation (45) and using assumption $\|M - \widetilde{M}\| \le c_1\sigma_r(M)$ for simplification, we obtain the statement of the lemma. $\qquad\square$

## F.2   Bounding $\|P - \widehat{P}\|_{1\to\infty}$

**Lemma 35.** *Let $P, \widehat{P}$ be as in Model II in §2. We have:*

$$\|\widehat{P} - P\|_{1\to\infty} \le 2\frac{\sqrt{n}\|\widehat{M} - M\|_{2\to\infty}}{\min_{j\in[n]}\|M_{j,:}\|_1}.$$

*Proof.* Starting with the definition of $\widehat{P}$, we get:

$$\|\widehat{P} - P\|_{1\to\infty} = \max_{i\in[n]}\|\widehat{P}_{i,:} - P_{i,:}\|_1 = \max_{i\in[n]}\left\|\frac{(\widehat{M}_{i,:})_+}{\|(\widehat{M}_{i,:})_+\|_1} - \frac{M_{i,:}}{\|M_{i,:}\|_1}\right\|_1$$

$$\le 2\max_{i\in[n]}\frac{\|\widehat{M}_{i,:} - M_{i,:}\|_1}{\|M_{i,:}\|_1} \le 2\frac{\sqrt{n}\max_{i\in[n]}\|\widehat{M}_{i,:} - M_{i,:}\|}{\min_{j\in[n]}\|M_{j,:}\|_1},$$

where the first inequality follows from Lemma 2 in [63] and the second by equivalence of norms. Moreover, note that the above inequality holds even in the case when $\|(\widehat{M}_{i,:})_+\|_1 = 0$ (and thus $\widehat{P}_{i,:} = \frac{1}{n}\mathbf{1}_n$), but the bound is vacuous in this case. $\qquad\square$

## F.3   Bounding $\|M - \widehat{M}\|_{\infty}$

**Lemma 36.** *Let $M, \widehat{M}$ be as in §2. Assume that there exists a sufficiently small universal constant $c_1 > 0$ such that $\|M - \widetilde{M}\| \le c_1\sigma_r(M)$. Then, there exists a universal constant $c_2 > 0$ such that*

$$\|\widehat{M} - M\|_{\infty} \le c_2\|M\|_{2\to\infty}\left(\frac{\|M-\widetilde{M}\|}{\sigma_r(M)}\|V\|_{2\to\infty} + \|V - \widehat{V}W_{\widehat{V}}\|_{2\to\infty}\right)$$

$$+ c_2\|M - \widehat{M}\|_{2\to\infty}(\|V\|_{2\to\infty} + \|V - \widehat{V}W_{\widehat{V}}\|_{2\to\infty}).$$

*Proof.* Similarly to the decomposition leading to Equation (45), we can upper bound the infinity norm error easily from the following decomposition:

$$\|\widehat{M} - M\|_{\infty} = \|\widehat{M}\Pi_{\widehat{V}} - M\Pi_V\|_{\infty}$$

$$\le \|\widehat{M}\|_{2\to\infty}\|\Pi_{\widehat{V}} - \Pi_V\|_{2\to\infty} + \|\widehat{M} - M\|_{2\to\infty}\|\Pi_V\|_{2\to\infty}$$

$$\le (\|\widehat{M} - M\|_{2\to\infty} + \|M\|_{2\to\infty})\|\Pi_{\widehat{V}} - \Pi_V\|_{2\to\infty} + \|\widehat{M} - M\|_{2\to\infty}\|V\|_{2\to\infty},$$

where we used the inequality (4) together with the triangle inequalities and the fact that projection matrices are symmetric. To bound $\|\Pi_{\widehat{V}} - \Pi_V\|_{2\to\infty}$, we use the same approach as that used in (51) (just replacing $U$ by $V$), and we obtain:

$$\|\Pi_{\widehat{V}} - \Pi_V\|_{2\to\infty} \leq \frac{\|V - \widehat{V}W_{\widehat{V}}\|_{2\to\infty}}{1 - \frac{2\|M - \widetilde{M}\|^2}{\sigma_r^2(M)}} + \left( \frac{\frac{2\|M-\widetilde{M}\|^2}{\sigma_r^2(M)}}{1 - \frac{2\|M-\widetilde{M}\|^2}{\sigma_r^2(M)}} + \frac{2\|M - \widetilde{M}\|}{\sigma_r(M)} \right) \|V\|_{2\to\infty}.$$

$\square$

## F.4  Bounding $\|P - \widehat{P}\|_\infty$

**Lemma 37.** *Let* $P, \widehat{P}$ *be as in Model II in §2. Assume that* $\mathcal{D} = \min_{i\in[n]} \|M_{i,:}\|_1 > 0$. *If* $\|\widehat{M} - M\|_{1\to\infty} \leq \frac{1}{2}\mathcal{D}$, *then*

$$\|\widehat{P} - P\|_\infty \leq 2\frac{\|\widehat{M} - M\|_\infty}{\mathcal{D}} + 2\frac{\sqrt{n}\|M\|_\infty}{\mathcal{D}^2}\|\widehat{M} - M\|_{2\to\infty}.$$

*Proof.* First note that for any $i \in [n]$:

$$\left| \|(\widehat{M}_{i,:})_+\|_1 - \|M_{i,:}\|_1 \right| \leq \|(\widehat{M}_{i,:})_+ - M_{i,:}\|_1 \leq \|\widehat{M}_{i,:} - M_{i,:}\|_1 \leq \frac{\|M_{i,:}\|_1}{2}, \qquad (52)$$

where the first inequality follows from the reverse triangle inequality, the second from $|\max(0,x) - y| \leq |x - y|$ for all $y > 0$ and $x \in \mathbb{R}$, and the last inequality follows from the assumption in the lemma. This implies that $\|(\widehat{M}_{i,:})_+\|_1 > 0$ for all $i \in [n]$, which further implies that $\widehat{P}$ is defined by: for all $i \in [n]$,

$$\widehat{P}_{i,:} = (\widehat{M}_{i,:})_+/\|(\widehat{M}_{i,:})_+\|_1. \qquad (53)$$

Furthermore, we have for all $i, j = 1, \ldots, n$,

$$|\widehat{P}_{i,j} - P_{i,j}| = \left| \frac{(\widehat{M}_{i,j})_+}{\|(\widehat{M}_{i,:})_+\|_1} - \frac{M_{i,j}}{\|M_{i,:}\|_1} \right| \leq \left| \frac{\widehat{M}_{i,j}}{\|(\widehat{M}_{i,:})_+\|_1} - \frac{M_{i,j}}{\|M_{i,:}\|_1} \right|$$

$$\leq \frac{1}{\|M_{i,:}\|_1} \left| \widehat{M}_{i,j} - M_{i,j} \right| + |\widehat{M}_{i,j}| \left| \frac{1}{\|(\widehat{M}_{i,:})_+\|_1} - \frac{1}{\|M_{i,:}\|_1} \right|$$

$$= \frac{1}{\|M_{i,:}\|_1} \left| \widehat{M}_{i,j} - M_{i,j} \right| + \frac{|\widehat{M}_{i,j}|}{\|M_{i,:}\|_1} \left| \frac{1}{1 + \frac{\|(\widehat{M}_{i,:})_+\|_1 - \|M_{i,:}\|_1}{\|M_{i,:}\|_1}} - 1 \right|$$

$$= \frac{1}{\|M_{i,:}\|_1} \left| \widehat{M}_{i,j} - M_{i,j} \right| + \frac{|\widehat{M}_{i,j}|}{\|M_{i,:}\|_1} \varphi \left( \frac{\|(\widehat{M}_{i,:})_+\|_1 - \|M_{i,:}\|_1}{\|M_{i,:}\|_1} \right)$$

where we define $\varphi(x) = |x/(1+x)|$ for all $x \in \mathbb{R}\backslash\{-1\}$. Note that if $|x| < 1/2$, then $\varphi(x) \leq 2|x|$, which combined with (52) gives

$$|\widehat{P}_{i,j} - P_{i,j}| \leq \frac{1}{\|M_{i,:}\|_1} \left| \widehat{M}_{i,j} - M_{i,j} \right| + \frac{2|\widehat{M}_{i,j}|}{\|M_{i,:}\|_1} \left| \frac{\|(\widehat{M}_{i,:})_+\|_1 - \|M_{i,:}\|_1}{\|M_{i,:}\|_1} \right|$$

$$\leq \frac{1}{\|M_{i,:}\|_1} \left| \widehat{M}_{i,j} - M_{i,j} \right| + \frac{2}{\|M_{i,:}\|_1^2} \left( \left| \widehat{M}_{i,j} - M_{i,j} \right| + |\widehat{M}_{i,j}| \right) \|\widehat{M}_{i,:} - M_{i,:}\|_1.$$

Using the assumption $\|\widehat{M} - M\|_{1\to\infty} \leq \frac{1}{2}\min_{i\in[n]}\|M_{i,:}\|_1$ again, we can group first two terms, and then use $\|\widehat{M} - M\|_{1\to\infty} \leq \sqrt{n}\|\widehat{M} - M\|_{2\to\infty}$ to get the statement of the lemma.

$\square$

# G Low-rank bandits: proofs of results from Section 4

## G.1 Gap-dependent guarantees

*Proof of Theorem 7.* First, we prove the result corresponding the best entry identification problem. We proceed in several steps.

**Step 1: entry-wise concentration.** We can easily verify that for all $\ell \geq 1$, for all $(i, j) \in [m] \times [n]$, we have

$$|\widehat{\Delta}_{i,j}^{(\ell)} - \Delta_{i,j}| \leq 2\|\widehat{M}^{(\ell)} - M^\star\|_\infty.$$

Therefore, applying Theorem 1, we have, for $\delta > 0$, and $T_\ell \geq c_1(m + n) \log^3((e^2(m + n)(mn)/\delta_\ell)$,

$$\mathbb{P}\left(|\widehat{\Delta}_{i,j} - \Delta_{i,j}| > 2C_1\sqrt{\frac{e(m + n)}{T_\ell} \log^3\left(\frac{e(m + n)mnT_\ell}{\delta_\ell}\right)}\right) \leq \frac{\delta_\ell}{mn}$$

for some $c_1, C_1 > 0$ sufficiently large. In particular, we can choose $C_1 = C(\mu^{11/2}\kappa^2 r^{1/2} + \mu^3\kappa r^{3/2}(m + n)/\sqrt{mn})$, and $c_1 = c\mu^4\kappa^2 r^2$, but under a homogeneous reward matrix these constants are $\Theta(1)$. Thus, by a union bound and always under the same conditions, we have

$$\mathbb{P}\left(\max_{(i,j) \in [m] \times [n]} |\widehat{\Delta}_{i,j} - \Delta_{i,j}| > 2C_1\sqrt{\frac{e(m + n)}{T_\ell} \log^3\left(\frac{e(m + n)mnT_\ell}{\delta_\ell}\right)}\right) \leq \delta_\ell.$$

Next, we wish to choose $T_\ell$ so that we have

$$\mathbb{P}\left(\max_{i,j}|\widehat{\Delta}_{i,j} - \Delta_{i,j}| \leq 2^{-(\ell+2)}\right) \geq 1 - \delta_\ell. \tag{54}$$

Note that in order for the above guarantee to hold, it is sufficient to have:

$$T_\ell \geq c_1(m + n) \log^3\left(\frac{e^2(m + n)(mn)}{\delta_\ell}\right),$$

$$T_\ell \geq 2\sqrt{e}C_1^2(m + n)2^{2(\ell-2)} \log^3\left(\frac{e(m + n)(mn)}{\delta_\ell}\right).$$

This can be achieved if we choose

$$T_\ell = \left\lceil C_3 2^{2(\ell-2)}(m + n) \log^3\left(\frac{2^{2(\ell-2)}(m + n)}{\delta_\ell}\right)\right\rceil, \tag{55}$$

for some positive constant $C_3 > 0$ large enough which can be determined explicitly and only depend on $c_1, C_1$. Indeed, this can be deduced from the basic fact that if $T_\ell^{1/3} \geq 2a \log(2a) + 2b$, then $T_\ell^{1/3} \geq a \log(T_\ell^{1/3}) + b$. We spare the reader these tedious calculations and only argue that such $C_3$ exists and can be computed explicitly.

**Step 2: Good events.** We define $S_\ell = \{(i, j) \in [n] \times [m] : \Delta_{i,j} \leq 2^{-\ell}\}$ and the good events under which we correctly find the best entry as

$$\mathcal{E}_\ell = \{\mathcal{A}_{\ell+1} \subseteq S_{\ell+1}\} \cap \{(i^\star, j^\star) \in \mathcal{A}_{\ell+1}\}.$$

We show that the good event $\mathcal{E}_\ell$ happens with high probability conditionally on $\mathcal{E}_1, \ldots, \mathcal{E}_{\ell-1}$. Observe that by independence of the entries sampled at epoch $\ell$ from those of the previous epochs, we have based on (54)

$$\mathbb{P}\left(\max_{i,j}|\widehat{\Delta}_{i,j} - \Delta_{i,j}| \leq 2^{-(\ell+2)}\Big|\mathcal{E}_{\ell-1}, \ldots, \mathcal{E}_1\right) \geq 1 - \delta_\ell$$

Now, conditionally on $\mathcal{E}_{\ell-1}, \ldots, \mathcal{E}_1$, under the event that $\max_{i,j} |\widehat{\Delta}_{i,j} - \Delta_{i,j}| \leq 2^{-(\ell+2)}$, if $(i, j) \in S_{\ell+1}^c \cap \mathcal{A}_{\ell+1}$ then

$$\widehat{\Delta}_{i,j}^{(\ell)} \geq \Delta_{i,j} - 2^{-(\ell+2)} > 2^{-(\ell+1)} - 2^{-(\ell+2)} = 2^{-(\ell+2)}.$$

Thus, we have

$$\mathbb{P}\left(\mathcal{A}_{\ell+1} \subseteq S_{\ell+1} \Big| \mathcal{E}_{\ell-1}, \ldots, \mathcal{E}_1\right) \geq \mathbb{P}\left(\max_{i,j} |\widehat{\Delta}_{i,j} - \Delta_{i,j}| \leq 2^{-(\ell+2)} \Big| \mathcal{E}_{\ell-1}, \ldots, \mathcal{E}_1\right) \geq 1 - \delta_\ell.$$

Furthermore, note that under the event $\max_{i,j} |\widehat{\Delta}_{i,j} - \Delta_{i,j}| \leq 2^{-(\ell+2)}$, we clearly have that $\widehat{\Delta}_{i^\star,j^\star} \leq 2^{-(\ell+2)}$ and since $(i^\star, j^\star) \in \mathcal{A}_\ell$ conditionally on $\mathcal{E}_{\ell-1}$ we conclude that

$$\mathbb{P}\left(\mathcal{E}_\ell \Big| \mathcal{E}_{\ell-1}, \ldots, \mathcal{E}_1\right) \geq \mathbb{P}\left(\max_{i,j} |\widehat{\Delta}_{i,j} - \Delta_{i,j}| \leq 2^{-(\ell+2)} \Big| \mathcal{E}_{\ell-1}, \ldots, \mathcal{E}_1\right) \geq 1 - \delta_\ell$$

**Step 3: Sample complexity.** First, we remark that when $\ell = \lceil \log_2(1/\Delta_{\min}) \rceil$, we have $S_\ell = \{(i^\star, j^\star)\}$. Therefore, under the event

$$\mathcal{E}_1 \cap \cdots \cap \mathcal{E}_{\lceil \log_2(1/\Delta_{\min}) \rceil}$$

the algorithm will stop after $\tau$ rounds, and recommend the optimal $(i^\star, j^\star)$, where

$$\begin{aligned}
\tau &\leq \sum_{\ell=1}^{\lceil \log_2(1/\Delta_{\min}) \rceil} T_\ell \\
&\leq \sum_{\ell=1}^{\lceil \log_2(1/\Delta_{\min}) \rceil} \left\lceil C_3 2^{2(\ell-2)}(m+n) \log^3\left(\frac{2^{2(\ell-2)}(m+n)}{\delta_\ell}\right) \right\rceil \\
&\leq \sum_{\ell=1}^{\lceil \log_2(1/\Delta_{\min}) \rceil} \left\lceil C_3 \frac{(m+n)}{\Delta_{\min}^2} \log^3\left(\frac{(m+n)\lceil \log_2(1/\Delta_{\min}) \rceil^2}{\Delta_{\min}^2 \delta}\right) \right\rceil \\
&\leq \log_2\left(\frac{1}{\Delta_{\min}}\right) \left\lceil C_3 \frac{(m+n)}{\Delta_{\min}^2} \log^3\left(\frac{(m+n)\lceil \log_2(1/\Delta_{\min}) \rceil^2}{\Delta_{\min}^2 \delta}\right) \right\rceil \\
&\leq \psi(n,m,\delta) := C_4 \frac{(m+n)\log(e/\Delta_{\min})}{\Delta_{\min}^2} \log^3\left(\frac{e(m+n)\log(e/\Delta_{\min})}{\Delta_{\min}\delta}\right)
\end{aligned}$$

where we recall the definition of $T_\ell$ in (55), that $\delta_\ell = \delta/\ell^2$, and where $C_4$ is a large enough universal constant. Hence, we have

$$\mathbb{P}\left((i_\tau, j_\tau) = (i^\star, j^\star), \tau \leq \psi(n,m,\delta)\right) \geq \mathbb{P}\left(\bigcap_{\ell=1}^{\lceil \log_2(1/\Delta_{\min}) \rceil} \mathcal{E}_\ell\right) \geq 1 - \delta. \tag{56}$$

This conclude the proof of the guarantee for the best entry identification. Note that we can immediately conclude from the above guarantee (56) that the sample complexity of SME-AE$(1/T^\alpha)$ for all $T \geq 1$, satisfies $\mathbb{E}[\tau \wedge T] \leq \psi(n,m,T^{-\alpha}) + T^{1-\alpha}$. Indeed, we have

$$\begin{aligned}
\mathbb{E}[\tau \wedge T] &= \mathbb{E}[(\tau \wedge T)\mathbb{1}_{\{\tau \leq \psi(n,m,T^{-\alpha})\}}] + \mathbb{E}[(\tau \wedge T)\mathbb{1}_{\{\tau > \psi(n,m,T^{-\alpha})\}}] \\
&\leq \psi(n,m,T^{-\alpha}) + T\mathbb{P}(\tau > \psi(n,m,T^{-\alpha})) \\
&\leq \psi(n,m,T^{-\alpha}) + T^{1-\alpha},
\end{aligned}$$

where the upper bound on the probability follows from (56) with $\delta = 1/T^\alpha$.

Next, we turn our attention to proving the regret upper bound. We define $\mathcal{E}_{good} = \{(\hat{\imath}_\tau, \hat{\jmath}_\tau) = (i^\star, j^\star), \tau \leq \psi(n, m, 1/T^2)\}$. We have

$$R^\pi(T) = TM_{i^\star, j^\star} - \mathbb{E}\left[\sum_{t=1}^{T} M_{i_t^\pi, j_t^\pi}\right]$$

$$= \mathbb{E}\left[\sum_{t=1}^{T} (M_{i^\star, j^\star} - M_{i_t^\pi, j_t^\pi})\mathbb{1}_{\{\mathcal{E}_{good}\}}\right] + \mathbb{E}\left[\sum_{t=1}^{T} (M_{i^\star, j^\star} - M_{i_t^\pi, j_t^\pi})\mathbb{1}_{\{\mathcal{E}_{good}^c\}}\right]$$

$$\leq \mathbb{E}\left[\sum_{t=1}^{T} (M_{i^\star, j^\star} - M_{i_t^\pi, j_t^\pi})\mathbb{1}_{\{\tau \leq \psi(n, m, T^{-2})\}}\right] + \Delta_{\max} T \mathbb{P}(\mathcal{E}_{good}^c)$$

$$\leq \mathbb{E}\left[\sum_{t=1}^{\infty} (M_{i^\star, j^\star} - M_{i_t^\pi, j_t^\pi})\mathbb{1}_{\{\tau \wedge \psi(n, m, T^{-2}) > t\}}\right] + \frac{\Delta_{\max}}{T}$$

$$\leq \mathbb{E}\left[\sum_{t=1}^{\infty} \bar{\Delta}\mathbb{1}_{\{\tau \wedge \psi(n, m, T^{-2}) > t\}}\right] + \frac{\Delta_{\max}}{T}$$

$$\leq \bar{\Delta}\psi(n, m, T^{-2}) + \frac{\Delta_{\max}}{T}$$

where in the second to last inequality, we used the tower rule together with the observation that $\mathbb{E}[(M_{i^\star, j^\star} - M_{i_t^\pi, j_t^\pi})\mathbb{1}_{\{\tau \wedge \psi(n, m, T^{-2}) > t\}}|\mathcal{F}_{t-1}] = \bar{\Delta}\mathbb{1}_{\{\tau \wedge \psi(n, m, T^{-2}) > t\}}$ where $\mathcal{F}_{t-1}$ is the $\sigma$-algebra defined by the observations up to time $t - 1$. This concludes the proof. $\qquad\square$

## G.2 Gap-independent guarantees

An immediate consequence of the regret bound in Theorem 7 is that we can have a gap-independent bound under some additional assumption. Let us define $\zeta = \Delta_{\max}/\Delta_{\min}$, then the regret bound becomes

$$R^\pi(T) \leq \frac{\zeta C_4 (m + n) \log(e/\Delta_{\min})}{\Delta_{\min}} \log^3\left(\frac{e(m + n)\log(e/\Delta_{\min})T^2}{\Delta_{\min}}\right) + \frac{\Delta_{\max}}{T}. \qquad (57)$$

At the same time, we also have the worst case bound

$$R^\pi(T) \leq \zeta\Delta_{\min}T. \qquad (58)$$

Taking the best of the two bounds (57) and (58) with the worst case choice for $\Delta_{\min}$, we get

$$R^\pi(T) = \tilde{O}\left(\zeta\sqrt{(n + m)T}\log^2((n + m)T)\right)$$

where the $\tilde{O}$ hides additional log-log terms in $m, n$ and $T$.

# H Related work

In this section, we first discuss the results for the estimation of a low-rank transition matrix presented in [63]. We then give a more detailed account of the related work for low-rank bandits.

## H.1 Low-rank transition matrix estimation

In [63], the authors try to estimate a low-rank transition matrix from the data consisting of a single trajectory of the corresponding Markov chain. In a sense, this objective is similar to ours in Model II(b). The main results of [63] are presented in Theorem 1. First observe that our results are more precise since we manage to get entry-wise guarantees. Then it is also worth noting that, in the case of homogenous transition matrices, the upper bound on $\|\widehat{P} - P\|_{1\to\infty}$ stated in Theorem 1 in [63] are similar to the upper bounds we establish in Corollary 6. However, to obtain such bounds, we believe that it is necessary to first derive guarantees for the singular subspace recovery in the $\ell_{2\to\infty}$ norm, as we do. The authors of [63] do not present any step with such guarantees for the estimation of the singular subspaces. We explain below why this step is missing and where the analysis towards the upper bound $\|\widehat{P} - P\|_{1\to\infty}$ breaks in [63].

**Proof of the guarantees for** $\|\widehat{P} - P\|_{1\to\infty}$ **in [63].** Note that in [63], the authors use $F$ in lieu of $M$. We keep our notation $M$ below to be consistent with the rest of the manuscript. In the proof of Theorem 1 in [63], the authors use the following decomposition:

$$\|\widehat{M}_{i,:} - M_{i,:}\| \le \|(\widehat{M}_{i,:} - M_{i,:})V\| + (\|\widehat{M}_{i,:} - M_{i,:}\| + \|M_{i,:}\|)\frac{C\|\widetilde{M} - M\|}{\sigma_r(M)}. \tag{59}$$

They apply concentration results on $\|(\widetilde{M} - M)V\|_{2\to\infty}$ (Lemma 8) and $\|M - \widetilde{M}\|$ (Lemma 7) to bound the two terms from above. More precisely, their proof includes (33) page 3217, a sequence of inequalities where these concentration results are used. In the fifth line of (33), the authors apply (31), the concentration result on $\|(\widetilde{M} - M)V\|_{2\to\infty}$, but to bound $\|(\widehat{M} - M)V\|_{2\to\infty}$ instead. Replacing $\widehat{M}$ by $\widetilde{M}$ is however not possible, and the analysis breaks here.

**Is there a simple solution?** We argue below that it is not easy to solve the aforementioned issue in the proof. We first claim that the two concentration bounds on $\|(\widetilde{M} - M)V\|_{2\to\infty}$ and $\|M - \widetilde{M}\|$ are not sufficient for bounding the first term from Equation (59). Specifically, for any row $i$:

$$\|(\widehat{M}_{i,:} - M_{i,:})V\| = \|(\widetilde{M}_{i,:}\widehat{V}\widehat{V}^\top - M_{i,:})V\| = \|(\widetilde{M}_{i,:} - M_{i,:})V + \widetilde{M}_{i,:}(\widehat{V}\widehat{V}^\top - VV^\top)V\|,$$

and in order to analyze the second term inside the norm, we need to deal with dependence between $\widetilde{M}$ and $\widehat{V}$. Doing this naively using the triangle inequality and Cauchy-Schwarz inequality yields:

$$\|(\widehat{M} - M)V\|_{2\to\infty} \le \|(\widetilde{M} - M)V\|_{2\to\infty} + \|\widetilde{M}(\widehat{V}\widehat{V}^\top - VV^\top)V\|_{2\to\infty}$$
$$\le \|(\widetilde{M} - M)V\|_{2\to\infty} + \|\widetilde{M}\|_{1\to\infty}\|V - \widehat{V}(\widehat{V}^\top V)\|_{2\to\infty}. \tag{60}$$

It is not clear how bounds on $\|(\widetilde{M} - M)V\|_{2\to\infty}$ and $\|M - \widetilde{M}\|$ imply a bound on $\|(\widehat{M} - M)V\|_{2\to\infty}$ since term $\|V - \widehat{V}(\widehat{V}^\top V)\|_{2\to\infty}$ does not seem to be directly bounded by these two terms. We can think of bounding $\|V - \widehat{V}(\widehat{V}^\top V)\|_2$ using Davis-Kahan's inequality:

$$\|V - \widehat{V}(\widehat{V}^\top V)\|_{2\to\infty} \le \|V - \widehat{V}(\widehat{V}^\top V)\|_2 \lesssim \frac{\|M - \widetilde{M}\|}{\sigma_r(M)},$$

where we neglect the higher order term (see Equations (47),(48),(49)). Then, with the upper bound on $\|M - \widetilde{M}\|$, we may obtain an upper bound on $\|\widehat{P} - P\|_{1\to\infty}$ but that does not have a fast decaying rate as that claimed in Theorem 1 in [63] or in our main theorems.

It is also worth noting that assuming proof of Theorem 1 in [63] holds or that more specifically, the series of inequalities leading to Equation (33) holds, one could greatly simplify the singular subspace recovery problem. In particular, since

$$\|\widetilde{M}(V - \widehat{V}\widehat{V}^\top V)\|_{2\to\infty} = \|(\widetilde{M} - \widehat{M})V\|_{2\to\infty} \le \|(\widehat{M} - M)V\|_{2\to\infty} + \|EV\|_{2\to\infty}$$

we can rewrite Equation (36) (wlog for symmetric matrix $M$ with eigenvector matrix $V$) as:

$$\|V - \widehat{V}(\widehat{V}^\top V)\|_{2\to\infty}$$
$$\leq \frac{1}{\sigma_r(M)}\left((2 + \frac{4\|E\|}{\sigma_r(M)})\|EV\|_{2\to\infty} + 2\|(\widehat{M}-M)V\|_{2\to\infty} + \frac{4\|MV\|_{2\to\infty}\|E\|}{\sigma_r(M)}\right). \quad (61)$$

Now if (33) in [63] was true, we could use the corresponding bound of the critical term $\|(\widehat{M} - M)V\|_{2\to\infty}$. This would not only greatly simplify proofs given in literature based on leave-one-out-technique, but also extend their work to Markov dependent random variables (which has not been done before). Lastly, note that we cannot skip estimation of singular subspaces by combining Equation (60) and (61) since inequality $2\|\widetilde{M}\|_{1\to\infty} < \sigma_r(M)$ does not hold in general.

## H.2 Low rank bandits

Here we survey models for low-rank bandits that have emerged recently in the literature but that are not directly related to our model. Nonetheless our guarantees can be exported there.

[31] considers a bi-linear bandit model which seems more general than that of considered [7]. Indeed, they assume that the observed reward in round $t$ after selecting a pair $(x, z) \in \mathcal{X} \times \mathcal{Z}$, is $x^\top \Theta z + \xi_t$ where $\mathcal{X} \subset \mathbb{R}^m$ and $\mathcal{Z} \subset \mathbb{R}^n$ are finite. They assume that $\Theta \in \mathbb{R}^{m\times n}$ is low rank. If we assume that $\mathcal{X} = \{e_1, \ldots, e_m\}$ and $\mathcal{Z} = \{e_1, \ldots, e_n\}$, we then recover our model and that of [7] with $M = \Theta$. However, we can also argue that if we restrict our attention to $m$ vectors from $\mathcal{X}$, say $\mathcal{X}' = \{x_1, \ldots, x_m\}$, that span $\mathbb{R}^m$, and $n$ vectors from $\mathcal{Z}$, say $\mathcal{Z}' = \{z_1, \ldots, z_n\}$, that span $\mathbb{R}^n$, then in our model and that of [7], $M_{i,j} = x_i^\top \Theta z_j$, for all $(i, j) \in [m] \times [n]$. Note that in this case, the rank of $M$ is equal to that of $\Theta$. In fact, in the first phase of the algorithm proposed by [31], the authors also restrict their attention to sets $\mathcal{X}'$ and $\mathcal{Z}'$ such that $\lambda_{\min}(\sum_{i=1}^m x_i x_i^\top)$ and $\lambda_{\min}(\sum_{i=1}^m z_i z_i^\top)$ are maximized. To simplify our exposition, we do not use the model presented by [31], instead we use that of [7].

[34] considers a generalized bandit framework with low rank structure which is rather different than the bandit framework we consider. There, the algorithm is based on the two stage idea introduced by [31], which consists in first estimating the subspace, then reducing the problem to a low-dimensional linear bandit with ambient dimension $nm$ but with roughly $n + m$ relevant dimensions. They are able obtain a minimax regret scaling as $(n + m)\sqrt{T}$. It is worth noting that both these works do not have gap-dependent bounds.

[27] is another relevant work. There, the authors consider a low-rank bandit problem similar to ours but slightly more restrictive. At time $t$, they recommend an arm $\rho(j)_t$ for each user $j$, and they observe the corresponds rewards. In other words they observe $m$ entries per round, while in our case we only observe one entry per round. They show that with an explore-then-commit algorithm, they attain a regret of order $\text{polylog}(n + m)T^{2/3}$. Their regret guarantees require an entry-wise matrix estimation guarantee with scaling comparable to ours. They use the result of Chen et al. [16] which again is only valid for independent entries and does not account for repetitive sampling. To remedy that they rely on ad-hoc pre-processing steps (see remarks 2, 3 and 4 in [27]). In our case, we believe that our matrix estimation guarantees can be immediately used in their setting and this would lead to a regret scaling of order $(n + m)^{1/3}T^{2/3}$ with the more reasonable constraint that we only observe one entry at each round. The authors also obtain an $\text{polylog}(n + m)\sqrt{T}$ guarantee but for rank-1 reward matrices only.