# OpenReview forum: "Spectral Entry-wise Matrix Estimation for Low-Rank Reinforcement Learning"
_NeurIPS.cc/2023/Conference — NeurIPS 2023 poster_

### Official Review · Reviewer_N4Ue · 2023-06-30

**Soundness:** 3 good
**Presentation:** 3 good
**Contribution:** 3 good
**Rating:** 6
**Confidence:** 4

**Summary:**

This paper presents a detailed analysis of matrix estimation problems in low-rank bandit and low-rank RL scenarios. The authors investigate the effectiveness of spectral-based matrix estimation methods and demonstrate their ability to accurately recover the singular subspaces of the matrix with minimal entry-wise error. Building upon these findings, the paper introduces a regret minimization algorithm for low-rank bandit problems and a best policy identification algorithm for reward-free RL in low-rank MDPs.

**Strengths:**

1. The paper tackles the challenging task of deriving the entry-wise error of low-rank matrix estimation, specifically focusing on the correlated noise case. This analysis extends existing results and enables the study of bandit and RL settings. The techniques employed, such as leave-one-out arguments and Poisson approximation, demonstrate technical prowess.

2. Despite being a theoretical paper, the writing is clear and easy to comprehend, enhancing its accessibility.



**Weaknesses:**

1. The paper fails to address the necessity of deriving entry-wise error in low-rank matrix estimation for bandit and RL settings. In the context of low-rank bandit problems, the primary objective is to identify the best entry, such as the largest value, in the matrix as quickly as possible. Consequently, the emphasis should be on selecting the arm with the highest reward, rather than achieving a small entry-wise error for the entire matrix. Entries with very small values do not necessitate extensive exploration. Thus, caring about entry-wise error may lead to unnecessary over-exploration, which is not desirable in low-rank bandit or low-rank MDP settings.


2. In low-rank bandit and MDP scenarios, the focus should primarily be on minimizing regret. Several existing approaches have achieved minimax regret bounds in these settings. For instance, Kang et al. (2022) addressed a more general low-rank matrix bandit problem and proved a minimax regret of $O((n+m)\sqrt{T})$. Jain and Pal (2023) also considered a similar low-rank matrix bandit problem and used a similar entry-wise matrix estimation result to achieve a regret bound of $polylog(n+m)\sqrt{T}$ for the rank-1 case. Unfortunately, the paper only discusses sub-optimal approaches in the main body, while the superior results of these references are mentioned in the last section of the supplementary materials.


3. The lack of numerical experiments to evaluate the proposed regret minimization algorithm and compare it against benchmark methods in the literature is a notable drawback. Merely comparing regret upper bounds does not provide a comprehensive understanding of the algorithms' actual performance. It would be more convincing to include extensive numerical experiments that directly compare the regret achieved by different algorithms, thereby justifying the usefulness of deriving entry-wise error of low-rank matrix estimation in bandit and RL settings.


**Questions:**

1. Could you elaborate on how deriving the entry-wise error of low-rank matrix estimation is justified in bandit and RL settings, providing detailed explanations and extensive numerical experiments?

2. In the discussion of related work, it is crucial to compare the paper's results with state-of-the-art approaches and provide a fair and objective analysis.



**Limitations:**

No numerical experiments were provided.

---

> ### Author Rebuttal · Authors · 2023-08-09
>
> Thank you for your valuable feedback and insightful review! Please find below our responses.
>
> *Answer to Weakness 1 and Question 1*
>
> **A. Necessity of entry-wise guarantees in low-rank bandits and RL.**  Thank you for bringing this point to our attention. Indeed, we did not explicitly motivate the need for entry-wise matrix estimation guarantees in low-rank bandits and RL. We will revise our manuscript accordingly. Below, we argue why entry-wise guarantees in low rank bandits and RL are useful.
>
> **A.1.** The need for entry-wise matrix estimation guarantees arises naturally in our analysis. In low-rank bandits, in order to obtain logarithmic regret bounds, we need control of the estimated gaps. This in turn requires entry-wise guarantees (see Appendix G.1 - Line 1090). In reward-free RL, in order to obtain PAC-bounds, we need to control the value difference error. This in turn requires control of matrix estimation errors in the norm $\Vert \cdot \Vert_{1 \to \infty}$ (see Lemma 12, Appendix A.3). Controlling this error is akin to controlling the entry-wise error in terms of analysis (see Appendix F).
>
> **A.2.**  Existing work on low-rank bandits showcases regret analysis that do not use entry-wise guarantees. However, these guarantees are only minimax while ours are gap-dependent, exhibit logarithmic scaling in the time horizon and even enjoy better minimax guarantees in some scenarios. Indeed, the work of Jun et al. (2019) (cited as [29] in our supplementary material) proposes a clever algorithm with a minimax regret guarantee scaling as $\tilde{O}((m+n)^{3/2} \sqrt{T})$. Their regret decomposition (Corollary 1 in [29]) suggests that both entry-wise guarantees and guarantees in Frobeinus norm yield the same results. The follow-up work of Kang et al. (2022) (cited as [32]) proposes an algorithm leveraging a novel estimator and attaining an improved minimax regret guarantees of order $\tilde{O}((m+n)\sqrt{T})$. In contrast, our work complements and, in some aspects, improves upon these results: **(i)** our analysis allows us to obtain logarithmic regret bounds which are gap-dependent $\tilde{O}((m+n)(\bar{\Delta}/\Delta_{\min}^2) \log^3(T))$. Both [29] and [32] do not provide gap-dependent guarantees with logarithmic regret; **(ii)** under the assumption that $\Delta_{\max}/\Delta_{\min} \le \xi$, our algorithm, SME-AE, achieves a minimax regret bound scaling as $\xi \sqrt{(m+n)T}\log(T)^2$. This is a better minimax guarantee than the ones presented in [29] and [32]; **(iii)** we also perform best-entry identification with provable guarantees (see Theorem 7). This problem remains at large under-explored.
>
> **A.3.** Existing work has also noted the importance and need of entry-wise guarantees but with limited success. Please refer the works of Bayati et al. (2022), Jain and Pal (2023), and  Shah et al. (2020) (cited as [7], [25], and [48], respectively, and discussed in Section 6 and Appendix H.2). [7] is the closest to our setting. There, they use a row enhancement procedure to obtain a matrix estimation guarantee in the norm $\Vert \cdot \Vert_{2\to \infty}$ (see Proposition 1 in [7]). They implicitly assume that one of the dimensions, say $m$ for example, is $\Theta(1)$ (see Assumption 1 and the discussion before it in [7]), thus making their guarantee equivalent to an entry-wise guarantee. Hence, their regret bounds (e.g., see Theorem 1 in [7]) may suffer an extra dependence on $\sqrt{m}$ while ours do not.
>
> Please note that we do not claim that entry-wise guarantees are absolutely needed but we don't see how to obtain our guarantees without.
>
> **B. Exploration in low-rank bandits and RL and entry-wise guarantees.**  The reviewer raises an exciting question which is that of optimal exploration under low rank structure. Unfortunately, this question is at large still open. Note however that our approach based on entry-wise guarantees already allows us to considerably trim the exploration process. Indeed, our algorithm, SME-AE, achieves a gap-dependent regret bounds of order $\widetilde{O}((m+n) (\bar{\Delta}/\Delta_{\min}) \log^3(T))$, thus showing that we do not scale with the number of arms $nm$. Also note that our algorithm, SME-AE, does not perform matrix estimation up to an arbitrary small entry-wise error but only up to the minimal gap $\Delta_{\min}$ which is necessary to obtain logarithmic regret.
>
> *Answer to Weakness 2 and Question 2.*
>
> **C. PAC guarantees in low-rank bandits and RL.** The reviewer commented  "In low-rank bandit and MDP scenarios, the focus should primarily be on minimizing regret." We believe that the problem of pure exploration in low-rank bandits and RL (a.k.a. PAC-RL) is also important. In low rank bandits, we have in fact also provided regret guarantees in addition to guarantees on best-entry identification. We believe this is a strength of our work rather than a weakness.
>
> **D. On related work.** First, we wish to clarify that we ended up putting the work of Kang et al. (2022) (cited as [32]) and Jain and Pal (2023) (cited as [25]) in Appendix H simply because we didn't have enough space in the main body. Moreover, we clearly stated in the main body (see line 318) that further details are provided on the related work (including [25] and [32]) in Appendix H. However, we understand the reviewer's sentiment that all relevant work should be mentioned in the main, especially those with state-of-the-art results. We thank the reviewer for bringing this to our attention and we will revise our manuscript accordingly.
>
> *Answer to Weakness 3.*
>
> **E. Experimental results.** We think that our theoretical contributions are numerous (matrix estimation under correlated noise and entry-wise guarantees + algorithms with better performance guarantees in low-rank bandits and RL). But, we agree that adding numerical experiments would nicely complement our theoretical findings. We intend to include experiments when revising our paper.

---

> > ### Comment · Reviewer_N4Ue · 2023-08-16
> >
> > Thanks for the detailed clarification. I have increased the score accordingly.

---

### Official Review · Reviewer_zbTi · 2023-07-06

**Soundness:** 4 excellent
**Presentation:** 3 good
**Contribution:** 3 good
**Rating:** 6
**Confidence:** 4

**Summary:**

The authors investigate matrix estimation in reinforcement learning and bandit settings with low-rank structures. They demonstrate the effectiveness of spectral-based methods in recovering matrix subspaces and minimizing entry-wise error. This enables the development of efficient RL algorithms tailored for low-rank bandits and Markov Decision Processes (MDPs).

This paper is solid, supported by robust results and technically intriguing proofs. It builds upon the recent trend of examining entry-wise error bounds for matrix completion problems and extends it to dynamic settings like RL and bandits. While the guarantees may not be surprising and the algorithms largely follow an explore-and-commit approach, the authors introduce an interesting technique using Compound Poisson Noise to transform a non-i.i.d. problem into an i.i.d. problem, which may have a larger implication for other applications.



**Strengths:**

- Theoretical Soundness
- Compound Poisson Noise Reduction for MDP is interesting
- Exploring the potential of entry-wise guarantees of matrix completion in a dynamic setting is in general interesting

**Weaknesses:**

While acknowledging the efforts invested in developing the theory, the novelty of the research appears limited given the well-established techniques of entry-wise recovery and leave-one-out. Moreover, the current algorithms are constrained to explore-and-commit and (pseudo) i.i.d. sampling schemas, compared to adaptive algorithms such as UCB, due to the limitations of the theoretical guarantees.

**Questions:**

How is the result for matrix completion with Poisson noise in this paper related to "Near-optimal entrywise anomaly detection for low-rank matrices with sub-exponential noise." In International Conference on Machine Learning, pp. 3154-3163. PMLR, 2021?

**Limitations:**

I do not perceive any immediate potential negative social impact.

---

> ### Author Rebuttal · Authors · 2023-08-09
>
> Thank you for your insightful review, and constructive feedback! We address your comments below.
>
>
> *Response to Weaknesses*
>
> **A. About the novelty of our analysis and of our algorithms.**
>
> **A.1.** As far as we are aware, the leave-one-out argument has been so far limited to a matrix plus noise model with independent noise entries. We feel that extending it to dependent noise entries was non-trivial. Notably, we establish entry-wise matrix estimation guarantees under Markovian noise structures (see our results for Model II in Section 3.3. and Appendix B.3)
>
> **A.2.** Note that our algorithm for the low-rank bandit problem is adaptive because it uses an adaptive stopping rule to end the exploration phase (this contrasts with naive Explore-Then-Commit (ETC) algorithms). We actually explain at the beginning of Section 4 that a naive ETC algorithm would lead to poor regret guarantees. Our algorithm remains simple, yet it achieves the best regret guarantees obtained so far.
>
> *Response to Questions*
>
> **B. Comparison with Farias et al., "Near-optimal entrywise anomaly detection for low-rank matrices with sub-exponential noise", ICML 2021.** Thank you for pointing us to this relevant paper, referred to as [0] below. We will cite it and discuss its results in the revised version of our manuscript. [0] provides entry-wise matrix estimation guarantees for random matrices with entries that are independent (see Theorem 2 in [0]). In contrast, we provide entry-wise matrix estimation guarantees for matrices with dependent entries (See our Theorems 1, 2 and 3). Both [0] and our submission rely on the leave-one-out argument for random matrices with sub-exponential entries. In our case, we use this argument as a sub-step of our analysis after performing the Poisson approximation. However, we believe that, as explained below, our final results are richer, more precise and actually needed for our RL applications.
>
> **B.1.** We provide guarantees on subspace recovery in $\Vert \cdot \Vert_{2\to \infty}$ (See (i) Theorem 1, 2, and 3) and guarantees in $\Vert \cdot \Vert_{1\to \infty}$ (see Theorem 2, and 3). This type of guarantees are not provided in [0]. The subspace recovery guarantees are useful as a sub-step of our analysis and we believe they are of independent interest. The guarantees in $\Vert \cdot \Vert_{1\to \infty}$ are useful in reward-free RL.
>
> **B.2.** The entry-wise guarantees of [0] are only expressed in terms of the matrix dimensions $n$ and $m$, while our guarantees exhibit a dependence on the number of observations $T$, the dimensions $n$ and $m$, and the confidence level $\delta$. Having guarantees with an explicit dependence for all  $T \ge 1$ and $\delta \in (0,1)$ is crucial in the design of our algorithm for low-rank bandits.
>
> **B.3.** At a more technical level, we cannot apply the proof of Theorem 2 from [0] to our setting unless the number of samples $T = \Omega(n\sqrt{n})$ in the homogeneous case, a far larger number compared to $T=\Omega( n\log n)$ in our setting. The authors of [0] require this condition in order to apply a well-known leave-one-out theorem (Lemma 6 in [0]) which requires $32 \kappa \max(\gamma,\phi(\gamma))\leq 1 $ where $\gamma$ and $\phi(\cdot)$ are quantities defined in the proof of Proposition 2 in [0]. Applied to our setting, this condition can be rewritten as $\sqrt{n \log (n)}(T\Vert M \Vert_{\infty} + 1)\kappa^2 \leq C T\Vert M \Vert_2$ (see the second assumption in Proposition 2 in [0]). In particular, for the homogeneous case, this condition requires $T = \Omega(n\sqrt{n})$ as mentioned above. Moreover, our proof uses tighter results (such as application of Bennett's inequality in our Lemma 29) which enables us to achieve very tight bounds (even at logarithmic scale).

---

> > ### Comment · Reviewer_zbTi · 2023-08-12
> >
> > Thank you for clarifying my questions!

---

### Official Review · Reviewer_j7uj · 2023-07-06

**Soundness:** 4 excellent
**Presentation:** 4 excellent
**Contribution:** 4 excellent
**Rating:** 8
**Confidence:** 3

**Summary:**

This paper provides a theoretical study of low-rank matrix estimation in three contexts:

**Context 1**: in the case of standard matrix estimation with uniformly sampled entries, where the authors prove a sample complexity comparable to approximate recovery results of $\tilde{O}(m+n)$ (when the rank is $O(1)$), with the following two main differences compared to existing methods:

a (makes the result easier). It is assumed the rank is known, and the function class restriction is based directly on an explicit low-rank assumption.
b (makes the result much harder and stronger). the error is measured with the maximum norm rather than Frobenius norm or expected loss.

Because of the general context of the paper which concerns mostly reinforcement learning (see other results below), these results are formulated as a reinforcement learning/bandit problem where the arms are arranged as pairs of integers and the matrix of rewards is assumed to be low-rank. However, note that independence of the noise components is required (cf. lines 689-693 in the supplementary).

**Context 2.1**

In this model, recovery guarantees are proved for estimating a low-rank transition matrix from observations of independent transitions. Here, the matrix must be square, with the dimension being equal to the number of states.


**Context 2.2**
In this more natural setting, we estimate the transition matrix from context 2.2 based only on full trajectories. Regret bounds proved in this case rely on the ergodicity of the Markov chain by splitting the expression of the average error over time into sums where the time steps are separated by a large number $\tau$, so that the distributions of the corresponding observations are approximately independent and distributed as the limiting distribution.



Finally, the authors introduce a novel algorithm for performing reinforcement learning in a low-rank bandit setting. The algorithm works in epochs: at a given epoch, with each iteration providing a more and more refined estimate of a set of candidate arms. Whilst the set of candidate arms has more than 2 elements, the algorithm recommends a uniformly random arm, after the set of candidate arms degenerates to a single element, the algorithm keeps recommending that one. A bound is shown on the expected amount of time it takes to identify the best arm. This culminates in a gap-independent regret bound of $\tilde{O}(T^{1/2})$ for this case, which to the best of my knowledge, solves a highly significant open problem in the nascent theory of low-rank bandits.



The proofs rely on an ingenious combination of existing ideas, with the most salient one being that of poison approximation: the matrix of observed rewards is approximated by a matrix with entries equal to $M_{i,j}T_i$ where $T_i$s are independent Poisson random variables. (this general technique has previously been introduced in [7)]. Then, concentration inequalities for such matrices are used in combination with bounds on the error in approximating the true matrix of observed rewards by this Poisson approximation.

==========Post-rebuttal=======

My doubts have been adequately addressed in the rebuttal and I will keep my score of **strong accept**. I remain convinced this is an excellent paper. However, I do hope the authors will take the trouble to add more detailed derivations and also tone down the statements and claims of novelty over the lack of independence of the $E_{i,j}$. Indeed, since this is an impressive contribution that many readers will be interested in studying, it would be a great service to the community to make the paper as approachable as possible.



========



**References**
[1] Nathan Srebro, "Rank, Trace-Norm and Max-Norm", COLT 2005.

[2] Ohad Shamir, Shai Shalev-Shwartz. Matrix Completion with the Trace Norm: Learning, Bounding, and Transducing. JMLR 2014.

[3] Yuxin Chen, Yuejie Chi, Jianqing Fan, Cong Ma, Yuling Yan. Noisy Matrix Completion: Understanding Statistical Guarantees for Convex Relaxation via Nonconvex Optimization. SIAM journal of Optimization, 2020.

[4] Prateek Jain, Soumyabrata Pal Online Low Rank Matrix Completion. ICLR 2023

[5] Sahand Negahban, Martin J. Wainwright. Restricted Strong Convexity and Weighted Matrix Completion: Optimal Bounds with Noise. JMLR 2012.

[6] Mitzenmacher, Michael and Upfal, Eli. Probability and computing: Randomization and probabilistic techniques in algorithms and data analysis. Cambridge UNiversity Press 2017.

[7] Yuxin Chen, Yujie Chi, Jianqing Fan, Cong Ma et al. Spectral Methods for data science: a statistical perspective.

**Strengths:**

**1** This is an excellent contribution to the field: although I am not completely sure about the originality (especially that of theorem  1 as compared to other results in [5]) since I am not fully up to speed on the reinforcement learning theory, I think there is a good chance this is groundbreaking work. The algorithm of successive arm elimination is simple and elegant, with a nearly optimal regret bound. Whilst it is true that the fact that explicit rank restriction is used (as opposed to, for instance, nuclear norm regularization) makes the matrix completion results seem a little trivial, the regret bound in the RL setting seems to be new and is a natural starting point to kick-start this branch of research.


**2** Aside from very minor things (mostly related to the contextualization of the work within the related works), this paper is excellently written. The proofs are very well organized, to an extent which is demonstrably superior to the majority of accepted papers I have reviewed.    The authors clearly organize the proof into steps (Poisson approximation, dilation trick, general concentration inequalities for Poisson matrices, specific calculations for the Poisson approximation to this problem, bounds on the error introduced by the approximation). Each parts of the proofs contain references to existing literature where relevant.

**Weaknesses:**

**1** One thing that threw me off a bit was the lack of contextualization and explanation of the first results (Theorem 1). In particular, on lines 90 to 92 (see also lines 125 to 130), it seems like the authors want to distance their results from standard matrix completion results by hinting that noise components are not independent (they are not stated to be independent at that point), when in fact it also seems this setting is exactly matrix completion with an explicit rank restriction. Indeed,  lines 689-693 in the supplementary appear to suggest independence **is** required. It would be quite appropriate to compare to existing results in this case as well. For a bounded loss, a similar sample complexity goes all the way back to [1] for approximate recovery with a bounded loss. For connected problems with the trace norm, we also have the result of [2] and the extremely impressive result of [3], which is much stronger than the results presented here due to improved dependence on the variance of the noise.

Similarly, I am also a bit confused by the statement on line 334 that "we do not assume the transition matrices are constructed based on a given restricted class of functions" since it appears there is a low rank assumption in the work.

**2**  Similarly, I feel like the related works on low-rank bandits missed some of the recent literature [4].

**3** In the low rank bandit result, the algorithm eventually recommends the same arm, that is, the same entry, all the time. This is highly restrictive in a recommender systems context: the algorithm isn't even able to identify the best item for each user, only the best user-item combination...







**References**
[1] Nathan Srebro, "Rank, Trace-Norm and Max-Norm", COLT 2005.

[2] Ohad Shamir, Shai Shalev-Shwartz. Matrix Completion with the Trace Norm: Learning, Bounding, and Transducing. JMLR 2014.

[3] Yuxin Chen, Yuejie Chi, Jianqing Fan, Cong Ma, Yuling Yan. Noisy Matrix Completion: Understanding Statistical Guarantees for Convex Relaxation via Nonconvex Optimization. SIAM journal of Optimization, 2020.

[4] Prateek Jain, Soumyabrata Pal Online Low Rank Matrix Completion. ICLR 2023

[5] Sahand Negahban, Martin J. Wainwright. Restricted Strong Convexity and Weighted Matrix Completion: Optimal Bounds with Noise. JMLR 2012.

[6] Mitzenmacher, Michael and Upfal, Eli. Probability and computing: Randomization and probabilistic techniques in algorithms and data analysis. Cambridge UNiversity Press 2017.

[7] Yuxin Chen, Yujie Chi, Jianqing Fan, Cong Ma et al. Spectral Methods for data science: a statistical perspective.

**Questions:**

**1** Could you clarify what you mean in lines 90 to 92 and lines 125 to 130 regarding the independence of the noise?
Although I can find such an example in the literature, it is seems unlikely to me that this result  (corollary 1) is new. Is there any similar result for matrix completion with the maximum absolute value of the error as the performance measure? Perhaps in [5]?

**2** Could you clarify to what extent theorem 1 and its proof differs from the existing results and proofs from [7] (what is the most similar result to Theorem 1 which was proved in [7])?

**3** ( I would really like to have the answer to this!) In lines 957 to 958, could you explain clearly ho you arrive at the equation on the bottom of page 33? I am a little stuck there, I need far more details than what is written as it seems you are using another equation from the same page without stating it. Also, did you really mean $\sigma_{2r}(M)$ in the denominator or did you mean $\sigma_r(M)$ instead (which is the quantity mentioned in the textual explanation on the line above)?

**4** Could you give a few more details in the page 32? It feels the reference to Lemma 4.16  in other work breaks the reader's momentum, it would be nice to write down the lemma again in the paper. Indeed, I think it is a key step in the argument that allows you to obtain this fascinating result with the max norm (theorem 1).

**5** Line 1098, second equation, do you mean $\ell+2$ instead of $\ell-2$ in the exponent


**6** In the second to last equation on line 1114 (page 42) should it not be $\lceil  \log_2(1/\Delta_{min})\rceil$ or $\log_2(e/\Delta_{min})$ instead of $\log_2(1/\Delta_{min})$? Again, I see that you are not being too careful about constants in general but there is no $O$ notation in this particular inequality.


**7** In line 90, you say that the entries are sampled "uniformly at random for simplicity", do you in fact require this assumption in the proof of the theorem?  (based on existing results for matrix completion with explicit low-rank assumptions, it would seem that this is not required. In general, it would be nice to add a bit more details to each of the theorem statements to clarify the assumptions.


**8** Is Lemma 25 known? It seems very general and standard, I feel like it is probably in some well known book.












**Minor comments/typos**

A citation would be nice on lines 226-227

It could be nice to reorganize sections C1 and C2.2 a little bit. For instance, in line 707, you are using a lemma which comes from a subsection that concerns the Markov problem  when proving a result about Model 1, it would be nice to make it clear that Lemma 22 doesn't, in fact, require any proof elements from the context of Model 2.

In the same vein, I think it would be nice, for completeness, to write down the proof of Lemma 22, even if it is very similar to the proof of Theorem 5.7 in [6]. It is a bit funny that the proof of Lemma 21 is there but not that of Lemma 22 since Lemma 21 is pretty much trivial whilst Lemma 22 is a bit more advanced. Making the paper fully self contained would be a nice way to get even more citations by further improving the encylopaedic quality of the appendix.



Line 1098 (page 41), second equation, I think you mean $4e$ not $2\sqrt{e}$, though I understand you are not being too bothered about constants given the final results are in $O$ notation


In the last inequality of the sequence of inequalities on page 43, and in line 1124 I think it would be nice to remind the reader of the definition of $\bar{\Delta}$ from line 239, since it is instrumental in deriving the equality on line 1124, which unfortunately relies on the fact that the recommendations are uniformly at random in $[m]\times [n]$ instead of the more natural $\mathcal{A}_{\ell}$.



Typos:

Line 951: missing "the"

Line 706: extra square bracket  inside the equation.

In line 727, I recommend adding more details for ho you prove that $\tau(\epsilon)\leq \tau$, it takes quite a bit of massaging of previous equations to get this with the correct constant!

Line 741: be joint probability==> be the joint probability

line 742 "for Poisson and multinomial model ==> "for the Poisson and multinomial models"

line 793 (proof of Lemma 25): "variables" ==> "variable"

line 957 add "following" before "condition

Line 980, Model\ref{10} should probably be Model\eqref{10}

line 1097  (page 41) You say " in order for the above guarantee to hold we require the following two conditions to hold. I think that is not technically true: the conditions below are what is needed to apply the specific previously established results, so technically those are sufficient rather than necessary conditions.


Line 1114 (page 42):  I think there was an error with copy and paste: "rounds" should be after $\tau$ and before the first, not the second comma of the line.

In the first equality after line 1122, using \left and \right might be nicer.


[1] Nathan Srebro, "Rank, Trace-Norm and Max-Norm", COLT 2005.

[2] Ohad Shamir, Shai Shalev-Shwartz. Matrix Completion with the Trace Norm: Learning, Bounding, and Transducing. JMLR 2014.

[3] Yuxin Chen, Yuejie Chi, Jianqing Fan, Cong Ma, Yuling Yan. Noisy Matrix Completion: Understanding Statistical Guarantees for Convex Relaxation via Nonconvex Optimization. SIAM journal of Optimization, 2020.

[4] Prateek Jain, Soumyabrata Pal Online Low Rank Matrix Completion. ICLR 2023

[5] Sahand Negahban, Martin J. Wainwright. Restricted Strong Convexity and Weighted Matrix Completion: Optimal Bounds with Noise. JMLR 2012.

[6] Mitzenmacher, Michael and Upfal, Eli. Probability and computing: Randomization and probabilistic techniques in algorithms and data analysis. Cambridge UNiversity Press 2017.

[7] Yuxin Chen, Yujie Chi, Jianqing Fan, Cong Ma et al. Spectral Methods for data science: a statistical perspective.

**Limitations:**

See "weaknesses", especially Weakness 3. In addition, it seems that the algorithm is a little unnatural in that it doesn't use the successive estimates of the set of candidate arms and merely recommends uniformly random entries amongst all arms until it is confident it has identified the best arm overall. It would be interesting to see if it is possible to prove similar guarantees for the algorithm which recommends a random arm in the set $\mathcal{A}_{\ell}$ instead of ($\mathcal{A}_1=[m]\times [n]$) at each time step $\ell$.

---

> ### Author Rebuttal · Authors · 2023-08-09
>
> Thank you for your careful review and very positive feedback.
>
> **A. Answer to Weakness 1 \& Question 1.** Thanks for mentioning these papers; we will cite them. We clarify below the differences between these papers and our contributions for Model I.
>
> **A.1.** [1, 2, 5] (see the refs in your review) do not provide entry-wise guarantees for the problem of matrix estimation with low-rank constraint under any noise assumptions.
>
> **A.2.** [3] provides entry-wise guarantees for a nuclear norm penalized estimator. [7] surveys entry-wise guarantees for spectral methods. The data model considered in these references is the matrix plus noise model, i.e., $M + E$, where the noise matrix, $E$, has independent entries. The setting we describe in Model I is different: if we write it in the form of a matrix plus noise model, we obtain that the noise matrix $E$ can be expressed as
>     $$\forall (i,j)\in [n]\times [m], \quad  E\_{i,j} =\widetilde{M}\_{i,j}-M\_{i,j} = \left(\frac{nm}{T}\sum\_{t=1}^T (M\_{i_t, j_t} + \xi\_t) 1\_{\lbrace (i_t, j_t)=(i,j)\rbrace}\right)-M\_{i,j}$$
> where we use the expression of $\widetilde{M}$ provided in eq. (10) in Appendix C.1. From this, we clearly see that the entries of $E$ are not independent. Therefore, the entry-wise guarantees of [3, 7] do not apply in our setting. In fact, the leave-one-out argument relies heavily on the independence of the noise entries.
>
> **A.3.** In [4], the authors have a model where $m$ entries are observed per round, whereas in Model I, we observe one entry only. In addition, to obtain entry-wise guarantees, they resort to a couple of algorithmic tricks, explained in Remarks 2, 3, and 4 of [4]. In contrast, we show that simpler spectral methods enjoy entry-wise guarantees without any additional tricks.
>
> **B. Answer to Weakness 2 -- Regarding [4].** We actually cited this work in the related work (ref. [25] in our supplementary material), but only discussed it in Appendix H.2. We will discuss it further in the main part of the paper.
>
> **C. Answer to Weakness 3.** Indeed, our algorithm eventually recommends the same (but provably optimal) arm/matrix entry. When devising SME-AE, we had the applications mentioned in the introduction of Bayati et al. [6] (ref. in our supplementary material), where recommending the same arm is not a problem. This could become a problem in other types of recommender systems. We believe that our approach can be extended to these systems.
>
> **D. Answer to Question 2.** The results in [7] that are the closest to our Theorem 1 are Theorems 4.2 and 4.5 (Chapter 4). However, they are not really comparable to our results. Indeed:
>
> **D.1.** In [7], the results for entry-wise guarantees are for a matrix plus noise model with independent entries -- see our answer A.2. to Weakness 1 for details. In our setting, the leave-one-out argument of [7] breaks. Our proof of Theorem 1 introduces new techniques such as the Poisson approximation, needed to deal with dependencies in the noise model.
>
> **D.2.** The leave-one-out argument in [7] is only valid when the entries are sub-Gaussian. For our purposes, the observed matrix has entries distributed as Compound Poisson r.v.. Hence, we had to use different concentration inequalities than those of [7]. In particular, we needed a Truncated version of matrix Bernstein inequality (see Proposition 27 and Lemma 29 in App. D).
>
> **D.3.** Finally, the results in [7] only focus on how the guarantees scale with matrix dimensions $n$ and $m$. In addition, we care about how they scale with the confidence level $\delta \in (0,1)$, and the number observations $T$. Tracking these dependencies is not trivial.
>
> **E. Answer to Question 3.** Thank you bringing this to our attention. We will detail this part in our manuscript. In the meanwhile, we clarify below the key steps leading to the equations lines 957 to 958. First, we wish to disclose a typo in the definition of $\widetilde{S}^{\ell}$. The correct definition is:
> $$\forall i,j \in [n+m], \qquad  \widetilde{S}^{(\ell)}\_{i,j} = \begin{cases} \widetilde{S}\_{i,j} & \text{if } i \neq \ell \text{ or }  j \neq \ell \\\\ S\_{i,j} & \text{otherwise}\end{cases}$$
> We can first establish $$\Vert\widehat{Q}W_{\widehat{Q}}-\widehat{Q}^{(\ell)} W_{\widehat{Q}^{(\ell)}} \Vert_F\le\Vert\widehat{Q}\widehat{Q}^\top -  (\widehat{Q}^{(\ell)})(\widehat{Q}^{(\ell)})^\top\Vert_F\le\frac{4\Vert(\widetilde{S}-\widetilde{S}^{(\ell)})\widehat{Q}^{(\ell)}\Vert_F}{\sigma_{2r}(S)}$$
> where the first inequality follows by definition of $W$ and the second follows from Davis-Kahan's inequality. Next, we have
> $$\Vert(\widetilde{S}-\widetilde{S}^{(\ell)})\widehat{Q}^{(\ell)}\Vert_F\le\Vert E_{\ell,:}\widehat{Q}^{(\ell)}\Vert_2+2\Vert E\Vert\Vert \widehat{Q} W_{\widehat{Q}}\Vert_{2\to\infty}+2\Vert E\Vert\Vert\widehat{Q}W_{\widehat{Q}}-\widehat{Q}^{(\ell)}W_{\widehat{Q}^{(\ell)}}\Vert_{2\to\infty}$$
> where the inequality follows by definition of $\widetilde{S}^{(\ell)}$ and a couple of triangular inequalities and norm definitions. From the inequalities above and with the assumption that $\Vert E\Vert\le 16\sigma_r(M)=16\sigma_{2r}(S)$, we recover the inequality in lines 957 to 958. Moreover, yes, it should be $\sigma_r(M) $ (or $\sigma_{2r}(S)$) in the inequality.
>
> **F. Answers to Questions 4 to 8.** Thank you for the many suggestions and thorough reading of our manuscript. We will add Lemma 4.16 for completeness instead of simply citing it. Yes, in line 1098,  $\ell - 2$ should be instead $\ell + 2$. In line 1114, $\log_2(1/\Delta_{\min})$ in the fourth inequality should be instead $\lceil \log_2(1/\Delta_{\min}) \rceil $ which we bound later by $\log_2(e/\Delta_{\min})$. Regarding sampling uniformly at random, we can relax this at the expense of technical clarity and presentation simplicity. We will provide more details for our theorems to further clarify their assumptions. Regarding Lemma 25, we ended up proving it because we couldn't find a similar result in the literature.

---

> > ### Comment · Reviewer_j7uj · 2023-08-18
> > **Addressed. Congratulations. Please add substantially more details in the derivations in the final version.**
> >
> > Many thanks for your many clarifications, and for agreeing to fix the typos.
> >
> > Regarding the lack of independence of the entries of $E_{i,j}$ in your setting, I now understand what you mean, thank you. However, it certainly doesn't feel very significant, since you still have independence of the $\xi_t$ for various $t$s. It is a classic issue in matrix completion that we need to be careful with repeated entries, as you are there, but it is usually just (highly) technical. However, I am still very convinced by the overall significance of the work and the other results. Still, it would be nice to incorporate part of this rebuttal in the paper to better explain what you mean there.
> >
> > Regarding A1, I understand what you mean. However, the concept of "entry-wise" isn't completely standard, and it is still worth comparing. I personally think it is nice, and doesn't diminish the value of your work at all, that the bounds scale similarly to those works, since your bounds are entry-wise (with a max over entries) and therefore much stronger.
> >
> > Thanks for the clarification for question 3 as well, and for the fantastic paper and results. I incline to keep my score.

---

### Official Review · Reviewer_JAik · 2023-07-09

**Soundness:** 3 good
**Presentation:** 3 good
**Contribution:** 3 good
**Rating:** 6
**Confidence:** 3

**Summary:**

This paper studies two problems in RL involving low-rank matrix estimation. It provides entry-wise estimation error bounds for simple spectral methods in low-rank bandits and low-rank MDP under different sampling mechanisms. Based on these, it provides performance guarantees for two algorithms designed for low-rank bandit and low-rank MDP respectively.

**Strengths:**

The paper makes contribution towards the estimation of low-rank matrices in the setting of bandits and MDP respectively, as well as how these results can be useful in devising efficient algorithms to make full use of the low-rank structure.

Specifically, the paper considers the challenging case of low-rank matrix estimation in the presence of dependent noise and provides sharp entry-wise estimation error bounds using involved techniques. This is likely to help the development of algorithms which are better fitted for low-rank settings.

**Weaknesses:**

1. Compared with standard matrix completion problem, Model I in this paper extends to the case of dependent noise, but it restricts the magnitude of noise terms $\xi_t$ to be upper bounded by $c_1\Vert M\Vert_{\infty}$ (see Page 3 Line 91). Therefore, the results here can’t reduce to the state-of-the-art results in matrix completion with independent noise, where the noise level can be orderwise larger than $\Vert M\Vert_{\infty}$ [1]. We suggest the authors to discuss why their condition becomes more stringent when the noise are independent.

2. There are a few typos in the paper. For example:

On Page 2 Line 71, it would be better to avoid using acronyms list “wlog”.

On Page 7 Line 224, “where in a first phase” —> “where in the first phase”

On Page 5 Line 175, “is a small when“

[1] Chen, Yuxin, Yuejie Chi, Jianqing Fan, and Cong Ma. "Spectral methods for data science: A statistical perspective." Foundations and Trends® in Machine Learning 14, no. 5 (2021): 566-806.

**Questions:**

On Page 7 Line 220, it is assumed that the low-rank bandit has an homogeneous rank-$r$ reward matrix M, while this is not mentioned in the corresponding result Theorem 7. This makes the setting kind of confusing. Can you provide more explanations on this?

**Limitations:**

This paper does not have potential negative social impact.

---

> ### Author Rebuttal · Authors · 2023-08-09
>
> Thank you for your valuable review and positive feedback! Please find below our responses.
>
> *Answer to Weakness 1.*
>
> **A. Relaxing the assumption on the noise upper bound.** We would like to thank you for highlighting this difference between our setting and those for matrices with independent noise (ex. Chen et al. (2021) cited as [14] in our supplementary material) and we think this is an interesting question that should be addressed in the updated version of the paper. However, we would like to stress the following two points:
>
> **A.1** First, the main focus of this paper has been obtaining error bounds with near-optimal scaling with respect to problem dimensions $n,m,T$, and addressing the problem of dependence between different entries of the observation matrix. Hence, in the analysis for Model I, we made a compromise by upper-bounding the standard deviation of the noise by $c \|\|M\|\|\_{\infty}$ for some universal constant $c > 0$ and simplifying the analysis - see Lemma 25 where we define $L = \max (\|\|M\|\|\_{\infty},\sigma)$.
>
> **A.2** We agree that state-of-the-art results for matrix recovery with independent noise have weaker assumption for the noise magnitude, but these results are applicable only to models with independent noise. The results we presented should not be reduced to independent setting, but considered as a first step to obtaining matrix recovery guarantees for matrices with dependent entries.
>
> *Answer to Weakness 2 -- Typos.*
>
> Thanks a lot for noticing. We will correct these typos.
>
> *Answer to Questions.*
>
> **B. On the setting of Theorem 7.** The assumption of homogeneous rank-$r$ reward matrix $M$ stays valid in the entirety of Section 4 and thus we also assume it when presenting Theorem 7. We will clarify this. Actually, we could remove this assumption, but at the expense of clarity and simplicity of the results. More precisely, to remove the assumption, we just need to use the explicit expressions of the  constants $c_1, C_1$ in the analysis. These constants depend on $\mu$, $\kappa$, $r$, $m$, and $n$ as described Line 1093 in Appendix G.

---

> > ### Comment · Reviewer_JAik · 2023-08-20
> >
> > Thank you for the clarification! I would like to maintain my initial evaluation to the paper.

---

### Decision · Program_Chairs · 2023-09-21

**Decision:**

Accept (poster)

**Comment:**

In this paper, the authors studied the effectiveness of spectral methods for two problems in reinforcement learning: low-rank bandits and low-rank MDP under different sampling mechanisms. Sharp entrywise estimation error bounds have been developed despite the presence of dependent noise. While entrywise statistical analysis has been widely conducted in the statistics literature, it was previously rarely available in the RL literature.  As one reviewer pointed out, it would be great if the authors could extend their theory to accommodate an enlarged range of noise levels.  A more detailed discussion about the necessity of deriving entrywise estimation bounds in these two RL contexts would also strengthen the paper.